# Longevity interventions modulate mechanotransduction and extracellular matrix homeostasis in *C. elegans*

Alina C. Teuscher[1,8], Cyril Statzer[1,8], Anita Goyala[1], Seraina A. Domenig[1], Ingmar Schoen [2,3], Max Hess[1], Alexander M. Hofer [1], Andrea Fossati [4,5], Viola Vogel [3], Orcun Goksel [6,7], Ruedi Aebersold [4] & Collin Y. Ewald [1] ✉

Dysfunctional extracellular matrices (ECM) contribute to aging and disease. Repairing dysfunctional ECM could potentially prevent age-related pathologies. Interventions promoting longevity also impact ECM gene expression. However, the role of ECM composition changes in healthy aging remains unclear. Here we perform proteomics and in-vivo monitoring to systematically investigate ECM composition (matreotype) during aging in *C. elegans* revealing three distinct collagen dynamics. Longevity interventions slow age-related collagen stiffening and prolong the expression of collagens that are turned over. These prolonged collagen dynamics are mediated by a mechanical feedback loop of hemidesmosome-containing structures that span from the exoskeletal ECM through the hypodermis, basement membrane ECM, to the muscles, coupling mechanical forces to adjust ECM gene expression and longevity via the transcriptional co-activator YAP-1 across tissues. Our results provide in-vivo evidence that coordinated ECM remodeling through mechanotransduction is required and sufficient to promote longevity, offering potential avenues for interventions targeting ECM dynamics.

The extracellular matrix (ECM) plays a crucial role in supporting cellular function and tissue integrity[1–3]. Composed of various proteins, including collagens, glycoproteins, and proteoglycans, the ECM provides mechanical support, stores growth factors, and regulates cellular homeostasis[1–4]. Aging is characterized by alterations in ECM structure and function[5,6]. In the absence of disease, a key signature of aging is the decline of expression of collagens, the major components of the ECM, in murine, rat, and human skin[7,8], whereas collagen-rich basement membrane ECM increases 100-fold in humans during aging[9]. Some collagens are synthesized once early in life and stay incorporated in the

ECM life long and can therefore become damaged with age[10], whereas other collagens are extensively remodeled and renewed within circadian rhythm[11]. These broad and different changes in tissue-specific ECM compositions during aging have been documented using proteomics approaches[5,12,13].

In addition, changes in ECM composition have been implicated in numerous diseases[14,15], highlighting the importance of understanding the dynamic nature of the ECM. For instance, cancer types can be identified, and adverse patient outcomes can be predicted based on their ECM composition, and circulating tumor cells can be identified

[1]Laboratory of Extracellular Matrix Regeneration, Institute of Translational Medicine, Department of Health Sciences and Technology, ETH Zürich, Schwerzenbach CH-8603, Switzerland. [2]School of Pharmacy and Biomolecular Sciences, Irish Centre for Vascular Biology, Royal College of Surgeons in Ireland, Dublin 2, Ireland. [3]Laboratory of Applied Mechanobiology, Institute of Translational Medicine, Department of Health Sciences and Technology, ETH Zürich, Zurich, Switzerland. [4]Department of Biology, Institute of Molecular Systems Biology, ETH Zürich, Zurich, Switzerland. [5]Department of Cellular and Molecular Pharmacology, University of California San Francisco, San Francisco 94158 CA, USA. [6]Department of Information Technology and Electrical Engineering, ETH Zürich, Zürich, Switzerland. [7]Department of Information Technology, Uppsala University, Uppsala, Sweden. [8]These authors contributed equally: Alina C. Teuscher, Cyril Statzer. ✉e-mail: collin-ewald@ethz.ch

based on their ECM gene expression[16–19]. Surprisingly, single-cell RNA sequencing data of ECM gene expression predicts cell type and development stage[20]. Thus, ECM composition does reflect cell identity, phenotypic state, health, or disease status. To conceptualize this, we defined this as the 'matreotype', which is a 'snapshot' of the ECM composition associated with or caused by a phenotype or physiological state, such as health, disease, or age[5]. Using RNA sequencing data, we have defined the youthful matreotype of humans, probed changes in gene expression upon drug treatment, and thereby predicted and validated several novel longevity drugs[21]. This illustrates that the matreotype has broad implications for biomedical research[22], yet the links between changes in ECM composition and the impact on aging are unclear.

Longevity-promoting interventions, both pharmacological and genetic, have been shown to influence ECM gene expression in *C. elegans*, mice, and humans[7,21,23]. Using *C. elegans*, we have previously shown that the effects of all so-far tested longevity interventions are abolished when non-essential collagens *col-10*, *col-13*, or *col-120*, are knocked down during adulthood[24]. This suggests that these three collagens act downstream of diverse longevity interventions or function as a licensing signal for longevity. Conversely, overexpressing any of these three collagens is sufficient to increase *C. elegans*' lifespan[24]. However, why these collagens are important and the underlying mechanism linking collagen remodeling to longevity remained unknown.

Here we monitor matreotypes (*i.e.*, ECM composition) encompassing from expression to protein to incorporation and maintenance in the matrix in vivo during aging. We show that certain cuticular collagens become crosslinked with aging, but others are remodeled out of the ECM during aging. Remarkably, we identify via three independent approaches a longevity-induced feedback loop controlling ECM remodeling: proteomics, a genetic screen for regulators of collagen homeostasis during old age, and lifespan assays performed on all major ECM categories of more than fifty-five thousand *C. elegans* all point to a hemidesmosome-containing functional unit that spans from the exoskeletal ECM through the hypodermis and basement membrane ECM to the muscles. Mechanistically, we show in vivo that hemidesmosome integrity is required for longevity by coupling mechanical forces to adjust ECM gene expression across tissues via transcriptional co-activator YAP-1. We demonstrate that an age-dependent uncoupling of mechanotransduction abolishes the feedback, thereby inhibiting prolonged ECM protein homeostasis and longevity. Thus, we provide mechanistic evidence that mechanocoupling or mechano-transduction are essential for promoting healthy aging.

## Results
### ECM composition changes during aging
We previously in-silico defined the matrisome in *C. elegans*, which are all proteins that form the ECM, such as 181 collagens, 35 glycoproteins, 10 proteoglycans, and 481 extracellular proteins that either associate with ECM or remodel the ECM[25]. ECM remodeling starts with proteases, which excise and degrade proteins from the ECM. Excised proteins are then replaced by de novo synthesized ECM proteins that are secreted and incorporated into the matrix with the help of proteases and cross-linking enzymes[26]. To capture this process, we assessed matrisome and adhesome dynamics by fluorescent reporters and performed a quantitative whole proteome analysis along an aging timeline, and combined them with previously published omics data on five different levels: (1) gene expression via RNA sequencing, (2) timing and localization of expression via promoter reporters in vivo, (3) matrisome protein levels via quantitative proteomics, (4) de novo synthesis of matrisome proteins based on SILAC-label-chase proteomics data, and (5) monitoring of selected matrisome proteins tagged with fluorescent proteins incorporated into the ECM in vivo during aging (Fig. 1 and Supplementary Fig. 1).

As expected, due to the technical limitations of each of these five approaches, the coverage from mRNA to actual protein in the ECM was incomplete; however, some patterns emerged. Starting in early adulthood, the majority of cuticular collagen (*col*) mRNA levels steeply declined during aging, accompanied by a decline in protein levels, de novo synthesis, and levels in the ECM (Fig. 1a–c, Supplementary Figs. 1, 2, Supplementary Data 1-3). The mRNA, protein, and de novo levels of the adhesome and some conserved collagens were unchanged and continuously expressed during aging (Fig. 1d–f, Supplementary Fig. 1i, Supplementary Data 1, 2). By contrast, the only category of matrisome genes that increased during aging were proteases and protease inhibitors that remodel the ECM, such as MMP/*zmp*, astacin metalloprotease/*nas*, cathepsin/*cpr*, and protease inhibitor/cystatin/*cpi* (Fig. 1g–i, Supplementary Fig. 1i, Supplementary Data 1–3).

Collagens make up the majority of proteins in the ECM[27]. During *C. elegans* matrix synthesis, collagens forming functionally distinct ECM substructures are temporally co-expressed, suggesting that interacting collagens cluster together[28]. Out of the 181 *C. elegans* collagens[25], we were able to assess the quantitative abundance data for 41 collagens proteins and mRNAs during aging (Supplementary Data 1). For these collagens, we observed three distinct dynamic patterns (I-III) during aging (Fig. 1j–o, Supplementary Data 1). Pattern I consists of 21/41 detected collagens for which the mRNA, protein levels, and abundance in the ECM steeply declined during early adulthood (*e.g., col-10, col-12, col-13, col-120, col-144*; Fig. 1j, k, Supplementary Fig. 1i, j, Supplementary Data 1–3). Pattern II consists of 6/41 detected collagens for which the mRNA steeply declined in early adulthood, but the protein levels and/or abundance in the ECM stayed unchanged or increased during aging (*e.g., col-19*; Fig. 1l–n, Supplementary Fig. 1i, j, Supplementary Data 1–3). Pattern III consists of 14/41 detected collagens for which the mRNA remained unchanged or mildly declined, but the protein levels and/or abundance in the ECM stayed increased during aging (*e.g., emb-9*; Fig. 1m–o, Supplementary Fig. 1i, j, Supplementary Data 1–3).

### Reduced Insulin/IGF-1 receptor signaling prolongs collagen pattern I expression during aging
Next, we asked which of these three dynamic collagen patterns are altered in vivo upon longevity interventions. To slow aging, we used *daf-2(RNAi)* to reduce Insulin/IGF-1 receptor signaling (rIIS). For pattern I, using promoter-driven transgenic animals, we observed that *daf-2(RNAi)* prolonged the expression of *col-120* mRNA during aging (Supplementary Fig. 3a, b, Supplementary Data 4). While COL-120 protein tagged with GFP gradually disappeared from the cuticular ECM during aging, slowing aging by rIIS showed COL-120 in the ECM for a prolonged time (Fig. 2a, b, Supplementary Fig. 3c, d, Supplementary Data 4). Similar dynamics were observed with other collagens from pattern I (Supplementary Fig. 3e-j, Supplementary Data 4).

To assess protein turnover, we tagged COL-120 with the photoswitchable fluorophore Dendra[29,30]. Because COL-120 starts to disappear from the ECM in early adulthood, we irreversibly photoconverted COL-120::Dendra from green to red fluorescence on day 2 of adulthood, let the animals age for two more days, and then assessed green versus red fluorescent COL-120::Dendra in the cuticle (Fig. 2c). If, during these two days, no new COL-120 would be synthesized, then the photoconverted area would stay red. If all photoconverted COL-120 were replaced (*i.e.*, turned over), the photoconverted area would turn green. We found that the photoconverted areas mostly stayed red, but new COL-120 collagens (in green) were added on top of the older COL-120 (in red; Fig. 2d, Supplementary Data 5). Since COL-120 levels gradually declined in the ECM during this time period, we included the levels of the outside regions to subtract this general decline from quantification in the photoconverted area. We found that the old COL-120 disappeared faster from the ECM than the new COL-120 was added (Fig. 2e,

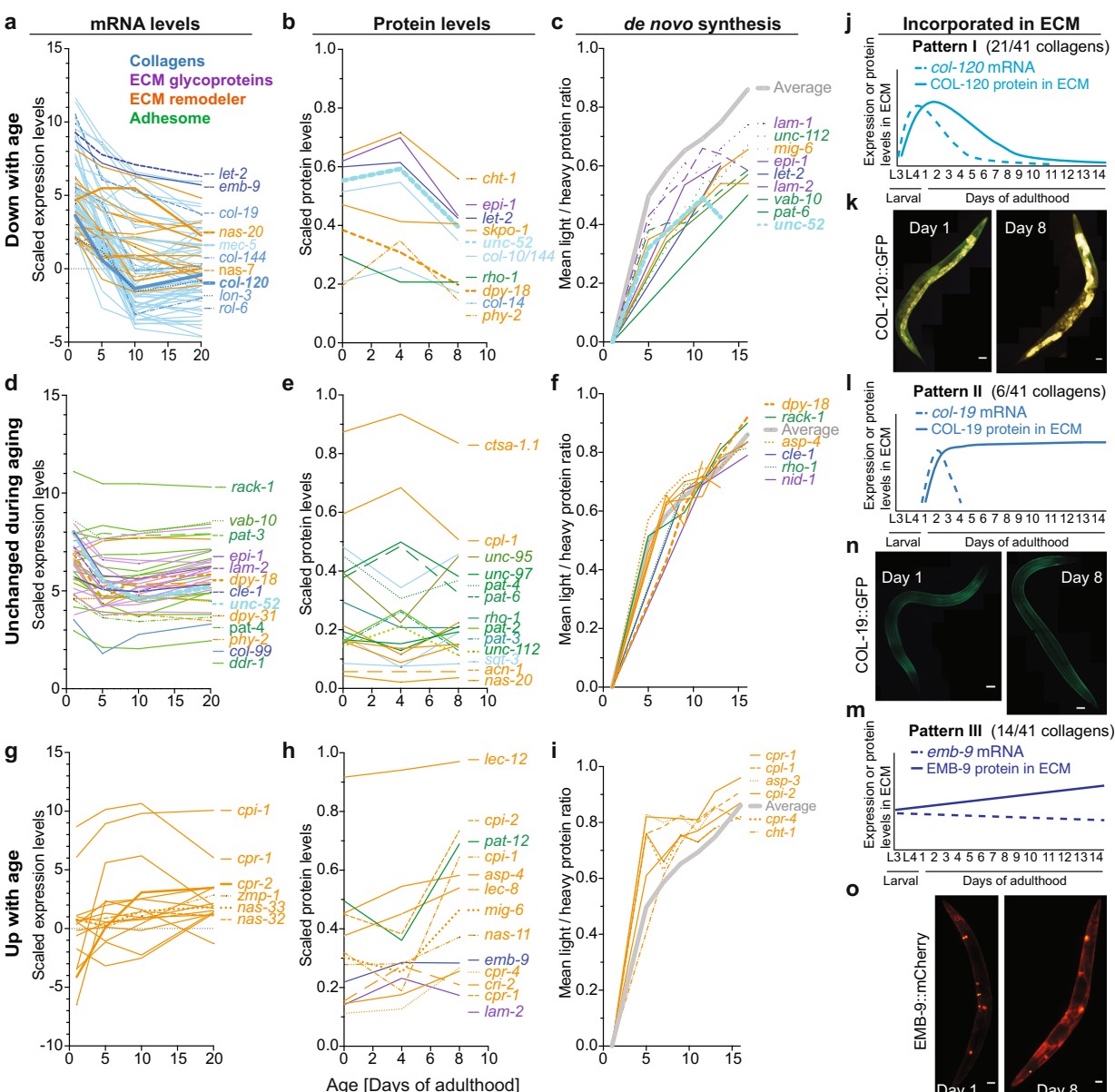

**Fig. 1 | Matrisome dynamics during aging. a, d, g** aging time course of matrisome mRNA levels were classified as down- (**a**), unchanged (**d**), or up-regulated (**g**) during aging based on the agreement between the Pearson and Spearman correlation coefficients when applied to the individual samples. (Source data: GSE46051, Supplementary Data 2). **b, e, h** Quantitative whole proteome analysis aging time course matrisome protein levels declining (**b**), unchanged (**e**), or increasing (**h**) during aging, taking into account the aging correlation of the individual samples (Data and statistics Supplementary Data 3). **c, f, i** Classification of the de novo protein synthesis rate of each protein into lower (**c**), unchanged (**f**), or elevated (**i**) during aging based on the difference between the synthesis rate of the individual protein to the overall mean protein production. (Source data: PMID: 25686393, Supplementary Data 3). **j–o** The aging time course of collagens that are incorporated into the ECM. **j, l, m** Model extrapolated from transgenic fluorophore tagged collagens shown as representative images of one independent biological trial (**k, n, o**). Note in **k**, the strong yellow-brownish fluorescence is autofluorescence from gut granules (see Methods for details). COL-120::GFP is in green localized in the cuticular ECM. Details in Supplementary Fig. 1, Supplementary Data 1. Scale bar = 50 μm.

Supplementary Data 5). Slowing aging by *daf-2(RNAi)* enhanced and prolonged the addition of newly synthesized COL-120 onto the old COL-120 in the ECM (Fig. 2e, Supplementary Data 5). This suggests that reducing Insulin/IGF-1 receptor-mediated longevity intervention counteracts the gradual loss of COL-120 from the cuticle by simply adding newly synthesized collagens to the older collagens that are continuously excised out of the cuticle.

**Age-dependent loss of mechanical tension of stably intercalated pattern II collagens is rescued by *glp-1*-induced longevity**

While the *col-19* mRNA declined rapidly during early adulthood, the GFP-tagged COL-19 stayed incorporated during aging, making COL-19

a representative member of the pattern II collagens. Upon *daf-2(RNAi)*, the cuticular COL-19 protein levels in the ECM compared to control remained unchanged during aging (Supplementary Fig. 3i, j, Supplementary Data 4), suggesting that these collagens once synthesized and incorporated into the cuticle would stay lifelong in this ECM.

During aging, collagens that are not replaced accumulate advanced glycation end products (AGEs), leading to crosslinking of collagens and stiffening of ECM[5]. Isolated *C. elegans* cuticles become stiffer with age[31] and show a marked increase in fluorescent spectral peaks reminiscent of AGE[32]. Based on this and our observation that COL-19 protein stayed incorporated in the ECM during aging, we hypothesized that COL-19 might become crosslinked, thereby altering

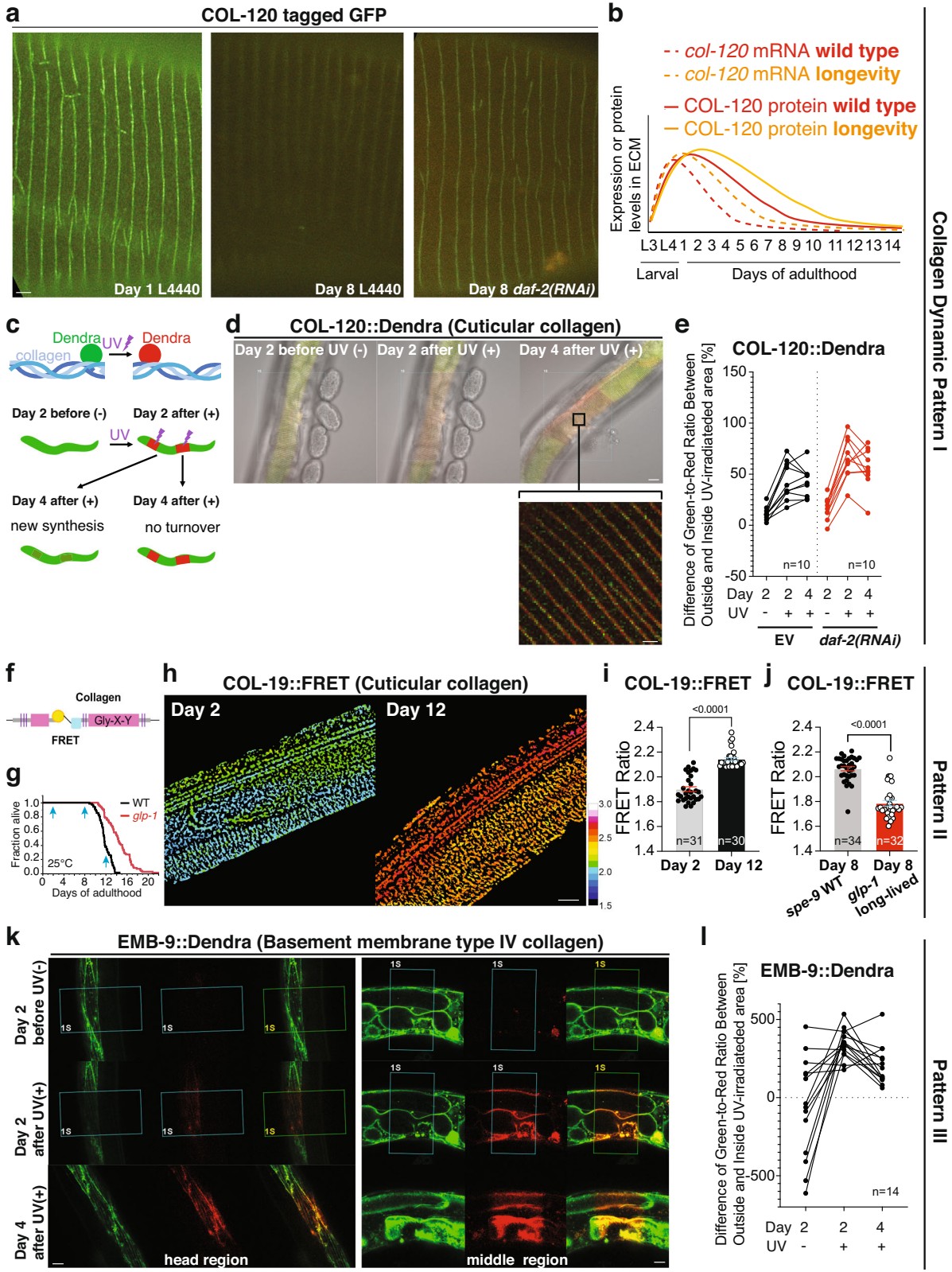

mechanical properties. To test this, we used a Förster resonance energy transfer (FRET) sensor incorporated into COL-19 (Fig. 2f) that has previously been used to read out mechanical stress and forces[33]. As expected for a mechanosensor, the FRET transmission of this COL-19::FRET increased when the animals were compressed between two coverslips (Supplementary Fig. 4a). To avoid external forces, we built a flow chamber for FRET measurements (Supplementary Fig. 4b).

We scored FRET ratios when animals were young (day 2 of adulthood), old but before death events occurred (day 8), and at very old age, when about 75% of the population had died (day 12; Fig. 2g, Supplementary Fig. 4c–f). We found a stark increase in FRET transmission during old age (day 12) compared to young (day 2), and this age-dependent increase in FRET transmission was lower in long-lived *glp-1* mutants at day 8 of adulthood compared to normal-lived wild-type controls

**Fig. 2 | Three patterns of collagen dynamics during longevity quantified in vivo.** **a**–**e** COL-120 dynamics as a representative for the pattern I collagen. **a** Collagen COL-120 tagged with GFP (LSD1000) in the cuticular ECM furrow vanished from day 1 to day 8 on control empty vector RNAi (L4440) but was still visible at day 8 of adulthood when *daf-2(RNAi)* started at L4. Representative images of one independent biological trial. Scale bar = 1 μm. **b** Model of COL-120 mRNA, protein, and incorporated ECM reporter dynamics. **c** Experimental workflow and the expected outcome of photoswitching COL-120 tagged dendra2 in vivo. **d** Representative images of photo-switched areas of LSD1061 COL-120::Dendra of one independent trial. Scale bars = 10 μm for overlay and 1 μm for inset. **e** Quantification of COL-120 turnover of the mid-body area. Each dot represents 1 animal (*n* = 10). See Supplementary Data 5 for statistical details and raw data. **f**–**j** COL-19 is shown as a representative of the pattern II collagen. **f** Schematic representation of COL-19::FRET transgene. (g) Lifespan of wild-type temperature-sensitive sterile background WT (LSD2052 *spe-9(hc88)*; COL-19::FRET) vs long-lived (LSD2053 *glp-1(e2141)*; COL-19::FRET) at 25 °C. Light blue arrows indicate sampling days for FRET imaging.

**h** Representative LSD2052 *spe-9(hc88)*; COL-19::FRET images in the 3D chamber on days 2 and 12 of adulthood at 25 °C of one independent trial. The increasing FRET ratio is color-coded from dark blue to red. Scale bar = 10 μm. **i** FRET ratio increased from day 2 to 12 during the aging of LSD2052 *spe-9(hc88)*; COL-19::FRET at 25 °C. Data are presented as mean $\pm$ SEM. *P* value was determined by an unpaired *t*-test, two-tailed. See Supplementary Data 6 for statistical details and raw data. **j** Long-lived LSD2053 *glp-1(e2141)*; COL-19::FRET showed lower FRET ratio compared to WT (LSD2052 *spe-9(hc88)*; COL-19::FRET) at day 8 of adulthood at 25 °C. Data are presented as mean $\pm$ SEM. *P* value was determined by an unpaired *t*-test, two-tailed. See Supplementary Data 6 for statistical details and raw data. **k**, **l** Type IV collagen EMB-9 is shown as a representative of pattern III collagen. **k** Representative images of photo-switched areas of NK860 EMB-9::Dendra. Scale bar = 10 μm. **l** Quantification of EMB-9::Dendra showed that newly synthesized EMB-9 is laid on top of the old matrix-incorporated EMB-9, which was not turnover. Each dot represents 1 animal (*n* = 14). See Supplementary Data 8 for statistical details and raw data.

(Fig. 2h–j, Supplementary Data 4). This suggests that *glp-1*-mediated longevity intervention counteracts age-dependent crosslinking of COL-19 collagens.

To assess whether the increase in FRET transmission of this COL-19::FRET sensor corresponds to a reduction in its extensibility due to crosslinking, we fixed *C. elegans* with the crosslinking-agent formaldehyde, which increased the FRET transmission compared to anesthetized animals (Supplementary Fig. 4g, h). However, treating *C. elegans* with agents that either increase or decrease AGEs had minor effects on COL-19::FRET ratios and on lifespan (Supplementary Fig. 4i–q, Supplementary Data 7), arguing against collagen crosslinking as the sole driver for the age-dependent increase in FRET transmission.

An alternative reason for age-dependent changes in FRET transmission could be a change in tissue tension. Tissue tension is established and maintained by cells pulling on the ECM, either within the tissue itself or in neighboring tissues. To assess tissue tension, we used sodium chloride to remove the internal osmotic pressure leading to wrinkling of the cuticle at young (day 2) and old (day 8) age. We found that the age-dependent increase of COL-19::FRET transmission was nullified by loss of internal pressure in wild-type and long-lived *glp-1* animals at day 8 of adulthood (Supplementary Fig. 4r–u). However, young animals still had lower FRET ratios after salt treatment, suggesting that not all of it is due to tissue tension, but some part might be due to collagen cross-linking. Our observation is consistent with a recent finding that under osmotic-shock-induced shrinkage of *C. elegans*, longevity interventions prevent the age-dependent increase of cuticular stiffness, which is nullified by knocking down pattern I collagen *col-120*[31], strengthening our model that longevity interventions promote ECM homeostasis also to counteract collagen crosslinking.

We conclude that age-related changes, including collagen cross-linking, occur, and cuticle integrity declines during aging partly due to the loss of cells adhering to the ECM and progressive loss of tissue tension, which is slowed by longevity interventions.

**Reduced Insulin/IGF-1 receptor signaling does not counteract the age-dependent accumulation of basement membrane collagens**

As type IV collagen EMB-9 is a representative member of the pattern III collagens, we examined EMB-9 tagged with Dendra. As before, we photoconverted on day 2 of adulthood and two days later quantified the green (new) to the red (old) ratio of this basement membrane collagen. The newly synthesized EMB-9 collagens were added to the old collagens resulting in a thickening of the basement membrane independent of *daf-2(RNAi)*-induced longevity (Fig. 2k, i, Supplementary Fig. 3k, l), Supplementary Data 4, 8). This observation is consistent with the thickening of human basement membranes (up to 100-fold) during aging[9].

Taken together, we mapped the dynamic ECM composition (*i.e.*, matreotype) during aging. Our data indicate that some ECM components are, once synthesized, incorporated and stay lifelong in the ECM, potentially accumulating damage, whereas other components are excised from the ECM, and yet other ECM components are continuously added to the ECM. This raises the question of which ECM changes are important for longevity.

**Longevity interventions counteract age-dependent ECM compositional changes**

To elicit what constitutes a youthful matreotype or ECM composition upon longevity interventions, we treated wild-type animals with *daf-2(RNAi)* to slow aging and compared the protein levels relative to control using proteomic data acquired at different time points of aging (Fig. 3a, Supplementary Data 9). In the longitudinal abundance data, we identified signatures that might reinstate a youthful matreotype (Fig. 3a, Supplementary Data 9). Consistent with our proteomics, across six different longevity interventions (dietary restriction, metformin, *glp-1*, *daf-2*, *isp-1*, *eat-2*) and datasets[34–38], the same signature of an increase of a subset of cuticular collagens (*col-*) protein levels, collagen-stabilizing and remodeling enzymes (*dpy-18*, *phy-2*, *bli-*, *nas-*, *zmp-*) and a decrease of cathepsin (*cpl-*, *cpz-*, *cpr-*) protease levels (Fig. 3b, Supplementary Data 9). We thus propose that longevity interventions mobilize compensatory adjustments in early adulthood to counteract age-related ECM changes, presumably to maintain homeostasis of the ECM proteins.

Given that the largest observed changes occurred with enzymes that remodel collagens and with collagens, we quantified the overall collagen levels during aging. We found that one-fifth of the total collagen mass normalized to total protein mass was lost during aging (Fig. 3c). The longevity-promoting *glp-1* mutants started with more collagen mass which declined at a similar rate during aging compared to the wild type (Fig. 3d). Adult-specific *daf-2* RNAi is sufficient to promote higher collagen levels at day 8 of adulthood[24], suggesting that longevity interventions slow the loss of collagen mass during aging.

The cuticle is the fifth largest body part of *C. elegans*, making up about one hundred thousand μm³ or 1/6th of the total volume[39]. Because many cuticular collagens decline during aging, we reasoned that there might be a thinning of the cuticle occurring and, thereby, a loss of barrier protection, similar to the age-dependent loss of collagen and thinning of the human skin[8]. However, based on electron microscopy images, the cuticle thickened in total by 0.197 μm (18%) during old age (days 7-15 of adulthood; Fig. 3e, Supplementary Fig. 5), consistent with previous observations[40,41]. It is unclear what underlies the thickening of the cuticle during aging, but given the massive decline in collagen levels, it might be other cuticular components such as the insoluble cuticulins, an accumulation of water, or a loosening of structural integrity as observed by EM[42].

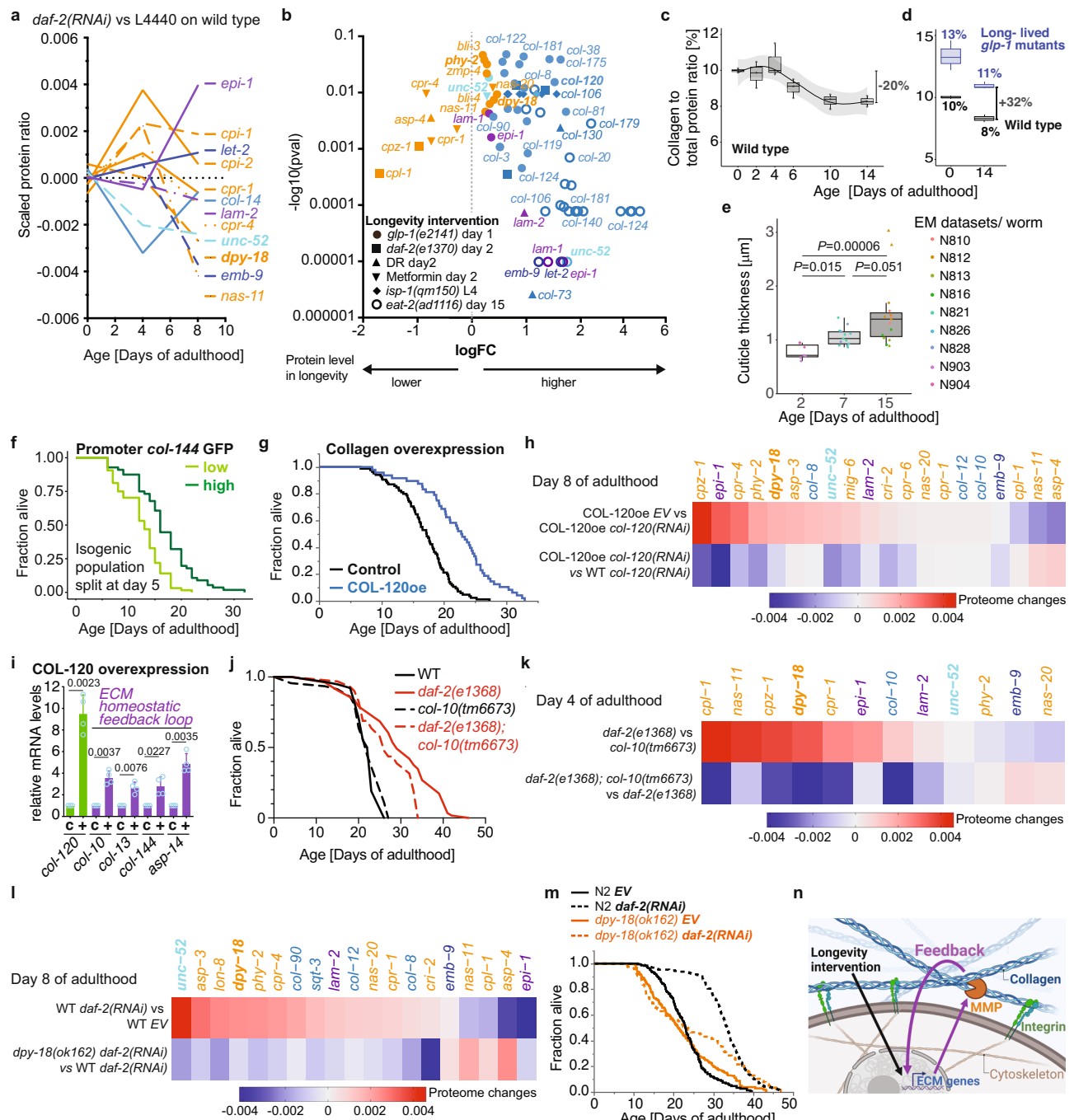

## A feedback loop is sufficient and required for ECM homeostasis and longevity

Our observed time course of the collagen mass changes coincides with the decline of pattern I collagens at the final days of reproduction (days 4-8 of adulthood) and with the growth rates in body size during adulthood, whereby after the final molt from L4 to adult, *C. elegans* continuously grows until day 6-8 of adulthood and then starts to shrink[43-45]. *C. elegans'* growth during early adulthood means an extension of the cuticular exoskeleton. We have previously shown that longevity interventions prolong this adult growth phase compared to wild type[45]. The longer this adulthood growth phase is, the longer-lived is an individual *C. elegans*[44]. To test whether this correlation is associated with prolonged production of the pattern I collagens, we used transgenic animals expressing GFP driven by the collagen *col-144* promoter, whose expression gradually declines during aging[21,46]. Since

functionally interacting collagens temporally cluster together[28], suggesting that they can be used interchangeably, we chose P*col-144*::GFP transgenic animals because they had the brightest expression from all pattern I transgenic lines, which technically simplified the following assay. We split this isogenic population grown in the same environment into high expression and low expression of P*col-144*::GFP transgenic animals at day 5 of adulthood. Animals expressing higher levels of P*col-144*::GFP at day 5 of adulthood lived longer than their genetically identical siblings with lower levels of *col-144*-driven GFP expressed (Fig. 3f, Supplementary Data 7), demonstrating that prolonged *col-144* expression is associated with longevity.

Given that *C. elegans* die from proliferating *E. coli*, its primary food source, we next asked whether longevity intervention-increased cuticular collagens protect against age-associated infection through improved barrier function. As previously reported, overexpressing

**Fig. 3 | A feedback loop of ECM homeostasis implicated in longevity. a** Time course of counterbalancing age-related changes in matrisome protein levels by *daf-2(RNAi)*-longevity intervention. Details in Supplementary Data 9. **b** Composite of proteomics data showing that distinct longevity interventions increased the normally age-related decrease of collagen levels and dampened the normally age-related elevation of extracellular proteases. Note that individual values of log fold changes (FC) from different proteomics datasets shown here as a composite are not comparable but should indicate the directionality of protein abundance change. For individual volcano plots, data, and details, see Supplementary Data 9. **c** The collagen over total protein content is displayed as a time course for a *spe-9 quasi-wild type* population. Boxplot shows the median (black line), 25th/75th percentiles (hinges), and 1.5*IQR (whiskers). 3 independent biological trials. **d** The collagen over total protein ratio is shown for *spe-9* and *glp-1* mutant populations at days 0 and 14 of adulthood. Boxplot shows the median (black line), 25th/75th percentiles (hinges), and 1.5*IQR (whiskers). 3 independent biological trials. **e** Cuticle thickness increases with age based on electron microscopy (EM) images (Source: wormimage.org). Individual *C. elegans* are represented as dots (EM dataset). Triangles indicate outliers. Boxplot shows the median (black line), 25th/75th percentiles (hinges), and 1.5*IQR (whiskers). *P* values are One-way ANOVA post hoc Tuckey. See Supplementary Fig. 5 and Supplementary Data 9 for details. **f** Isogenic population of *col-144* promoter GFP (LSD2002 *spe-9(hc88)*; Pcol-144::GFP) *C. elegans* were split at day 5 of adulthood into high and low expressing GFP individuals. **g** Collagen COL-120oe (LSD2017) overexpression increased lifespan compared to control (wild type with *rol-6(su1006)* co-injection marker LSD2013) on UV-inactivated bacteria. **h** Differences in protein abundance ratios are displayed for COL-120oe and wild-type populations undergoing control and *col-120* RNAi treatment. **i** Overexpression of COL-120oe (LSD2017) increased mRNA levels of other collagens and ECM proteases by qRT-PCR at day 1 of adulthood. N=4 independent biological samples in duplicates (each over 200 L4 worms). Mean ± SEM. *P* values relative to WT were determined by a one-sample *t*-test, two-tailed, with a hypothetical mean of 1. **j** Collagen *col-10(tm6673)* mutation partially suppressed *daf-2(e1368)* reduced insulin/IGF-1 receptor signaling longevity at 20 °C. **k** Changes in protein abundance ratios are shown for *daf-2(e1368)*, *col-10(tm6673)*, and *daf-2(e1368); col-10(tm6673)* animals. **l** The effect of *daf-2* RNAi on changes in protein abundance ratios is shown in wild type and *dpy-18(ok162)* populations. **m** Prolyl 4-hydroxylase *dpy-18(ok162)* mutation partially suppressed reduced insulin/IGF-1 receptor signaling longevity upon adulthood-specific knockdown of *daf-2* at 20 °C. **n** Model of the extracellular matrix homeostasis feedback loop. Created with BioRender.com. **f, g, j, m** For details, raw data, and statistics, see Supplementary Data 7.

COL-120 extends *C. elegans*' lifespan[24], and we show that on dead bacteria COL-120 overexpression in the observer-unbiased lifespan machine still increases lifespan to the full amount (Fig. 3g, Supplementary Data 7), excluding the idea that higher cuticular collagen levels would extend lifespan by preventing bacterial infection through improving barrier function. Furthermore, inducing endogenous *col-120* expression post-development was sufficient to increase lifespan[47]. As previously reported, not all collagens, when overexpressed, increase lifespan[24]. To expand on this, we assessed several more collagen overexpression and their effects on lifespan. Overexpressing collagens that were not altered by longevity interventions were insufficient to increase lifespan (Supplementary Fig. 6, Supplementary Data 7), suggesting that longevity-promoting collagens have unique properties.

To identify downstream mechanisms mobilized by collagen overexpression, we used COL-120 overexpressing (COL-120OE) animals and treated them either with a control empty vector (EV) or *col-120(RNAi)* from L4 and performed proteomic analyses on day 8 of adulthood. We found abundance changes in proteins governing the cytoskeleton dynamics (*tbcb-1*/Tubulin-specific chaperone B, *pat-6*/parvin, *ifd-2*/intermediate filament), as well as proteins involved in pathogen and oxidative stress response, and metabolism, respectively (Supplementary Data 9). Interestingly, we observed changes enhancing ECM composition (*i.e.*, matreotype) that were specific to COL-120 overexpression (Fig. 3h, Supplementary Data 9). These include enhancement of cuticular collagens, enzymes that remodel cuticles and stabilize collagen, basement membrane components, as well as reduction of some age-dependent upregulated proteases (Fig. 3h). We confirmed by qRT-PCR that overexpression of COL-120 leads to upregulation of transcripts coding for other collagens and ECM-remodeling enzymes (Fig. 3i), suggesting a feedback loop between ECM composition and ECM production by cells. This suggests a model in which the abundance of one key collagen is read out to adjust the abundance of other ECM components, an important feature for assembling a functional matrix.

To identify the extent of ECM remodeling and downstream pathways, we compared *daf-2(e1368)* with and without *col-10* at day 4 of adulthood. As the loss of *col-10* blunts the longevity of *daf-2* (Fig. 3j, Supplementary Data 7), we found that the enhancement of metabolism, detoxification, and stress defense depended on *col-10* but also on changes in the cytoskeleton (Supplementary Data 9). For the matrisome changes, the most significantly enhanced ECM components by *daf-2*-induced longevity depending on *col-10* was prolyl 4-hydroxylase/ *dpy-18*, an enzyme important for collagen stabilization (Fig. 3k,

Supplementary Data 9), reinforcing the idea of a feedback loop and demonstrating the importance to remodel the ECM for the cellular reprogramming upon longevity interventions.

To further test this idea, we compared wild type with a P4H *dpy-18(ok162)* loss-of-function mutant treated either with control or *daf-2(RNAi)* from L4 and performed proteomics at day 8 of adulthood (Fig. 3l, Supplementary Data 9). We found that the longevity-matreotype upon *daf-2(RNAi)* was reverted to the aging-matreotype of wild type in the *daf-2(RNAi)*-treated P4H *dpy-18(ok162)* mutants (Fig. 3l). Consistent with these findings P4H function did not affect wild-type lifespan, but the enhancement of P4H during aging was important for longevity. The *daf-2(RNAi)*-mediated longevity was abolished in P4H *dpy-18(ok162)* mutants (Fig. 3m, Supplementary Data 7), demonstrating that collagen stability is required for the dynamic change of ECM composition associated with healthy aging and for lifespan extension.

Therefore, these results show that longevity interventions prolong the dynamic ECM homeostasis during aging. Levels of key ECM components are read out and communicated back into the cell to initiate a response that balances all the multifold components that make up the matrix (Fig. 3n). Interventions increasing key collagen levels are sufficient to drive enhanced remodeling during aging, whereas blocking this collagen enhancement leads to a collapse of this enhancement (Fig. 3n). Furthermore, the enhancement of ECM is sufficient and required for cellular reprogramming upon longevity interventions.

## Screening identifies mechanotransduction genes as regulators for prolonged collagen expression

To investigate the above-described feedback loop and how longevity interventions prolong the maintenance of collagen dynamics, we designed a target screen to assess ECM homeostasis during old age. Since the decline in the pattern I collagen transcription preceded its decline or remodeling out from the cuticle, we visually scored *col-144* promoter-driven GFP levels (P*col-144*::GFP) that progressively declined from day 1 to day 8 of adulthood based on intensity (Fig. 4a, Supplementary Fig. 7a, b, see Supplementary Data 10 for detailed screen description, choice of mutants, and validating results). Gene categories of candidate hits included autophagy, metabolism, molting, pathogen innate immune response, and signaling (Supplementary Fig. 7c, Supplementary Data 10). To our surprise, most hits were in the matrisome and adhesome gene category (Supplementary Fig. 7c, Supplementary Data 10). We mapped these candidates in an anatomical model displaying the four tissue layers: The cuticle (1) is attached

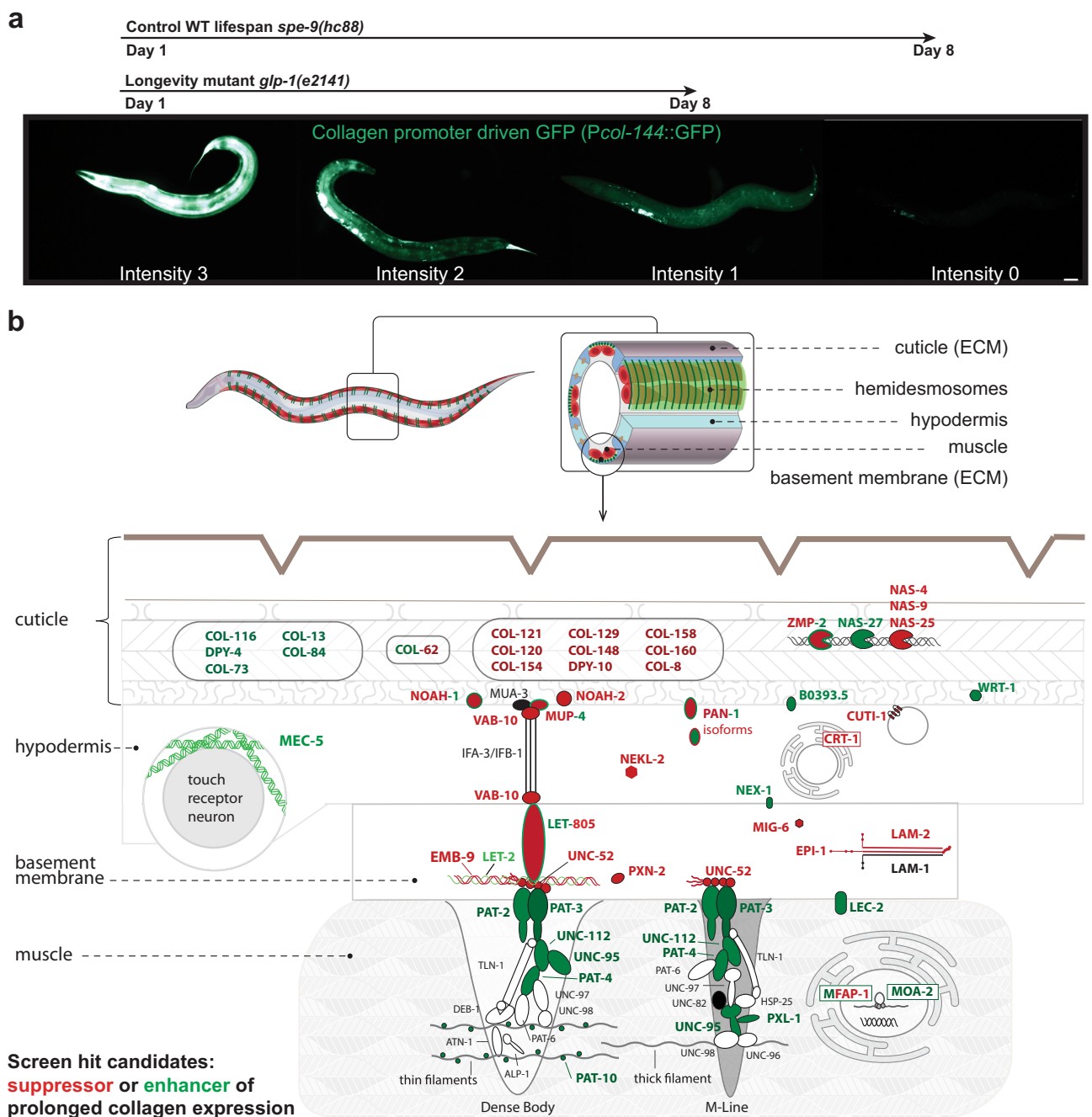

**Fig. 4 | RNAi screen for transcriptional ECM regulators in old age. a** Scoring scheme for the P*col-144*::GFP *C. elegans* RNAi screen. Above, indicated with arrows are the usual decline of the fluorescence of the wild-type background *spe-9*(*hc88*) and long-lived mutant backgrounds *glp-1*(*e2141*). Representative images of one independent biological trial. Scale bar = 50 μm. **b** The hits of the screen are shown in the predicted layers from *C. elegans* cuticle to body wall muscle (not drawn to

scale). In green, RNAi hits led to higher P*col-144*::GFP at day 8 of adulthood. In red, RNAi hits that suppressed normally higher P*col-144*::GFP of long-lived mutants at day 8 of adulthood. The proteins in black were also tested but did not show a significant up- or down-regulation, while the proteins in white were not included in the screen. For details and validation experiments, see Supplementary Data 10.

to the hypodermis (2), which is attached to the basement membrane (3), which is attached to the body wall muscles (4; Fig. 4b). We found that knocking down genes that form or remodel the cuticle either function as an enhancer (green) or suppressor (red) of prolonged collagen expression (Fig. 4b, Supplementary Data 10). RNAi of hemidesmosome genes that anchor the hypodermis to the muscles (*mup-4/ matrilin, vab-10/ dystonin, let-805/ myotactin-fibronectin repeats*) via the basement membrane (*emb-9/collagen type IV, let-2/ collagen type IV, unc-52/ perlecan, epi-1/ laminin alpha, lam-2/ laminin gamma*) were generally required for prolonged hypodermal collagen expression

(Fig. 4b, Supplementary Data 10). By contrast, knocking down genes that form the adhesome and are localized in the muscle (*pat-2/ integrin alpha, pat-3/ integrin beta, unc-112/ FERMT1, pat-4/ integrin-linked kinase, tln-1/ talin, pxl-1/ paxillin, deb-1/ vinculin, pat-10/ troponin C*) upregulated and prolonged P*col-144*::GFP expression in the hypodermis (Fig. 4b, Supplementary Data 10). Thus, components of hemidesmosome-like structures and genes important for hemidesmosome formation or stability are required, whereas proteins localized more proximal surrounding CeHD or involved in downstream signaling in the muscle modulated the prolonged hypodermal

collagen expression. Strikingly, these hemidesmosomes are required for both enabling and transmitting mechanical forces (mechanotransduction) during embryo elongation[48]. This suggests that the underlying mechanism of the feedback loop for prolonged collagen expression during aging might be mediated via mechanical force coupling across tissues.

## Progressive decline in colocalization of basement membrane components with adhesome during aging

For mechanotransduction to occur, both collagen type IV ([EMB-9]$_2$ [LET-2]) and perlecan (UNC-52) need to interact with the integrin receptors composed of the heterodimers of integrin alpha INA-1 or PAT-2 with integrin beta PAT-3 (Fig. 4b)[49]. To monitor the coupling of the basement membrane to integrin signaling during aging, we assessed the colocalization of collagen EMB-9 tagged with mCherry and integrin receptor beta PAT-3 tagged with GFP. Despite the age-dependent increase in EMB-9, UNC-52, and PAT-3 levels in muscular attachment structures, we found a progressive decline in colocalization of EMB-9 with PAT-3 during days 1 to 8 of adulthood, which was rescued by longevity intervention *daf-2(RNAi)* (Fig. 5a–e, Supplementary Fig. 8a, b, Supplementary Data 11).

Because perlecan UNC-52 is at the interface between collagen EMB-9 and integrin receptor PAT-3, we used a temperature-sensitive perlecan *unc-52(e699, su250)* at the semi-permissive temperature of 20°C and found that loss of *unc-52* function accelerated the loss of colocalization of EMB-9 form PAT-3 during day 1 to day 8 of adulthood (Figs. 5a, b), presumably leading to detachment of muscle from the basement membrane. This is consistent with the observation that these *unc-52(e699, su250)* animals become progressively paralyzed in the midbody region at permissive temperature (15°C) during aging[50] but not in the head region, coinciding with the increased colocalization observed in the posterior head region (Fig. 5a).

## Mechanical coupling is required for longevity

To understand the relationship between loss of mechanical tension across tissues and longevity, we used the perlecan *unc-52(e699, su250)* paralysis phenotype as a functional read-out, and hypodermal collagen reporter (P*col-144*::GFP) as a read-out for the loss of mechanical coupling to gene expression across tissues. We noticed that the perlecan *unc-52(e699, su250)* had higher baseline P*col-144*::GFP expression during development and throughout adulthood (Supplementary Fig. 8c, d, Supplementary Data 12), indicating that the strength of mechanical coupling determines gene expression levels across tissues, which is consistent with a feedback loop to adapt exoskeleton cuticle strength with muscle strength.

We found that *daf-2(RNAi)* postponed the perlecan *unc-52(e699, su250)* paralysis phenotype during adulthood by two days at a permissive temperature of 15°C and a semi-permissive temperature of 20°C (Fig. 5f, Supplementary Fig. 8e). This is consistent with a previous study showing a delay of *unc-52*-paralysis in long-lived *glp-1* mutants[51], suggesting that this delay in paralysis is due to the general improvement of protein homeostasis promoted by longevity interventions. By contrast, the prolonged collagen gene expression and extreme longevity upon *daf-2(RNAi)* were blunted and completely abolished by perlecan *unc-52(e699, su250)* mutations at 15°C or 20°C, respectively (Figs. 5f, g, Supplementary Fig. 8e, Supplementary Data 7, 12). This demonstrates that proper tissue coupling is required to promote hypodermal collagen expression for the systemic longevity effects.

## Relevance of the matrisome and adhesome for longevity

To identify which of these components are functionally important for longevity, we measured the lifespan of 35'795 individuals, including 39 matrisome and adhesome mutants treated either with control or *daf-2(RNAi)* (Fig. 6a, Supplementary Data 7) using the lifespan machine[52]. We found requirements for *daf-2*-longevity across all the different

matrisome categories (Supplementary Fig. 9), demonstrating the essential interplay of these molecular components to form a proper functional network. In line with our screening data and proteomics data, components of the hemidesmosome and adjacent ECMs were required for longevity (Fig. 6a), with *unc-52* mutants showing the strongest epigenetic requirements for *daf-2*-longevity (Fig. 6a).

Consistently, hemidesmosome component *vab-10/* plectin loss-of-function mutants were shorter-lived and blocked *daf-2*-longevity (Fig. 6b, Supplementary Data 7). Hemicentin *him-4* is essential for hemidesmosome anchoring and mechanotransduction[53], and *him-4* was required for longevity (Fig. 6c, Supplementary Data 7). Hemidesmosomes interact with integrin receptors, and both integrin alpha (*ina-1*) and the sole integrin beta (*pat-3*) were required for *daf-2*-longevity (Fig. 6d, Supplementary Data 7). Because these hemidesmosome genes, when completely knocked out, are embryonic lethal, and to exclude any developmental effects and to test another longevity pathway, we used long-lived *glp-1(e2141)* animals and knocked down *unc-52* or *pat-3* by RNAi starting at day 2 of adulthood. Adulthood-specific knockdown of *unc-52* or *pat-3* had no lifespan effects on normal-lived control animals but abolished the longevity of *glp-1(e2141)* mutants (Fig. 6e, Supplementary Data 7). Thus, integrin receptors are required for longevity but do not drive normal aging, consistent with the observation that only longevity interventions enhance pattern I collagen expression. This implicates an active functional role of mechanotransductive signaling in this feedback loop to promote longevity.

## Yap-1 is required for longevity and collagen homeostasis

To determine the downstream effectors of this hemidesmosome-dependent feedback loop, we assessed known cytoskeleton remodelers, including talin/TLN-1, integrin-linked kinase (ILK/PAT-4), focal adhesion kinase *kin-32*, and RHO-associated kinase (ROCK/LET-502) (Fig. 5f, Supplementary Fig. 10, Supplementary Data 7), as well as mechano-responsive mediators, such as the conserved Yes-associated protein transcriptional co-activator YAP, which is implicated in the transcriptional response to ECM stiffness and cytoskeletal organization[54,55]. Treatment of *yap-1(tm1416)* putative null mutants with *daf-2(RNAi)* revealed that the loss of *yap-1* function blunted longevity upon reduced Insulin/IGF-1 signaling (Fig. 7a, Supplementary Data 7). We did not observe any enhanced YAP-1 nuclear localization upon *daf-2(RNAi)* but observed higher levels of YAP-1, which was potentiated at 25°C (Figs. 7b, c, Supplementary Data 13). Interestingly, under normal conditions, most YAP-1:GFP was observed in the cytoplasm, and only a very faint YAP-1::GFP signal was detected at hemidesmosome-containing structures (Fig. 7d–h, Supplementary Data 13). Upon *daf-2(RNAi)*, YAP-1 translocated from the cytoplasm to localize at the hemidesmosome-containing structures between the apical and basal VAB-10/plectin/dystonin (Fig. 7d–h, Supplementary Data 13). This might suggest that YAP-1 could help read out mechanical changes occurring at these hemidesmosome-containing structures. In mammals, YAP responds to a broad range of mechanical cues, from shear stress to cell shape and extracellular matrix rigidity, and is considered a mechano-sensitive transcriptional regulator[55–57]. To test YAP-1's mechano-sensitive role in *C. elegans*, we placed L4 animals for 3 days under 12 Pa pressure (Supplementary Data 13). We found that YAP-1 levels increased under these mildly higher pressure conditions (Fig. 7i, Supplementary Data 13), suggesting its expression responded to mechanical compression. If our model of hemidesmosomes regulating collagen expression via mechanical tensions was correct, then placing *C. elegans* under these mild pressure conditions should prolong collagen expression during aging. To control for general gene expression changes under pressure, we generated *C. elegans* expressing mCherry in body wall muscles and neonGreen in the hypodermis under the *col-120* promoter, whose expression rapidly declined in early adulthood (Fig. 7j). Normalized to control muscular mCherry expression, which

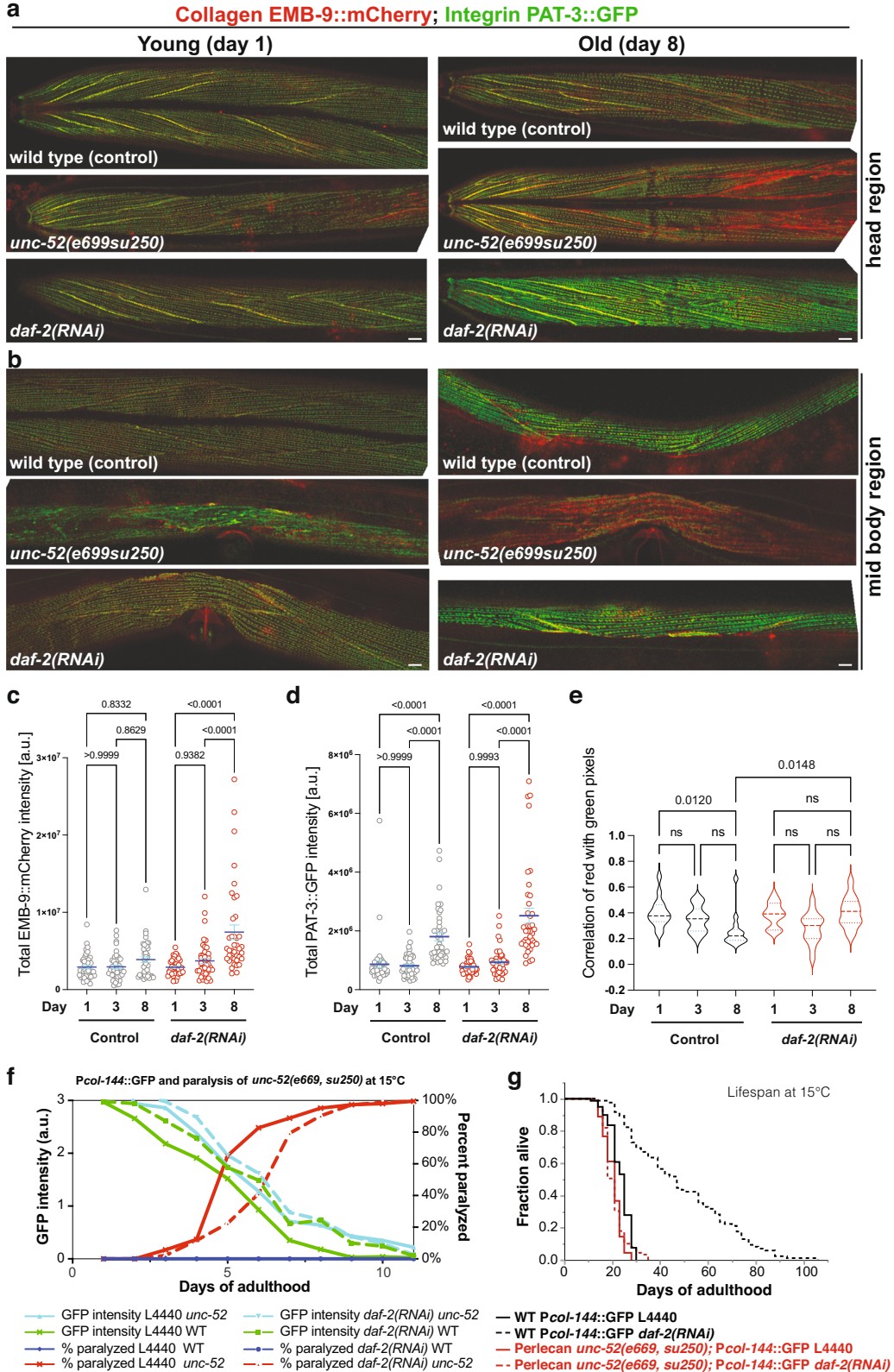

did not change under pressure, we found that *col-120* expression was enhanced by these 3 days of mild pressure (Fig. 7k, Supplementary Data 14). We confirmed this enhanced collagen expression under pressure by *col-144* promoter-driven GFP in a sterile background avoiding FUdR (Fig. 7l, Supplementary Data 14). Weakening the hemidesmosome force transduction by *unc-52* temperature-sensitive mutations, as before, showed already higher *col-144* expression under

normal conditions but blunted the enhancement of collagen expression upon pressure (Fig. 7m, Supplementary Data 14). Furthermore, the knockdown of *yap-1* abolished this pressure-mediated collagen expression (Fig. 7n, Supplementary Data 14), suggesting that YAP-1 requirements for longevity are due to responding to mechanical tension changes from hemidesmosomes to coordinate collagen expression (Fig. 7o).

**Fig. 5 | Disassociation of co-localization of collagen type IV from integrin β receptor. a**, **b** Confocal image overlays of collagen type IV (EMB-9::mCherry) in red and integrin **β** receptor (PAT-3::GFP) in the green of the head region (**a**) and mid-body region (**b**) at day 1 and day 8 of adulthood. The yellow color indicates colocalization. Anterior to the left, ventral side down. Scale bar = 10 μm.
**c**, **d** Quantification of total EMB-9::mCherry (**c**) and PAT-3::GFP (**d**) intensity levels. Individual dots represent animals. Mean ± SEM. One-way ANOVA for the *P* value. 1 independent biological trial is shown. Raw data and statistics are in Supplementary Data 11. **e** Quantification of colocalization by correlation of per-pixel red (EMB-9::mCherry) and green (PAT-3::GFP) intensities of the midbody region (**b**), which declined during aging in wild type but was maintained upon *daf-2(RNAi)* longevity intervention. One-way ANOVA for the *P* value. **a**–**e** For individual pictures, raw data, and statistical analysis, see Supplementary Data 11. **f** Longevity intervention *daf-2(RNAi)* prolonged collagen expression (P*col-144*::GFP) of wild type (green line) during aging but not in a perlecan *unc-52(e669, su250)* mutant background (aquamarine line) at permissive temperature 15℃. The age-dependent *unc-52(e669, su250)* mutant paralysis phenotype (red line) was delayed by *daf-2(RNAi)* (dashed red line) at 15℃. For details, see Supplementary Data 12. **g** Continuing with these same animals, *unc-52(e669, su250)* mutants completely suppressed *daf-2(RNAi)* longevity at 15℃. For details, see Supplementary Data 7.

## Discussion

There is a general appreciation of the importance of the progressive decline of ECM integrity during aging, yet, monitoring ECM integrity in vivo and mechanisms that promote healthy aging through improved ECM maintenance are challenging and less explored. Although there are inherent technical limitations with several approaches, by combining the establishment of matreotypes using proteomics and in-vivo reporter systems, we identify at least three distinct dynamic ECM composition changes during aging. Similar to mammalian aging, the synthesis and turnover of the pattern I collagens are lost[8] (*e.g.*, consistent with the dispensable soma theory of aging), whereas pattern II collagens are not turned over and remain lifelong in the ECM, becoming crosslinked[5] (*e.g.*, accumulation of damage theory of aging), and pattern III collagens are continuously synthesized and thicken the ECM[9] (*e.g.*, hypergrowth theory of aging). All these changes lead to a stiffer but, at the same time, mechanically weaker ECM in old age[5].

We show that longevity interventions counteract the accumulation of collagen crosslinking and prevent ECM-cell mechanical detachment. We identify a regulatory feedback loop reading out the ECM integrity via mechano-sensitive hemidesmosomes to signal into the cells via YAP-1 to adjust ECM components to improve the mechanical properties of the surrounding matrices (Fig. 7o; Supplementary Discussion). In human skin fragility disorders, dysfunctional hemidesmosomes lead to the activation of YAP1[58]; although a direct interaction is unknown, it suggests the conservation of the physiological relevance of a yet unexplored mechanism of hemidesmosome integrity, YAP1, and ECM homeostasis. We further show that *daf-2* longevity not only requires YAP-1 for lifespan extension but also enhances YAP-1 to localize to the hemidesmosomes, perhaps to sensibilize mechanical readout from hemidesmosomes. Although YAP-1 is a mechanism that contributes to organismal lifespan extension through mechanotransduction, other pathways are likely to intersect as well.

Interestingly, in human skin aging, hemidesmosomal integrity is a readout and essential for stem cell maintenance[59,60]. Although there is a vast body of evidence that implicates age-related dysfunctional ECM activating YAP1 in vitro and ex vivo[61], we demonstrate in vivo and non-invasively that altering physical pressure promotes *yap-1*-dependent collagen expression for adaptation. Subsequently, forcing collagen overexpression reprograms metabolism, stress defense, and cellular homeostasis, promoting longevity. This might explain the observation that certain collagens are overexpressed in centenarians based on proteomic signatures[62]. Furthermore, our experimental findings with *C. elegans* fit a mechanobiological regulation model (Fig. 7o) observed in mammalian arterial walls, which includes the interactions of smooth muscles, endothelial cells, fibroblasts, and ECM remodeling[63] (Supplementary Discussion, Supplementary Fig. 11).

Thus, we provide the causal evidence for mechanotransduction functionally coordinating organismal lifespan extension, which adds to the body of evidence that ECM integrity might classify as a hallmark of aging and directs the field towards further investigations to target ECM homeostasis for healthy aging.

## Methods

### *C. elegans* strains

*Caenorhabditis elegans* strains were grown on NGM plates with OP50 *Escherichia coli* bacteria at 20℃. The Bristol N2 was used as a wild-type *C. elegans* strain. Most strains were obtained from the Caenorhabditis Genetics Center [CGC]: BC184 *dpy-14(e188) unc-13(e51) bli-4(s90)/unc-15(e73)* I, BC10074 *dpy-5(e907)* I; sEx10074[Pemb-9::GFP + pCeh361], BC11902 *dpy-5(e907)* I; sEx11902 [Pcol-129::GFP + pCeh361], BC12229 *dpy-5(e907)* I; sEx10002 [Pcutl-23::GFP + pCeh361], BC12275 *dpy-5(e907)* I; sEx12275 [Pcut-6::GFP::GFP + pCeh361], BC12533 *dpy-5(e907)* I; sEx12533 [Pcol-89::GFP + pCeh361], BC12900 *dpy-5(e907)* I; sIs11600 [Pmec-5::GFP+ pCeh361], BC13149 *dpy-5(e907)* I; sEx13149 [Phim-4::GFP + pCeh361], BC13560 *dpy-5(e907)* I; sIs13559 [Pcol-59::GFP + pCeh361], BC13623 *dpy-5(e907)* I; sEx13623 [Pcri-2::GFP + pCeh361], BC13861 *dpy-5(e907)* I; sIs13252 [Plet-2::GFP + pCeh361], BC14295 *dpy-5(e907)* I; sEx14295 [Pgpn-1::GFP+ pCeh361], CB937 *bli-4(e937)* I, OD761 *let-502(sb118)* I, MT3100 *tln-1(n1338)* I, CB698 *vab-10(e698)* I, VC117 *vab-10(gk45)* I, GOU2043 *vab-10a[cas602[vab-10a*::gfp]] I, RB776 *kin-32(ok166)* I, VC855 *cle-1(gk364)* I, CZ5847 *spon-1(ju402)* II; juEx1111, HE250 *unc-52(e669su250)* II, BT24 *rhIs23*[GFP::HIM-4] III, CB1372 *daf-7(e1372)* III, GG37 *emb-9(g34)* III, JK2729 *dpy-18(ok162)* III, RB1574 *cut-6(ok1919)* III, RW1522 *pat-2(st538) unc-32(e189)* III; *stEx10* [*pat-2(+); rol-6(su1006)*], RW3550 *pat-4(st551)/unc-45(e286)* III, NJ268 *pat-3(rh96)* III, NG144 *ina-1(gm144)* III, HE250 *unc-52(e669, su250)* III, CH1445 *unc-119(ed3)* III; *cgEx198* [BLI-1::GFP + *unc-119(+)*], HS428 *dpy-22(os26)* X; *osEx89* [COL-10::GFP + *dpy-22(+)*], MH2051 *kuIs55* [LON-3::GFP + *unc-119(+)*]; pYSL3G3, NG2517 *him-5(e1490)* V; [INA-1::GFP + *rol-6(su1006)*], NK248 *unc-119(ed4)* III; *qyIs10* [LAM-1::GFP + *unc-119(+)*] IV, NK2583 *unc-52(qy80* [NeonGreen::UNC-52]), NK358 *unc-119(ed4)* III; *qyIs43* [PAT-3::GFP + INA-1(genomic) + *unc-119(+)*], NK364 *unc-119(ed4)* III; *qyIs46* [EMB-9::mCherry + *unc-119(+)*], NK651 *unc-119(ed4)* III; *qyIs108* [LAM-1::Dendra + *unc-119(+)*], NK696 *unc-119(ed4)* III; *qyIs127* [LAM-1::mCherry + *unc-119(+)*], NK860 *unc-119(ed4)* III; *qyIs161* [EMB-9::Dendra + *unc-119(+)*], ML2501 *let-805(mc73[let-805*::gfp + *unc-119(+)])* *unc-119(ed3)* III, NK2446 *qy41* [*lam-2*::mKate2] X, NK2479 *qy49[pat-2*::2xmNG] III, RP247 *trIs30* [Phim-4::MB::YFP + Phmr-1b::DsRed2 + Punc-129nsp::DsRed2], TP12 *kaIs12* [COL-19::GFP], WS3403 *opIs170* [SDN-1::GFP::*unc-54* 3'UTR + *lin-15(+)*], WT30 *unc-119(ed3)* III;*wtEx30* [CUTI-1::GFP + *unc-119(+)*], TJ1060 *spe-9(hc88)* I; *rrf-3(b26)* II, CB4037 *glp-1(e2141)* III, LD1036 *daf-2(e1370); him-8(e1489)*, DR411 *dpy-13(e184)/daf-15(m81)* IV, NF773 *fbl-1(k201)* IV, NG57 *epi-1(gm57)* IV, NX3 *col-121* IV, RB1165 *col-99(ok1204)* IV, TM8818 *col-101(tm8818)* IV, VC40161 *col-120* IV, CB5664 *dpy-31(e2770)* III; *sqt-3(e2809)* V, CB6335 *dpy-31(e2919)* III;*sqt-3(e2906)* V, NF198 *mig-17(k174)* V, TM6673 *col-10(tm6673)* V, VC718 *cri-2(gk314)* V, VC40556 *col-12* V, BT12 *him-4(rh319)* X, CB1503 *mec-5(e1503)* X, CB2199 *lon-2(e678); daf-6(e1377)* X, CB3275 *lon-2(e678) mec-5(e1504)* X, CH1878 *dgn-2(ok209) dgn-3(tm1092) dgn-1(cg121)* X; *cgEx308* [pJK600/dgn-1(+) + pJK602/dng-1p::GFP + rol-6(su1066)], EG199 *nas-37(ox199)* X, GG37 *let-2(g37)* X, RB970 *ddr-1(ok874)* X, RB2035 *asp-4(ok2693)* X, RB568 *svh-5(ok286)* X, RT3574 *lin-15B&lin-15A(n765)* X; *ihIs35* [YAP-1::GFP::*unc-54* 3'UTR + *lin-15(+)*], SP2163 *sym-1(mn601)* X, VC233 *gpn-1(ok377)* X, VC1645 *mltn-13(gk766)* X, VC1697 *mltn-13(gk807)* X, *yap-1(tm1416)* X.

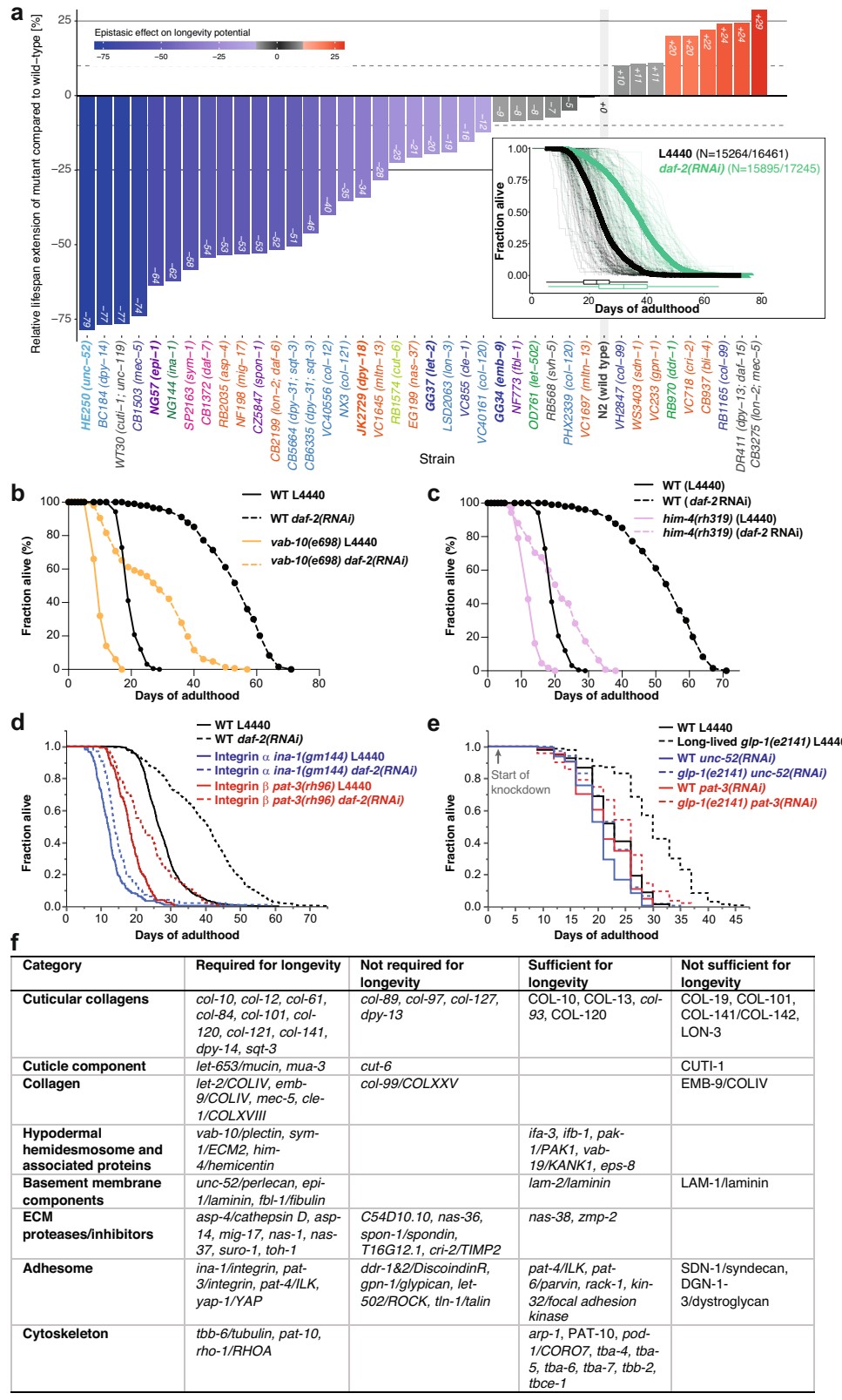

**Fig. 6 | Functional matrisome is of key importance for *C. elegans* longevity.**
**a** Epistatic effect of matrisome mutations on rIIS-mediated longevity potential. Bars represent the relative differences in mean lifespan extension (%) measured in matrisome mutants compared to wild-type animals within the same batch.
**b** Mutation in *vab-10*/plectin/dystonin shortened wild-type (WT) lifespan and blocked longevity upon reduced insulin/IGF-1 receptor signaling at 20°C.
**c** Mutation in *him-4*/hemicentin shortened wild-type (WT) lifespan and blocked longevity upon reduced insulin/IGF-1 receptor signaling at 20°C. **d** Mutations in integrin α and β suppressed *daf-2(RNAi)* longevity at 20°C. **e** Knocking down perlecan/*unc-52* or integrin β/*pat-3* starting at day 2 of adulthood suppressed germ cell-less (*glp-1(e2141)*) mediated longevity. **f** Summary of all matrisome and adhesome genes implicated in longevity. RNAi or genetic mutants are in italics, overexpression is in capital letters. **a**–**f** For details, statistics, and additional trials, see Supplementary Data 7.

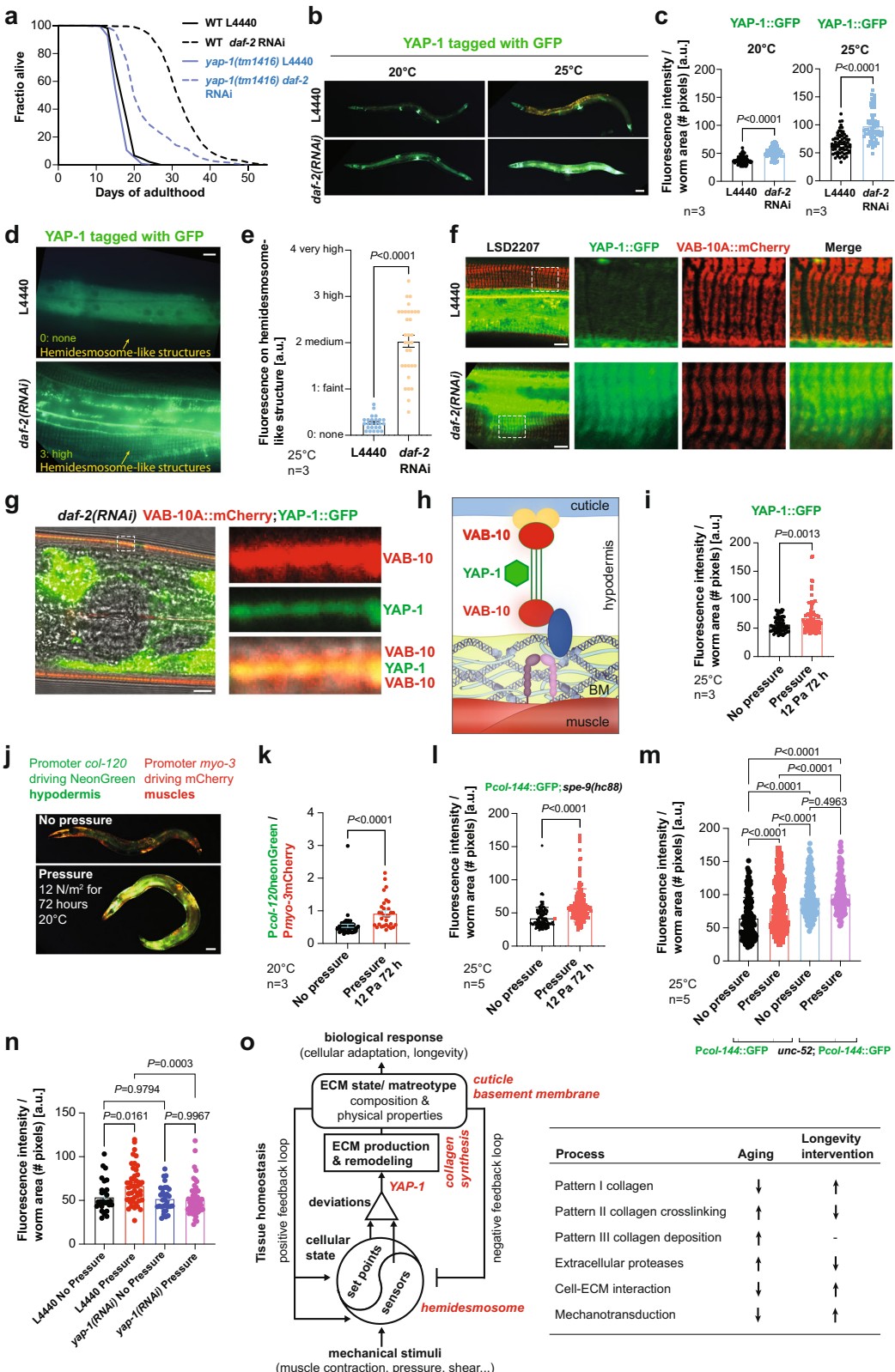

The strains AH3284 *pat-3(zh105[PAT-3(Y792F)])*, AH3437 *zh117* [GFP::TLN-1], AH4617 *zh115* [PAT-3::GFP] III, DMS1020 *dmals40* [*col-101*p::*col-101*::GFP (40 ng/µl); *unc-54*p::mCherry (40 ng/µl)], OD761 *let-502(sb118)*, ML2600 *vab-10(mc100*[VAB-10A::mCherry+loxp])I, CS678 *col-141(lf)*, CS637 Ex [COL-141COL-142(oe); Pmyo-2::GFP], *jgls5* [ROL-6::GFP;TTX-3::GFP], TU1 *unc-119(ed3)*III; [P*col-99*[16655]::S0001_

pR6K_Amp_2xTY1ce_EGFP_FRT_rpsl_neo_FRT_3xFlag] dFRT::unc-119, LSD1001 [*pha-1(e2123)*; *col-19*(FRET between exon)-version C3 + PHA-1(+)], VH2847 *hdls73* [COL-99::GFP, pha-1(+)] were gifts from other labs.

The last group of strain was generated by UV integration, crossing, or injection in our lab: LSD1000 *xchEx001* [(pRedFlp-Hgr)(*col-*

**Fig. 7 | YAP-1 is required for longevity and pressure-induced collagen expression during aging. a** Loss-of-function mutants of *yap-1(tm1416)* blunted longevity upon reduced Insulin/IGF-1 signaling. For raw data, additional trials, and statistics, see Supplementary Data 7. **b, c** YAP-1::GFP expression was increased upon reducing insulin/IGF-1 signaling. **b** Representative images show increased expression of *ihIs35* YAP-1::GFP in animals fed with *daf-2* RNAi as compared to L4440 control at 20°C and 25°C. Scale bar = 50 μm. **c** The graph shows the quantified data for the YAP-1::GFP expression. Individual dots represent the number of animals. Data are represented as columns with mean ± SEM. *n* = 3 independent biological trials. Welch's *t*-test, two-tailed was used for significance analysis for the *P* value. For raw data and statistical details, see Supplementary Data 13. **d, e** YAP-1::GFP expression localizes at hemidesmosome-containing structures was increased upon reducing insulin/IGF-1 signaling at 25°C. **d** Representative images show increased expression of *ihIs35* YAP-1::GFP in animals fed with *daf-2* RNAi as compared to L4440 control at 25°C. Scale bar = 10 μm. **e** The graph shows the distribution of the categories of hemidesmosome-containing structures. Individual dots represent the number of animals. Data are represented as columns with mean ± SEM. *n* = 3 independent biological trials. Welch's *t*-test, two-tailed was used for significance analysis for the *P* value. For raw data and statistical details, see Supplementary Data 13. **f, g** Upon *daf-2(RNAi)* at 25°C, more YAP-1 was colocalized with VAB-10. The boxed areas are enlarged on the right showing the *ihIs35* YAP-1::GFP (green channel), *mc100* VAB-10A::mCherry (red channel), and merge of the confocal images using the transgenic strain LSD2207. Scale bar = 10 μm. Representative images of one independent biological trial. For additional images and individuals, see Supplementary Data 13. **h** Schematic of YAP-1 localizing at the hemidesmosome-containing structure between the apical and basal VAB-10 based on confocal images shown in (**g**). Not drawn to scale. **i** YAP-1::GFP expression was increased upon constant pressure for three days at 25°C. Individual dots represent the number of animals. Data are represented as columns with mean ± SEM. *n* = 3 independent biological trials. Welch's *t*-test, two-tailed was used for significance analysis for the *P* value. For raw data and statistical details, see Supplementary Data 13. **j, k** Pressure induces collagen expression in the hypodermis. **j** Representative images of LSD1126 P*col-120*::mNeonGreen; P*myo-3*::mCherry transgenic *C. elegans* either kept under 12 Pa pressure for 72 h starting at L4 at 20°C or not. Scale bar = 50 μm. **k** Quantification of 3 independent biological trials (n=3). An unpaired *t*-test, two-tailed was used for significance analysis for the *P* value. For raw data and statistical details, see Supplementary Data 14. (**l**) Promoter-driven *col-144*::GFP expression in a temperature-sterile background (LSD2002) was increased upon constant 12 Pa pressure for three days at 25°C. Individual dots represent the number of animals. Data are represented as columns with mean ± SEM. *n* = 5 independent biological trials. Welch's t-test, two-tailed was used for significance analysis for the *P* value. For raw data, additional trials also at 20°C, and statistical details, see Supplementary Data 14. (**m**) Mutations in *unc-52(e669,su250)* blunted promoter-driven *col-144*::GFP expression upon constant 12 Pa pressure for three days at 25°C. Individual dots represent the number of animals. Data are represented as columns with mean ± SEM. *n* = 5 independent biological trials. One-way ANOVA was used for significance analysis for the *P* value. For data and statistical details, see Supplementary Data 14. **n** Knockdown of *yap-1* abolished the promoter-driven *col-144*::GFP expression upon constant 12 Pa pressure for three days at 25°C. 1 independent biological trial is shown. One-way ANOVA was used for significance analysis. For data, additional trials, and statistical details, see Supplementary Data 14. **o** Proposed biomechanical model of ECM homeostasis and longevity. The right model is adapted for our *C. elegans* finding from the mechanobiological regulation of arterial walls, which include smooth muscles, endothelial cells, fibroblasts, and ECM (by Humphrey and Schwartz 2021)[63]. The left model depicts the ECM homeostasis-related process that declines during aging but is counteracted by longevity interventions.

*120*[30044]::S0001_pR6K_Amp_2xTY1ce_EGFP_FRT_rpsl_neo_FRT_3x-Flag]dFRT::unc-119-Nat]; pRF4 [*rol-6(su1006)*], LSD1106 *pha-1(e2123)* III; *xchEx105* [P*col-120*::NeonGreen; *pha-1*(+)], LSD1107 xchEx017 [P*col-120*::NeonGreen; P*col-12*::DsRed], LSD2001 *xchIs001* [P*col-144*:: GFP; *pha-1*(+)], LSD2043 *xchIs012* [(pRedFlp-Hgr) (*col-120* [30044]::S0001 pR6K Amp 2xTY1ce EGFP FRT rpsl neo FRT 3xFlag) dFRT::unc-119-Nat]; pRF4 *rol-6(su1006gf)*, LSD2051 [*col-19*(FRET between exon)-version C3 + PHA-1(+)] was made by UV integration of LSD1001 and outcrossing 12 times, LSD2052 *spe-9(hc88)*; [*col-19*(FRET between exon)-version C3 + PHA-1(+)], LSD2052 *pha-1(e2123)*; [col-19(FRET between exon)-version C3]; *spe-9(hc88)* I, LSD2053 *glp-1(e2141)*; [*col-19*(FRET between exon)-version C3 + PHA-1(+)], LSD2063 *kuIs55* [LON-3::GFP + *unc-119*(+) pYSL3G3 rollers]; *spe-9(hc88)* I, LSD2117 *xchIs016* [P*col-19*::GFP], LSD2002 *spe-9(hc88)* I ; *xchIs001* [P*col-144*:: GFP; *pha-1*(+)], LSD2003 *glp-1(e2141)* III ; *xchIs001* [P*col-144*:: GFP; *pha-1*(+)], LSD2117 *xchIs016* [P*col-19*::GFP], LSD2122 *spe-9(hc88)* I; *daf-2(e1370)* III; *xchIs001* [P*col-144*:: GFP; *pha-1*(+)], LSD1061 *xchEx062* [P*col-120*::col-120::dendra2; pha-1 (+)]; pha-1(e2123)III, LSD2191 *xchIs001* [P*col-144*:: GFP; *pha-1*(+)] X; *unc-52(e669su250)*, LSD2197 *xchIs001* [P*col-144*:: GFP; *pha-1*(+)] X; *pat-3(rh96) III*, LSD2147 *qyIs46* [P*emb-9*::emb-9::mCherry + *unc-119*(+)] ; *pat-3(zh115* [PAT-3::GFP]); unc-119(ed4) III, LSD2161 *qyIs46* [P*emb-9*::emb-9::mCherry + *unc-119*(+)]; *unc-52(e669su250) II* ; *pat-3(zh115* [PAT-3::GFP]); *unc-119(ed4)* III, LSD2207 *vab-10(mc100*[VAB-10A::mCherry+loxp])I; *ihIs35* [YAP-1::GFP::*unc-54* 3'UTR + *lin-15*(+)].

### Cloning of transgenic constructs
The plasmid pXCH8 (P*col-120*::NeonGreen) was built and purchased from Vectorbuilder. The promoter sequence originates from *col-120* fosmid clone WRM0622A_D10(pRedFlp-Hgr)(col-120[30044]::S0001_pR6K_Amp_2xTY1ce_EGFP_FRT_rpsl_neo_FRT_3xFlag)dFRT::unc-119-Nat from the TransgeneOme Project, while the plasmid DG398 (Slot2 ENTRY vector for mNeonGreen::3xFlag) served as backbone.

### Generation of mutant *C. elegans* strains
PHX2339 *col-120(syb2339)* was generated by SunyBiotech, and is a 1085 deletion of the Y11D7A.11 covering the ATG to the TAA stop and was 4 times outcrossed.

### Generation of transgenic lines
LSD1106 *pha-1(e2123)* III; *xchEx105* [P*col-120*::NeonGreen; *pha-1*(+)]. Generated by injecting 50 ng/μl pXCH8 (P*col-120*::NeonGreen) with 50 ng/μl pBX (*pha-1* (+)) co-injection marker into *pha-1(e2123)*III mutants. LSD1107 *xchEx017* [P*col-120*::NeonGreen; P*col-12*::DsRed] was generated by injecting 50 ng/μl pXCH8 (P*col-120*::NeonGreen) with 50 ng/μl P*col-12*::DsRed into N2.

LSD2001 *xchIs001* [P*col-144*:: GFP; *pha-1*(+)] was generated by integration of Ex [P*col-144*:: GFP; pha-1(+)] via UV light irradiation and 8 times outcrossing with N2 animals.

The strain LSD2043 *xchIs012* [(pRedFlp-Hgr) (col-120 [30044]::S0001 pR6K Amp 2xTY1ce EGFP FRT rpsl neo FRT 3xFlag) dFRT::unc-119-Nat]; pRF4 rol-6(su1006gf) was generated by microinjecting 1 ng/μl of WRM0622A_D10 fosmid (https://transgeneome.mpi-cbg.de/transgeneomics/index.html) together with 50 ng/μl pRF4 *rol-6(su1006gf)* into N2. The *xchEx001* strain was then stably integrated into the genome via UV light irradiation and outcrossed 8 times.

We obtained the strain LSD2117 *xchIs016* [P*col-19*::GFP] by injecting 50 ng/μl pJA1 [P*col-19*::GFP] (gift by Ann Rougvie) into N2, followed by UV irradiation and 8 times outcrossing. We UV-integrated collagen overexpressing strains: LSD2013 *xchIs005* [pRF4 *rol-6(su1006gf)* (100 ng/μL]] (=co-injection marker control), LSD2014 *xchIs006* [P*col-13*::COL-13genomic (50 ng/μL); pRF4 *rol-6(su1006gf)* (100 ng/μL)], LSD2017 *xchIs009* [P*col-120*::COL-120genomic (50 ng/μL); pRF4 *rol-6(su1006gf)* (100 ng/μL)],

LSD2018 *xchIs010* [P*col-10*::COL-10genomic (50 ng/μL); pRF4 *rol-6(su1006gf)* (100 ng/μL)], LSD1061 was generated by injecting 50 ng/μl pXCH4 (P*col-120*::col-120::dendra2) together with 50 ng/μl pBX (*pha-1* (+)) into LSD9 (*pha-1 (e2123)*III). For selection and maintenance of transgenic animals, *the C. elegans* were placed at 25°C.

### Imaging of matrisome and adhesome
The genes comprising the *C. elegans* matrisome and Adhesome are curated in Supplementary Data 1.

Unless indicated, animals were kept on NGM plates. For the images, the developmental stages from egg to larval L4 were selected from a mixed plate under the stereoscope and immediately imaged. In

preparation for the imaging of day 1 and day 8 of adulthood animals, L4 animals were transferred from NGM plates on plates containing 50 μM FUdR and imaged when they reached their respective age. Depending on whether the fluorescent protein (FP) tag hindered proper secretion and incorporation of the core-matrisome protein, some portion of the FP-tagged ECM protein became stuck in the endoplasmic reticulum (ER). In these cases, we largely ignored cytosolic/ER FP signals and focused on FP surrounding plasma membranes that are incorporated into ECM. The fluorescence of the animals was graded on a scale from 0 to 3 intensity. Intensity 3 indicates the highest fluorescence observed. Relative to the highest observed fluorescence of a given reporter line, a gradient scale in 0.5 intervals were categorized and scored, with 0 indicating no fluorescence above the background.

For imaging, we used the BX-51-F Tritech™ Research bright-field fluorescence microscope with a DFK 23UX236 camera, IC Capture 2.4 software, and a triple-band filter from Chroma Technology Corp (described in[64]). We used 2 mM Levamisole hydrochloride dissolved in the M9 buffer to immobilize the animals for imaging.

### Analysis of collagen-tagged GFP fluorescence intensity

For the analysis of our collagen::GFP strains, we used a Python script written by Elisabeth Jongsma and Jeliazko Jeliazkov in ImageJ[21]. The code is designed to measure the GFP intensity in *C. elegans* animals while ignoring the gut autofluorescence. The program takes the area of interest selected from the digital image and compares the intensities for the green and red channels within each pixel. *C. elegans* autofluorescence appears as yellow in the images, a blend of red and green[64]. To remove the autofluorescence without affecting the GFP signal, the red channel intensities are subtracted from the green. Furthermore, signals below a certain intensity threshold are regarded as background noise and ignored. The program then counts all remaining pixels with intensities in the green channel and adds up the total intensity (it also gives the number of pixels and the mean intensity per pixel). The resulting image can be printed to check if the thresholds were placed properly. The data was further analyzed and visualized using GraphPad Prism 8.2.0. *P*-values were calculated using a two-way ANOVA.

### In-silico expression and proteomics analysis

Published datasets were obtained directly from the corresponding supplementary material or through the sequence read archive (SRA). RNA-sequencing datasets were subjected to quality control, quantification[65], and subsequent linear modeling[66]. Data cleaning and analysis were performed in R (dplyr, ggplot2, clusterProfiler) and using the *C. elegans* Matrisome Annotator (http://ce-matrisome-annotator.permalink.cc/)[25].

### COL-19::FRET imaging

*C. elegans* were anesthetized with 25 mM sodium azide (Sigma, S2002-100G) in M9 for live imaging. They were imaged on a coverslip (Menzel Gläser, 24x60mm) covered with a smaller coverslip, which was stuck together with either nail polish (2D or compressed) or with double-sided tape (Sury AG, 3M/9473M25) (3D flow chamber for aging experiments).

*For the formaldehyde crosslinking experiment:* LSD1001 on day 1 of adulthood was fixed in 4% formaldehyde (Sigma-Aldrich, 158127) and imaged the next day. 4% formaldehyde was dissolved in PBS by heating up to 60°C for approximately 2 hours. The formaldehyde solution was sterile filtered and stored at -20°C. We imaged the cuticle as a planar structure in 2D. Therefore 16 μl sodium azide solution was added to a coverslip, and *C. elegans* were subsequently added to this drop. Next, the drop was covered with a smaller coverslip containing nail polish on each corner for attachment. *C. elegans* prepared with nail polish were

imaged at 2048 x 2048 pixel resolution, laser power 20%, zoom 1.5, 400Hz, line accumulation 3.

*For the aging experiments:* We prepared a flow chamber to image *C. elegans* without exerting external mechanical forces (flow chamber, 3D). Therefore 16 μl sodium azide solution was added to a coverslip, and C. elegans were subsequently added to this drop. Next, the drop was covered with a smaller coverslip containing stripes of double-sided sticky tape (50 μm thickness (3M-VHB; Sury AG, S1473-M25)) on two sides for attachment. *C. elegans* prepared in a flow chamber were imaged at 1024 x 1024 pixel resolution, laser power 25%, zoom 3, 700Hz, line accumulation 4, and z-step size 0.13 μm.

*For chemical manipulation of the FRET ratios during aging:* LSD2052 *pha-1(e2123)*; COL-19(FRET between exon)-version C3; *spe-9(hc88)* animals were bleached, and eggs were distributed on NGM plates to grow at 25°C. At day 4 of adulthood, they were transferred on NGM plates containing 2% ribose (Sigma, R7500-100G) and nystatin (2.5 ml/1l NGM) (ThermoFisher, 11548886), or 1 mM genipin (Sigma, G4796-25MG), 10 mM MGO (Sigma, 67028-100ML) or 100 mM aminoguanidine (Sigma, 396494-25G). Also, some *C. elegans* were transferred on NGM plates as a control group. After day 4 of adulthood, all *C. elegans* were grown at 20°C and were imaged on day 7 of adulthood. For another approach, LSD2052 animals were placed on day 1 of adulthood on NGM plates containing ribose and nystatin or 1 mM genipin at 25°C. These samples were put at 20°C on day 4 of adulthood and imaged on day 7 of adulthood too.

Imaging was performed with a Leica TCS SP5 confocal microscope (Leica, Microsystems, Mannheim Germany). Cerulean was excited with the 458 nm laser, and emission was detected in the range 470-515 nm (donor channel) and 520-600 nm (acceptor channel). Images were taken with a 63 x PL APO CS 1.4 oil objective (pinhole diameter of 95.5 μm). A maximum intensity projection was performed on z-stacks with ImageJ before quantitative analysis. FRET ratio images were constructed in ImageJ and further visualized and analyzed with LAS AF Lite (Leica Microsystems) and Inkscape. The ImageJ macro for FRET ratio calculation is provided in the last tab of Supplementary Data 12 (FRETanalysisMACRO).

### Imaging and photoconversion of COL-120::Dendra2 and EMB-9::Dendra2 in *C. elegans*

For the RNAi experiments, L4 animals were placed on respective RNAi plates for one generation. To age synchronize the animals, only L4 C. *elegans* of the F1 generation were selected and moved to fresh RNAi plates containing 50 μM FUdR. The animals were imaged on day 2 and day 4 of adulthood, with the photoconversion performed only on the second day of adulthood. For imaging, they were transferred into a drop of M9 onto 2mm thick, 3% agar pads on microscope slides. For the confocal images, an Olympus FluoView 3000 microscope was used. On day 2, images of the region behind the pharynx and of the vulva region were taken, both before and after the dendra2 photoconversion. Before photoconversion, dendra2's excitation maxima are at 490 nm and the emission maxima at 507 nm, similar to EGFP; after photoconversion, they change to 553 nm and 573 nm, in the red spectrum. In this study, Dendra2 was photoconverted by using 2% power of the 405 nm laser for 6 sec with 8 μ/s at a resolution of 1024 x 1024. For the analysis, in each image, the *C. elegans* body inside the photoconverted region was manually delineated as a mask, within which the ratio of the total red to green signal intensity is calculated as an indicator of relative amounts of respective proteins. This value was normalized for each image, *i.e.*, per animal and time point, by taking its difference from the similar ratio computed for masks on either side outside the photoconverted region. The data was further analyzed and visualized using GraphPad Prism 9.1.1. *P*-values were calculated using the One-way ANOVA. For details, see Supplementary Data 11.

## *C. elegans* proteomics

Approximately 1000 - 3000 *C. elegans* were harvested and washed 4 times by centrifugation and resuspension in physiological M9 buffer. Samples were then frozen at -80°C until extraction for all samples in parallel. 500 μL of extraction solution (8 M Urea, 25 mM NH4HCO3, Protease inhibitor (cOmplete tab, Roche Switzerland, 0.25 tablets per 1 ml of extraction solution) and 2 mM Na3VO4, 1mM PMSF) was added to each *C. elegans* pellet on ice. The samples were then processed by bead bouncing three times for 150 seconds using pre-chilled sample holders (-20°C). After centrifugation at 15000 g for 15 minutes (4°C), the supernatant was subjected to total protein quantification, and the sample was adjusted to 100 μg of protein. TCEP (Sigma-Aldrich) was added to 5 mM final concentration, and the samples were incubated at RT for 30 minutes. Reduced cysteines were alkylated with 10 mM iodoacetamide (Sigma-Aldrich) for 1 hr in the dark at room temperature. Samples were diluted 8x in 50 mM ammonium bicarbonate to reduce the urea concentration to 1 M, and protein digestion was performed overnight at 37 C by adding 2 μg of trypsin (Promega) per sample.

The day after, the samples were acidified with the addition of 5% TFA to achieve pH < 3. Desalting was performed using C18 spin columns (Nest group) as suggested by the manufacturer. Columns were wetted with 200 ul (1 CV) of 100% ACN and then equilibrated with 2 CV of 0.1% FA. Following sample loading, the resin was washed three times with 1 CV of 0.1% FA and 5% ACN. Peptides were eluted twice with 0.5 CV of 50% ACN in 0.1% FA and dried under vacuum. The dried peptides of MS-buffer (0.1% FA) with 1:30 iRT peptides (Biognosys) spiked in. The samples were then injected on a TripleTOF 5600 (Sciex, Concord, Canada). Peptides were separated at nano-flow liquid chromatography (NanoLC Ultra 2D, Eksigent) with a flow rate of 300 nL/min using a NanoSpray III source with a heated interface (Sciex, Concord, Canada). The source voltage was 2 kV. The used emitter (Peek 30 cm) was manually packed with 3 μm Reprosil pur (Maisch) beads. The peptides were separated using a 90 min linear gradient from 5% to 30% Buffer B (98% ACN and 0.1% formic acid in HPLC grade H2O) in Buffer A (2% ACN and 0.1% formic acid in H2O).

For data acquisition, the instrument was operated in positive ion, with high sensitivity SWATH-mode using 64 variable-width windows precursor isolation scheme, between 350 and 1500 m/z with a 1 m/z one-sided overlap. The Updated SWATH-window scheme is essentially described by Collins et al., 2017[67]. Accumulation time was set to 250 ms for the precursor survey scan and 50 ms for each of the 64 MS2 fragment ion scans, which resulted in a total cycle duty time of 3.5 s. Dynamic collision energy (CE) and collision energy spread (CES) was optimized for the fragmentation of each peptide following the rolling collision energy formula (CE= m/z * Slope + Intercept) with a collision energy spread of 15 eV. The SWATH data was searched in Spectronaut v13 (Biognosys) using a *C. elegans* FASTA (4352 entries) and directDIA with default BGS settings. Downstream analysis was performed in R. The mass spectrometry proteomics data have been deposited to the ProteomeXchange Consortium via the PRIDE partner repository with the dataset identifier PXD046470.

## RNAi clones and libraries

*Generation of RNAi clones:* We cloned the 942 bp *col-120* cDNA into pL4440, validated the correct insertion and sequence, and transformed this plasmid (pLSD051) into HT115.

*Generation of RNAi screening libraries:* For the target RNAi screen, we worked with 5 RNAi bacteria libraries containing selected RNAi clones to knock down specific gene classes or categories. The kinase library, two transcription factor libraries (bZip and TXN-factor Libraries), and the metabolism library was a generous gift from Gary Ruvkun (Harvard Medical School). We constructed our Matrisome library based on our definition of the *C. elegans* matrisome[25]. The library contains 652 RNAi clones of the 719 *C. elegans* matrisome genes. For

the missing 67 genes, no RNAi clones were available. The bacteria were picked from either the ORF-RNAi or the Ahringer RNAi libraries (both available from Source BioScience). Bacteria glycerol stocks of the clones were transferred with a pipette into 96-well plates, each also containing control wells with control bacteria RNAi clones (L4440 (empty vector control), *daf-2, bli-3, daf-16, skn-1, gfp, col-144*) and some empty wells (LB mixed with glycerol).

The same procedure was followed for generating the two validation screen libraries. Validation library I consisted of hits from the previous screen rounds, together with selected clones of hits from the P*col-12*::dsRed expression screen performed by the Ewbank lab and selected clones of genes from our literature research. Validation Library II consisted of hits from the Validation Library I screen.

## RNAi screen

The screen was performed on 96-well plates, each well containing 150 μl NGM with 100 μg/ml Ampicillin and 1mM IPTG, seeded with 8 μl concentrated RNAi bacteria. The bacteria were grown overnight in 96-deep-well plates in 800 μl LB containing 100 μg/ml Ampicillin and 12.5 μg/ml Tetracycline. The next morning, another 700 μl LB with Ampicillin and Tetracycline was added. After four additional hours of growth, the plates were centrifuged, the supernatant was discarded, and each well was filled up with 35 μl LB containing 100 μg/ml Ampicillin and 1mM IPTG to seed the 96-well NGM plates. For preparing the *C. elegans*, we used plates containing gravid adult *C. elegans*. We age-synchronized the animals at stage L1, by dissolving the parent but leaving the eggs intact and letting them hatch in an M9 medium with cholesterol, without food, so they stage arrest until all eggs were hatched. The next day, we placed 25 animals of our screening strains in wells of 96-well containing Normal Growth Medium (NGM); each plate was seeded with one of the clones from the 96-well library plates. *C. elegans* grew up at 25°C until day 2 of adulthood, as this temperature is needed to activate *glp-1* and *spe-9* mutations during development and make them sterile, later moving them to 20°C. The plates were scored on adulthood day 1 and day 8 under a fluorescence stereoscope, and the fluorescence of the animals was graded on a scale from 0-3 intensity. The result was counted as a hit when the average of the three to four replicates was at least 0.5 (lowest visible difference) over the control. For the gene ontology enrichment, WormCat was used[68].

## Time-course measurements of fluorescent expression reporter strains

The strains for time-course measurements were placed as L4s on plates containing 50 μM FUdR and were scored on the indicated days under a fluorescence stereoscope. For each animal, the fluorescence was graded on a scale from 0-3 in 0.5 steps. Per measuring rounds, 20-30 *C. elegans* were used. The data was further analyzed and visualized using GraphPad Prism 8.2.0. *P*-values were calculated using a two-way ANOVA.

## EMB-9/PAT-3 co-localization experiments

For the imaging of a potential co-localization of the EMB-9::mCherry and PAT-3::GFP in *C.elegans*, the animals were placed as L4s on plates containing 50 μM FUdR. Images were taken on day 1, day 3, and day 8 of adulthood using an Olympus FluoView 3000 microscope. For the RNAi experiments, the animals were placed as eggs on plates seeded with control L4440 RNAi or *daf-2* RNAi bacteria. The RNAi NGM plates contained 100 μg/ml of Ampicillin and 1mM IPTG. The L4 animals were moved to RNAi NGM plates that additionally contained 50 μM FUdR.

For the analysis of colocation, we separately assessed this for the body wall muscles around the head region, midbody region, and tail region by imaging and manually selecting these regions for each *C. elegans* (all images and area selections are provided in Supplementary Data 11). In our study, *colocation* is defined as very close proximity (beyond the pixel resolution of the employed microscopy imaging) of

the proteins and hence their fluorescence marker signals. To that end, red and green intensity (signal) at each pixel indicates the amount of the corresponding protein within the space covered by that image pixel. If these intensities are *similar* for the same pixel, this would mean similar amounts of each molecule within that pixel (thus in "very close proximity" per definition). Instead of the concept of similarity, *i.e.*, being "equal", we chose to quantify their correlation (*i.e.*, the null-hypothesis of them being in relation with a fixed ratio) because intensity equality between red and green markers would require strict constraints, such as the imaging system scaling raw reading similarly to all RGB values, these different markers reacting the exact same way to optic excitation, each marker staining the molecules in the same fashion, etc. Instead, a correlation analysis checks for an arbitrary (linear) relation model between these markers occurring together in pixels (*i.e.*, colocating). Accordingly, we computed the correlation of per-pixel red-to-green signal across the image, per animal, and time point. All pixel quantifications and calculations are provided in Supplementary Data 11. Since the structures we aim to quantify are smaller than the pixel size, the intensity red/green quantifies how many/much collagen and integrin exist in each pixel. Then, observing more green where the red is, and vice versa (*i.e.*, the correlation metric we used) indicates their co-occurrence spatially. The term "disassociation" refers to our quantification of the loss of spatial relationships between collagen (red) and integrin (green).

## Lifespan assays

Manually lifespan assay. In brief, L4 animals were picked onto culturing plates containing 50 μM FUdR. TJ1060 *spe-9(hc88)* I; *rrf-3(b26)* II and CB4037 *glp-1(e2141)* III were grown at 25°C until day 2 of adulthood and then placed on RNAi plates (without FUdR) at 20°C for the remainder of the lifespan. The *spe-9(hc88)* is a temperature-sensitive sterile mutation[69]. SPE-9 is a transmembrane protein on the sperm required for oocyte interaction.

Lifespan machine assays. In brief, L4s were washed onto RNAi culturing plates containing 50 μM FUdR until day 5 of adulthood at 20°C, and then placed on special RNAi plates and placed into the lifespan machine. For experimental details, setup, raw data, and statistics, see Supplementary Data 8.

Comparative automated lifespan analysis: To compare the longevity-promoting effects of *daf-2* RNAi on different *C. elegans* matrisome mutants, the mean lifespan of *daf-2* RNAi was plotted against the corresponding mean lifespan on L440. N2 data from multiple lifespan assays was compiled, representing the general lifespan extension of *daf-2* RNAi in N2. For the analysis, the different strains were grouped according to their respective matrisome category. The code can be found in Supplementary Data 7 and on Github (https://github.com/Ewaldlab-LSD/DiagonalPlots).

*For compound-treated lifespans:* TJ1060 *C. elegans* were bleached, and the eggs were distributed on NGM plates to grow at 25°C. On day 4 of adulthood, 100 *C. elegans* were placed on 10 NGM plates containing 2% ribose and nystatin and another 100 on 10 control plates. LSD2052 worms were also bleached, and eggs were distributed on NGM plates at 25°C. This time, 50 *C. elegans* were transferred on two plates with 1 mM genipin, 50 on two DMSO (90.4 μl DMSO/10ml NGM) (Aldrich, M81802) plates, and another 50 on two control plates. All samples were put at 20°C on day 4 of adulthood. In the following days, plates were checked for dead animals every day. Lifespan experiments with aminoguanidine were performed by using the lifespan machine. N2 worms were bleached and incubated in M9 for 2 days at 20°C on a rotor to synchronize the animals at the L1 stage. L1 animals were then transferred on NGM plates and grown to larval stage L4. From that time on, nematodes were grown on NGM plates containing FUdR. On day 4 of adulthood, we transferred approximately 30 worms on one plate, 4 plates in total per experimental condition. The plates were prepared

with 1 mM, 50 mM, and 100 mM aminoguanidine and control plates without any interventions. All plates contained FUdR and nystatin.

## Quantification of cuticle thickness using electron microscopy images

Transmission electron microscopy (TEM) images of *C. elegans* age day 2, day 7, and day 15 were downloaded from wormimage.org. Only transverse sections were considered as they were thought to minimize the error in measurement due to oblique cutting angles. Regions distorted by the cutting process were excluded from the analysis. Suitable regions were manually traced, and the resulting binary masks (colored curves, one pixel thick) were exported to Matlab (Supplementary Fig. 5a–c). As the borders of every cuticular layer were traced separately, both the total thickness and thickness of the individual layers could be measured.

To determine the mean distance between two-layer borders (*i.e.,* the mean thickness of the cuticle layer) represented by two binary masks, A and B, the Euclidean distance transform of mask A was calculated, leading to $A_{dmap}$. The Euclidean distance transform of A assigns to every pixel in A the distance to the nearest non-zero pixel. By element-wise matrix multiplication of $A_{dmap}$ and B, the nearest distance to A of every pixel in B can be calculated, leading to vector b (Supplementary Fig. 5d). By repeating the process while swapping A and B, the closest distances to B for every pixel in A can be calculated. The distance between two curves, A and B, was then defined as the mean value of the shortest distance to the other curve for every pixel in both A and B. The data set used contained digitized TEM images of various different magnifications (*m*) (1200–15500x). Every image contained a scale bar of a length of 2.5 cm. For analysis, the length of the scale bar in pixels (*s* [px]) was measured. The conversion factor (*f* [μm/px]) was calculated according to formula (1) and multiplied with the final measurements to convert them to micrometers.

$$f = 25000/(m \cdot s) \tag{1}$$

## Fluorescence microscopy of YAP-1::GFP expression

Strain YAP-1::GFP was maintained on OP50 NGM plates at 20°C. Gravid adults were treated with sodium hypochlorite solution to obtain eggs. The eggs were grown on L4440 and *daf-2* RNAi plates until the L4 stage. At L4, the plates were transferred to 25°C or kept at 20°C. Then, on day 1 of adulthood, approx. 30 animals were mounted onto a 2% agarose pad and anesthetized with 20 mM levamisole for imaging. Animals were captured at 10X with one or two fields of view and then stitched together later using ImageJ. Quantification of the total fluorescence intensity was done by running a python script, GreenIntensityCalculator (available publicly in Github-Ewaldlab: https://github.com/Ewaldlab-LSD) in ImageJ[21]. Statistical analysis of the quantified data was done using Graph Pad Prism software (GraphPad Prism 9.0). The experiment was performed in three independent biological batches.

## Fluorescence microscopy for YAP::GFP expression on hemidesmosome-containing structures

Strain *ihIs35* YAP-1::GFP was maintained on OP50 NGM plates at 20°C. Gravid adults were treated with sodium hypochlorite solution to obtain eggs. The eggs were grown on L4440 and *daf-2* RNAi plates until they became young adults. At the young adult stage, the plates were transferred to 25°C. Then, on day 3 of adulthood, approx. 30 animals were mounted onto a 2% agarose pad and anesthetized with 20 mM levamisole for imaging. Animals were captured at 100X with one posterior field of view. Quantification of the distribution of the categories of hemidesmosome-like phenotype was done manually.

Category 0-1 were allotted to absent to faint appearance of hemidesmosome-containing structures (as in most L4440 control conditions) while 2-5 category represent the graded intensity of the present hemidesmosome-containing structure (as observed to be more the case in *daf-2* RNAi condition). Statistical analysis of the quantified data was done using Graph Pad Prism software (GraphPad Prism 9.0). The experiment was performed in four independent biological batches.

## Confocal microscopy for colocalization of YAP-1::GFP with VAB-10::mCherry

Transgenic strain LSD2207 YAP-1::GFP; VAB-10A::mCherry was grown on L4440 or *daf-2* RNAi plates at 20°C. At the young adult stage, animals were shifted to 25°C. After 24h, animals were mounted onto a 2% agarose pad and anesthetized with 20 mM levamisole for confocal imaging. Animals were captured with a confocal laser scanning microscope (Olympus Fluoview FV3000), using 60X oil objective. Images were taken at an intensity of 0.7% for the red channel and 30% for the green channel, at 4.00 zoom, at a resolution of 1024 × 1024 pixels, and with a gain of 1.0. The experiment was performed in three independent biological trials.

## Pressure application and quantifying GFP intensity

Approximate 30 L4 animals were picked onto the center of a 60 mm OP50 FUdR (no FUdR for LSD2002 animals because containing temperature-sensitive sterile mutation *spe-9(hc88)*) containing plate for each condition in technical replicates of two. Then, with the help of a sterilized spatula, a 2 cm x 2 cm chunk of agar was cut and transferred to a glass slide gently. After that, another glass slide was pressed upon this agar chunk gently, sealing the glass slides with the help of tape. For the pressure condition, an inverted 50 mL water-filled falcon was placed on top. The approximate pressure was calculated as the force divided by the area. Thus, the pressure was $(0.05 \text{ kg weight} \times 9.81 \text{ m/s}^2 (=g))/ (0.04 \text{ m}^2 \text{ agar area}) = 12.3 \text{ Pa}$. This pressure device was kept at 20°C or 25°C, as indicated in the Fig. legend, for three days, after which the animals were imaged. To assess fluorescence intensity, the agar chunk was transferred from the pressure device to a fresh OP50 plate, and let the animals moved out of the agar for about half an hour. Animals were then mounted onto a 2% agarose pad and anesthetized with 20 mM levamisole for imaging. They were captured at 10X with one or two fields of view and then, stitched together later, using ImageJ. Quantification of the total fluorescence intensity was done by running a python script, GreenIntensityCalculator (available publicly in Github-Ewaldlab: https://github.com/Ewaldlab-LSD) in ImageJ[21]. Statistical analysis of the quantified data was done using Graph Pad Prism software (GraphPad Prism 9.0).

## Reporting summary

Further information on research design is available in the Nature Portfolio Reporting Summary linked to this article.

## Data availability

The data generated in this study are provided in the article and its Supplementary Data 1-14. Data are available via ProteomeExchange with the identifier PXD046470.

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

## Acknowledgements

We thank Mariam Baghdady, Kieran Toms, Vira Chea, and Katharina Tarnutzer for help with the screen, Elisabeth Jongsma for her help scoring GFP, Özlem Altintas for help with lifespan scoring, Katrien De Bock and Ewald lab members for critical discussions of the manuscript,

Ann Rougvie for pJA1 [P*col-19*::GFP] plasmid, Michel Labouesse for OD76, ML2600 strains, Alex Hajnal for AH4617, AH3437, AH3284 strains, Cathy Savage-Dunn for CS637, CS678, Harald Hutter for VH2847 strain, Taina Pihlajaniemi for TU1 strain, WormBase for curated gene and phenotype information, Tobias Schwarz and the support of the Scientific Center for Optical and Electron Microscopy ScopeM of the Swiss Federal Institute of Technology ETHZ. Some strains were provided by the CGC, which is funded by the NIH Office of Research Infrastructure Programs (P40 OD010440). Figure 3n was created with BioRender.com (Publication license JH23SSQB6M). Funding from the Swiss National Science Foundation PP00P3_163898 and 190072 to ACT, CS, AG, and CYE, and the European Research Council AdvG grant 670821 to RA is acknowledged.

## Author contributions

All authors participated in analyzing and interpreting the data. C.Y.E., A.C.T., and C.S. designed the experiments. C.S., A.G., A.M.H., S.A.D., and C.Y.E. performed lifespan assays. A.M.H. and C.S. performed the comparative automated lifespan analysis. C.S. and AF performed proteomics. S.A.D. performed FRET-sensor experiments. I.S. programmed FRET image analysis. C.S. performed the bioinformatic analysis. M.H. quantified cuticle thickness. O.G. programmed and performed the image analysis. A.G. and C.Y.E. performed the pressure experiments. A.C.T. performed all other experiments. R.A. and V.V. supervised consolidated experimental procedures and edited the manuscript. C.Y.E. wrote the manuscript in consultation with the other authors.

## Funding

## Competing interests

The authors declare no competing interests.
