## [Peer Review File · Nature Communications]

Longevity interventions modulate mechanotransduction and extracellular matrix homeostasis in *C. elegans*REVIEWER COMMENTS

Reviewer #1 (Remarks to the Author):

It is widely acknowledged that ECM quantity and integrity is tightly linked to the ageing process, yet how the ECM coordinates longevity and body fitness remains largely unknown. In this manuscript titled “Longevity interventions modulate mechanotransduction and extracellular matrix homeostasis in *C. elegans*”, Teuscher et al., uncovered an intricate feedback network involving age-dependent ECM remodeling, hemidesmosomes, mechanical tension and YAP signaling. By employing multiple large-scale screening and omics techniques, the authors identified three classes of collagens with distinct dynamic patterns during *C. elegans* ageing. The authors then introduced six different types of longevity intervention methods and showed that aging retardation has significantly impact on each class of ECM proteins, which in turn affects the lifespan. Interestingly, overexpression of a single ageing-regulated collagen is sufficient to extend *C. elegans* lifespan and enhance ECM composition, whereas collagens that were not altered by longevity have no such effect. It suggests that the longevity-promoting function is only restricted to a specific sub-group of ECM proteins. By utilizing three different methods to alter mechanical tension in the hypodermis, combined with a targeted RNAi screen, the authors identified the mechanical force as the key link between ECM remodeling and the ageing process. Further investigation revealed that the mechano-sensitive hemidesmosome structure with YAP-1 as its physical interaction partner and effector is responsible for mediating this crosstalk. In general, the discoveries reported by this manuscript are intriguing, timely and important. The data are of high quality, the logic is sound and the conclusions are solid. A particular strength of this study is the use of multiple independent approaches to test a single hypothesis at different angles, and careful elimination of other possibilities by additional experiments. I only have a few minor recommendations to further improve this manuscript before publication:

1. It is seemingly contradictory that the *unc-52(e699,su250)* mutation enhances *col-144* expression (Figure S8C-D), higher *col-144* expression extends lifespan (Figure 3F), yet the *unc-52(e699,su250)* mutants not only have shorter lifespan but also abolishes lifespan extension by *daf-2* (RNAi)(Figure 5G). Some explanations should be provided to address this dilemma.

2. Another potential contradiction is that on one hand, mildly higher pressure (tension increase in the hypodermis) promotes *col-144* expression (Figure 7L). On the other hand, RNAi that causes muscle dysfunction and animal paralysis (*pat-2*, *pat-3*, *unc-112*, etc.) also promotes *col-144* expression (Figure 4B). And it is generally believed that loss of muscle contraction decrease tension in the hypodermis (Nature, 2011; Development, 2019). Some discussion is needed to sort out this problem. For example, could there be a dose-dependent effect of hypodermal tension on collagen expression? Or is it possible

that apical versus basal hypodermal tension changes have opposite effects on YAP activity and collagen expression.

3. The previous publication by Ewald et al., Nature 2015 is closely related to this study. It should be cited and its content compared with these new findings. For example, some longevity intervention and collagen overexpression data included in this manuscript were already reported in the 2015 paper and therefore should be acknowledged. The authors also need to discuss how much does the SKN-1-mediated process contribute to longevity-regulated collagen remodeling, and whether there are any crosstalks/interactions between SKN-1 and this newly discovered feedback loop .

4. The graphic abstract could benefit from a few modifications, as the current version does not have a good enough presentation of the involvement of the ageing process in this feedback loop.

5. A few rounds of text editing is recommended to remove typing mistakes such as “LET-80S” (line 28, should be LET-805), and “disappeared faster from the ECM faster than...”(line127, one “faster” should be deleted).

6. Panel labels should be added to supplementary Figure 8A.

Reviewer #2 (Remarks to the Author):

Using proteomics and in-vivo reporter systems, Teuscher A.C. et al., identify at least three distinct dynamic ECM composition changes during aging. The authors go on to identify a regulatory feedback loop sensing ECM integrity via a mechano-sensitive mechanisms involving Yap-1, that responds to IIS longevity interventions. I appreciate how much work the authors have put into this manuscript. The work presented is extensive. However, the manuscript suffers from a major problem. The interchangeable use of “diverse” longevity interventions in ECM parameters measured isn’t justified and obscures the interpretation of the data. Likewise, the interchangeable use of distinct reporters when no justification seems plausible makes it hard to evaluate whether the interpretation of the data is correct. Major efforts need to be made to standardize the manuscript from an experimental point of view for a clearer and consistent message and to gain support for publication from this reviewer.

Points to be addressed.

1. Why constantly refer to longevity interventions for generalization when only two interventions were tested? Please refrain from using generalizations.

2. How is Figure 1K representative of data in Figure 1J? It looks that for Col-120::GFP there's a clear higher signal at D8 than D1. Also, the expression seems to be mostly localized in the intestine rather than the extracellular matrix. Why is this the case and does this "mis-localization" have any role? For example, is this expression in the intestine responsible for its lifespan effects and other proteomic changes observed throughout the manuscript? Tissue specific expression of Col-120::GFP experiments should be performed (e.g lifespans, proteomics). In Figure 1 J and K, and Supplementary Figure 1 J and K, the differences in col-120 mRNA and especially COL-120 protein levels across different timepoints do not correspond entirely (e.g., the levels of COL-120::GFP signal drop much more drastically in the Figure 1 J than in the corresponding table in Supplementary Figure 1 J) – how can this be explained? Were the same reporters used for both figures? Also, how did the authors assign the numerical values to expression levels in the Supplementary Figure 1?

3. "While COL-120 protein tagged with GFP gradually disappeared from the cuticular ECM during ageing, slowing ageing by RfS showed COL-120 in the ECM for a prolonged time (Figure 2A, 2B)".

I agree with this but isn't it to be expected? More important is whether this change is proportional to their lifespan extension. In other words, is there any gain in healthspan. Likewise, similar measurements should be performed for the Dendra reporter. Why was daf-2 RNAi used? Are these effects similar in daf-2 worms of different mutant alleles (class I vs class II)? More importantly, is COL-120 a good example to assess this question, considering that it disappears from cuticle quite rapidly, hence the Dendra conversion and follow-up measurements were done in worms that are only Day 2 and Day 4 adults, respectively?

4. "We found a stark increase in FRET transmission during old age (day 12) compared to young (day 2), and this age-dependent increase in FRET transmission was lower in long-lived glp-1 mutants at day 8 of adulthood compared to normal-lived wild-type controls (Figure 2H-J, Supplementary Table 4). This suggests that longevity interventions may counteract age dependent crosslinking of pattern II collagens."

I don't understand why glp-1 mutants are being used here and not daf-2 RNAi for consistent reporting and mechanistic understanding. Why are distinct longevity interventions being clustered as one? These interventions have specific mechanisms involved and should be treated independently. I do not agree with a wide generalization as the one being made by the authors. As before, for figure 2I and 2J, measurements should be done in proportion to their respective longevities since improvements in chronological age are expected.

5. When discussing the results depicted in Figure 3B, the authors mention that different longevity interventions counteract the age-related changes in protein abundances (lines 210-212). Nevertheless, the graph depicts only the changes for Day 1/Day 2, or even L4 worms, except in the case of eat-2 mutants, where the data for Day 15 are shown. What is the reasoning behind showing all data in one graph given that the abundance of each protein is regulated in an age-dependent manner with distinct expression patterns with age? Ideally, one would show data for proteome changes in older worms across all these longevity interventions. As such, this panel does not sufficiently support the respective claims being made. Since I do not think this is fair to ask for such experiment to be done, I would ask the authors to consider focusing on the longevity interventions that this work primarily focuses on (daf-2 and glp-1) and perform proteomics experiments in late life and at mean/median lifespan for each intervention compared to control.

6. “during ageing, and that longevity intervention started with more collagen mass which declined at a similar rate during ageing (Figure 3C, 3D).”

This goes with my previous comments that improvements seem to occur early in life but do not seem to improve aging rate and makes one wonder about the potential implications of such findings and their general interest to the aging field.

Likewise, In line 188, the subtitle does not correspond to the text in lines 194-195 (i.e., “independent of daf-2 RNAi longevity interventions”) – the authors should clarify this discrepancy. Do longevity interventions slow or not the accumulation of basement membrane collagens? Also, most data seems to suggest a delay rather than a slowing down. In the same paragraph, the authors mention that the observation made is consistent with thickening of the human basement membranes with age. However, similarly as in the point 3, the photoconversion and quantification were done at Days 2 and 4, respectively, therefore, not in worms aged enough to describe this as an aging/anti-aging effect.

7. Also In Figure 3 and the paragraph related to it, the authors argue for existence of a feedback loop that is based on the abundance of a certain collagen (e.g., COL-120, COL-144, COL-10). If the abundances of other matrisome proteins are changed in response to this, and can be modulated via longevity interventions, how do the authors explain that similar changes in levels of these “key collagens” are not present between the longevity models and the worms with COL-120 OE (see panels 3B, 3H, 3K and 3I)? Isn't this the central tenet of this manuscript? Yet, the data seems not to support this convincingly.

8. “To test whether this correlation is associated with prolonged production of the pattern I collagens, we used transgenic animals expressing GFP driven by the collagen col-144 promoter, whose expression gradually declines during aging 5,31” .

The same experiments done here for col-120 should be done for col-19 and emb-9. Including splitting animals based on GFP levels of the reporters (col-19 and emb-9), and rnai (col-19 and emb-9) and OE studies (except for emb-9 which has already been done), for lifespan. If there is any generalization possible in the mechanisms being proposed, and the rationale employed by the authors, why did the emb-9 oe did not lead to a shortened lifespan? This is a critical point, since the classification of the three patterns proposed in figure 1 may be mechanistically meaningless without concrete proof that such classification is linked to a biological effect (e.g lifespan), which is what the authors are trying to imply in this manuscript.

9. "Furthermore, overexpressing collagens that were not altered by longevity interventions was insufficient to increase lifespan (Supplementary Figure 6, Supplementary Table 7), suggesting the unique properties of longevity-promoting collagens. To identify downstream mechanisms mobilized by collagen overexpression".

How are these two statements compatible? On one hand the authors state that no generalization can be made and then use one specific intervention to find "generalized mechanisms" of collagen overexpression! This is unfortunately, very confusing and unconvincing.

10. Why were daf-2 mutants used for J and K and RNAi for LM. Given the different longevity curves from these two interventions, how much does this affect the interpretation of the results? Do the authors observe the same results if a daf-2 mutant is used instead of the mutant for 3L and 3M?

11. "Since the decline in the pattern I collagen transcription preceded its decline or remodelling out from the cuticle, we scored col-144 promoter driven GFP levels (Pcol-144::GFP) that progressively declined from day 1 to day 8 of adulthood (Figure 4A, Supplementary Figure 7A, 7B)"

Same issue as before, why wasn't the col-120::GFP promoter used when all the work has been done so far with this gene? When the authors themselves state that each collagen has a unique properties, without additional data showing that col-144 and col-120 have exactly the same biological properties, I am not convinced about the value of this screen in the context of the data up until this point.

Reviewer Response letter NCOMMS-23-00789-T

Nature Communications manuscript NCOMMS-23-00789-T

REVIEWER COMMENTS

Reviewer #1 (Remarks to the Author):

It is widely acknowledged that ECM quantity and integrity is tightly linked to the ageing process, yet how the ECM coordinates longevity and body fitness remains largely unknown. In this manuscript titled “Longevity interventions modulate mechanotransduction and extracellular matrix homeostasis in *C. elegans*”, Teuscher et al., uncovered an intricate feedback network involving age-dependent ECM remodeling, hemidesmosomes, mechanical tension and YAP signaling. By employing multiple large-scale screening and omics techniques, the authors identified three classes of collagens with distinct dynamic patterns during *C. elegans* ageing. The authors then introduced six different types of longevity intervention methods and showed that aging retardation has significantly impact on each class of ECM proteins, which in turn affects the lifespan. Interestingly, overexpression of a single ageing-regulated collagen is sufficient to extend *C. elegans* lifespan and enhance ECM composition, whereas collagens that were not altered by longevity have no such effect. It suggests that the longevity-promoting function is only restricted to a specific sub-group of ECM proteins. By utilizing three different methods to alter mechanical tension in the hypodermis, combined with a targeted RNAi screen, the authors identified the mechanical force as the key link between ECM remodeling and the ageing process. Further investigation revealed that the mechano-sensitive hemidesmosome structure with YAP-1 as its physical interaction partner and effector is responsible for mediating this crosstalk. In general, the discoveries reported by this manuscript are intriguing, timely and important. The data are of high quality, the logic is sound and the conclusions are solid. A particular strength of this study is the use of multiple independent approaches to test a single hypothesis at different angles, and careful elimination of other possibilities by additional experiments. I only have a few minor recommendations to further improve this manuscript before publication:

Response: We thank the reviewer for this positive feedback and addressed the recommendations below.

1. It is seemingly contradictory that the *unc-52(e699,su250)* mutation enhances *col-144* expression (Figure S8C-D), higher *col-144* expression extends lifespan (Figure 3F), yet the *unc-52(e699,su250)* mutants not only have shorter lifespan but also abolishes lifespan extension by *daf-2* (RNAi)(Figure 5G). Some explanations should be provided to address this dilemma.

Response: This is an important point the reviewer has raised. Others had pointed to this before, and we hoped our model and the short explanation we had provided would suffice to explain these results. At first, we also perceived this as a dilemma until discussions with experts in the

mechanobiology field. The model we came up with was very similar to the shear force model in blood vessels by Humphrey and Schwartz (see below, copied from Fig 3 of their paper <https://doi.org/10.1146/annurev-bioeng-092419-060810>).

After reading more about systems biology, we realized that this dilemma only exists in a linear model without feedforward and feedback loops (Uri Alon, *An Introduction to Systems Biology*, CRC Press). Due to shortening our manuscript and prompted by your question, we realized that we mistakenly excluded explaining our model shown in Figure 7P (copied below).

Allow us to explain as a direct response below, and an adapted version of this is found as a new supplementary text file. Please note that although this proposed model can explain many results, further testing is needed in future research.

Proposed biomechanical model of ECM homeostasis and longevity

Based on our findings here, we propose a model that is adapted from the mechanobiological regulation of arterial walls, which includes the interactions of smooth muscles, endothelial cells, fibroblasts, and ECM remodeling (by Humphrey and Schwartz 2021; doi: 10.1146/annurev-bioeng-092419-060810). The mechanical homeostasis model requires an input (mechanical stimulus), a set point to where the homeostasis should be regulated back (negative feedback), a sensor telling the actual state, and an output (biological response). The cells most likely “perceive” the difference between the set points to what is sensed by the sensor and adapt to the difference to reset to the set point (homeostasis). For instance,

increased blood pressure will thicken, whereas decreased blood pressure will thin the arterial wall to reset the intramural stress toward the set point. Thus, re-establishing equilibrium and stability (i.e., homeostasis) by integrating mechanical stimuli with biological feedback processes.

In our proposed model (Figure 7P), the hemidesmosomes act as the mechanical sensor and set point, integrating the mechanical stimuli coming from muscle contraction, internal (hypodermal) and external pressures, and shear forces. C. elegans is basically a liquid-filled tube that needs to maintain internal pressure. The hemidesmosomes (ceHD) are central since they connect through the basement membrane via the integrins to the body wall muscles and span through the hypodermis via the intermediate filaments and anchoring to the exoskeleton (the cuticle). We refer here to this as the apparatus: coupling muscle strength through the ceHD to the resistance of the cuticle). When the body wall muscles contract, the force generated needs to be transmitted via ceHD to the exoskeleton (cuticle) in order for the worm to move (like a muscle to bone to generate movement). Thus, the ceHD couple two distant and distinct ECMs (basement membrane and cuticle) and are essential to transmit the forces from the muscle to the exoskeleton for the motility and movement of C. elegans.

Description of the proposed biomechanical model

Let us assume that post-development, the hemidesmosomes have established the set point and sensor, and the system is in homeostasis. When we place worms under external pressure and when the muscle pulls on the ceHD, the deviation-actuation by the set point and sensor will be different and will lead to adapting the cuticle resistance by inducing the expression of cuticular collagens (col-120 or col-144, Figure 7J-M). Our genetic data suggest that this mechanical deviation-actuation is translated into an upregulated gene expression of these cuticular collagens by nuclear translocation of the transcription factor YAP1. This ECM production/assembly/remodeling will subsequently strengthen the cuticle. With a certain time delay, the strengthening of the cuticle is then sensed by the pulling forces of the ceHD, and we assume that this ultimately leads to a negative feedback loop stopping cuticular collagen expression, thereby driving the system back to equilibrium and homeostasis again.

Evidence for our proposed biomechanical model for this work and other groups

Our RNAi screen identified several molecular targets interfering with this feedback loop. Anatomically, genes that blocked this feedback loop were either components of the hemidesmosomes, responsible for building or maintaining the ceHD, or closely associated with the ceHD (Figure 4B). Genetic findings from other groups make sense if interpreted with our ceHD biomechanical model. For instance, in our recent review, these genetic findings are discussed in light of a mechanobiological model of ceHD (accepted in Journal of Cell Science: JOCES/2023/260987. Mechanotransduction through hemidesmosomes during aging and longevity. Collin Yves Ewald and Alexander Nystrom). The paragraphs below in “...” are an adapted version of an excerpt of this review.

“The cuticular extracellular matrix (ECM) binds to specific transmembrane receptors, namely MUA-3 (referred to as fibrillin in mammals) and MUP-4 (also known as matrilin). These

receptors connect to VAB-10 (plectin) located in the plasma membrane of the apical hypodermis (see Figure 4). Intermediate filaments traverse the hypodermis and extend to the basal site, where VAB-10 once again binds to the LET-805 receptor (myotactin) for anchoring into the basement membrane ECM (see Figure 4). The basement membrane, composed of collagen IV and UNC-52 (known as perlecan), interacts with a heterodimer formed by PAT-2 (integrin α) and PAT-3 (integrin β) on the muscle cell surface (see Figure 4) (Zhang and Labouesse, 2010). Collectively, these elements constitute the force-bearing HD-like structure in *C. elegans* (ceHD).

In addition to their role in force transmission and cytoskeleton adaptation, the ECM proteins near the ceHDs also influence ECM remodeling. For example, ECM proteins containing the zona pellucida domain, NOAH-1 and NOAH-2 (Figure 4, screen hits), play a crucial role in maintaining mechanoreceptor potentials and cuticular ECM remodeling (Frand et al., 2005; Vuong-Brender et al., 2017). During the development of *C. elegans*, cuticular ECM remodeling takes place during molting. Interestingly, the loss of *unc-52*, *pat-3*, or *unc-95* (corresponding to paxillin in mammals) leads to molting defects associated with the ECM (Frand et al., 2005; Zaidel-Bar et al., 2010). By contrast, mutations in muscle myosin *unc-54*, which is essential for muscle contraction, or in *unc-13*, which is important for neurotransmitter release, do not result in the expression of cuticular collagen (Broday et al., 2007). The *C. elegans* cuticle surface is characterized by circumferential ridges known as annuli, and these annuli attach to ceHDs at the lower points called furrows (McMahon et al., 2003). Observations using scanning electron microscopy indicate abnormal, branched, or flat cuticular annuli in *unc-52* mutants, whereas *unc-13* mutants, which exhibit defects in neurotransmitter release, do not display such abnormalities (Broday et al., 2007). This suggests that defects in *unc-52*/perlecan can impair the proper morphology of the cuticle ECM, as this basement membrane heparan sulfate proteoglycan is a critical component for ceHD function. Additionally, *pxn-2* (peroxidase) is involved in promoting sulfhydryl crosslinks of basement membrane collagen IV, thereby regulating its mechanical properties. Defects in *pxn-2* can be mitigated by mutations in ceHD components like *let-805*, *vab-10*, and *unc-52* (Gotenstein et al., 2018), suggesting that changes in either the cuticular or basement membrane ECM are detected and mediated by ceHDs.

Aligning with the concept that ceHDs coordinate ECM remodeling through mechanical induction of gene expression, it is essential to understand the physiological effects of mechanical manipulation on the worm. Mechanical compression of ceHDs by subjecting *C. elegans* to hypergravity hampers the migration of motor neurons over the muscle, and mutations in *vab-10*, *unc-52*, and other ceHD components rescue neuronal migration (Kalichamy et al., 2020). Conversely, stretching ceHDs exposes the SH3 domain of VAB-10, enabling mechanosensitive signaling crucial for embryonic elongation (Suman et al., 2019). Notably, hydrostatic pressure increases the production of cuticular collagen *col-107* mRNA, along with lifespan (Watanabe et al., 2020). These findings indicate that mechanical compression affects a wide range of physiological implications from development to aging.”

Implications of mutations in the hemidesmosomes based on our proposed biomechanical model:

The mutation in perlecan *unc-52*(e699,su250) is from birth, meaning that these animals develop with “less intact” or weaker hemidesmosome structures when forces pull on them. The forces of

the muscle will be transduced through the hemidesmosome to the cuticle, which acts as an exoskeleton and resistance for the worms to move (Graphical Abstract). Given the idea that the hemidesmosomes act as a set point and sensor (Figure 7P), a “weaker” hemidesmosome will lead to a higher set point of cuticular collagen col-144, presumably adapting muscle-pulling force to strengthen parts of the cuticle. Presumably, muscle strength, transducer, and exoskeleton are adjusted and coupled to each other. We will refer to this coupling as an apparatus (muscle-hemidesmosome-cuticle). When we place worms under pressure, the muscles have to work harder to move and thus adapt the whole apparatus, including some cuticular collagens, which we observe as an increase of Pcol-144::GFP.

We think the hemidesmosomes are the integrators and also the sensor of this force transduction. Since at normal conditions, unc-52(e699,su250); Pcol-144::GFP worms have higher levels of col-144 compared to wild type, but under pressure where wild-type Pcol::GFP animals show an increase in col-144 expression, the unc-52(e699,su250); Pcol-144::GFP worms do not (Figure 7M), we think that unc-52(e699,su250) mutants have a higher set point from birth and thus higher Pcol-144::GFP levels, but fail to adapt to the pressure since the sensor is broken.

The actuation-mediated deviation of the set point and the sensor are the readouts, and our data show that the nuclear translocation of yap-1 is the response to this deviation, initiating the response that ultimately restores homeostasis. However, unc-52(e699,su250) mutants probably failed to integrate the difference between the set point and sensor, which explains that under pressure, unc-52(e699,su250) did not increase col-144 gene expression (Figure 7M). Thus, the higher Pcol-144::GFP levels of unc-52(e699,su250) compared to wild-type background might not be perceived as an “overexpression” in an unc-52(e699,su250), but we think this is due to just a higher setpoint of col-144 expression.

At permissive temperature (15°C), the penetrance of unc-52(e699,su250) mutation becomes visible during aging, as the midbody region of these mutants becomes progressively paralyzed (Ben-Zvi et al., 2009). This is due to the collapse of proteostasis in early C. elegans adulthood, leading to faulty proteins, such as the mutated perlecan unc-52(e699,su250) (Ben-Zvi et al., 2009). As the perlecan is located and couples the ceHD to the integrin, a faulty perlecan will lead to the uncoupling of the apparatus. We observed this with the unc-52(e699,su250) mutants appearing less defective on day 1 (Figure 5A, 5B) compared to day 8, which correlates with midbody regions being mainly affected (Figure 5B) by the age-related and progressive functional loss observed during aging. This suggests that during aging, the progressive decline of ceHD function will abolish the biological output. Longevity interventions (reducing Insulin/IGF-1 signaling and glp-1) can delay the age-related decay of ceHD, presumably by improving protein homeostasis, but only to a certain extent (by 2-3 days).

Connection between integrin integrity, structure, physiology, and lifespan

Integrins transmit mechanical forces from the ECM to the cytoskeleton, and vice versa, whereby they get physically connected to the cytoskeleton by linker proteins, including talin/TLN-1 and

integrin-linked kinase (ILK/PAT-4), and RHO-associated kinase (ROCK/LET-502) regulate the dynamic reorganization of cytoskeletal proteins by regulating the phosphorylation of myosin motors ¹. Adult-specific upshift to 20°C of temperature-sensitive loss-of-function ROCK/let-502 mutants did not suppress daf-2(RNAi) longevity (Supplementary Figure 10A, Supplementary Table 7). While the dephosphorylation of the β -integrin receptors at tyrosine 792 in the membrane-proximal NPXY motif promotes integrin activations via talin recruitment ², phospho-deficient integrin PAT-3(Y792F) mutants, as well as talin/tln-1 mutants treated with daf-2(RNAi), were still long-lived (Supplementary Figure 10B, Supplementary Table 7). Loss of focal adhesion kinase kin-32 increased lifespan (Supplementary Figure 10C, Supplementary Table 7). Heterozygous ILK/pat-4 mutants were shorter-lived and blocked daf-2-longevity (Supplementary Figure 10D, Supplementary Table 7), consistent with severe pat-4 knockdown leads to detachment of the cytoskeleton and shortening of lifespan, whereas mild pat-4 knockdown has a mild effect on cytoskeleton detachment and increases lifespan ³. Consistent with our screening hits and proteomics data are cytoskeleton remodelers implicated in longevity by our and other groups (Figure 6F) ⁴⁻⁷. This points towards a hemidesmosome-to-integrin-to-cytoskeleton remodeling axis to mediate downstream mechanotransduction and organismal longevity.

Implications of collagen overexpression based on our proposed biomechanical model:

Given this line of argument, then why would a multi-copy transgenic overexpression or CRISPR-cas9d activation of COL-120 (doi: 10.17912/micropub.biology.000730) increase lifespan? In the case of multi-copy transgenic, thousands of col-120 genes are present and accessible to transcription factors, but without a signal that leads to gene activation, nothing will happen. Only when there is an activation, then the signal is many-fold stronger, leading to an overproduction of collagen COL-120.

An overexpression should lead to longevity if mechanisms independent of hemidesmosome are at work. However, our genetic evidence indicates that, for some reason, we do not yet fully understand the sensor (i.e., ceHD) or why the mechanical signal is required for the biological output, in this case, longevity. This idea of a mechanotransductive-licensing signal is in line with findings of oligodendrocyte progenitor cells where the mechanical signal overrides physiological signals to maintain OPC activity during aging (<https://doi.org/10.1038/s41586-019-1484-9>).

In summary, our data suggest a biomechanical model linking ceHD with two distant ECMs to regulate physiological outcomes important for promoting longevity. We hope by providing this extensive explanation in the Supplement, interested readers might find the explanation for this novel concept and model explaining the seemingly contradictory results. Clearly, more work is needed to dissect out details and intricacies and update this model in future work (which is beyond the scope of this paper).

2. Another potential contradiction is that on one hand, mildly higher pressure (tension increase in the hypodermis) promotes col-144 expression (Figure 7L). On the other hand, RNAi that causes muscle dysfunction and animal paralysis (pat-2, pat-3, unc-112, etc.) also promotes

col-144 expression (Figure 4B). And it is generally believed that loss of muscle contraction decrease tension in the hypodermis (Nature, 2011; Development, 2019). Some discussion is needed to sort out this problem. For example, could there be a dose-dependent effect of hypodermal tension on collagen expression? Or is it possible that apical versus basal hypodermal tension changes have opposite effects on YAP activity and collagen expression.

Response: Our proposed feedback mechanism predicts that mild perturbations of these muscle integrin signaling genes weaken the force transduction from the muscle to pull on the cuticle, which in turn increases collagen expression to synchronize/adapt muscle strength to cuticle resistance (like our muscles pull on the bones for movement). This enhancement of cuticle resistance/strength, in turn, increases lifespan. By contrast, a severe perturbation that would uncouple the mechanical linkage would abolish mechanotransduction and longevity. And because RNAi penetrance leads to a reduction of function but not complete loss of gene function, it is likely that the above-listed integrin signaling and cytoskeleton genes (“pat-2, pat-3, unc-112, pat-4, tln-1, pxl-1, deb-1, pat-10”) when completely knocked out would also block longevity. Null alleles of most of these genes are embryonic lethal. To test the effects of severe perturbation of these genes on lifespan, we used heterozygous ILK/pat-4 mutants and found that they were shorter-lived and blocked the daf-2-longevity effect (Supplementary Figure 10D, Supplementary Table 7). This result is consistent with the observation that a severe pat-4 knockdown leads to detachment of the cytoskeleton and a shortening of lifespan, whereas a mild pat-4 knockdown has a mild effect on cytoskeleton detachment and increases lifespan (Nishimura et al., 2014). This suggests that the force from the muscle adapts the resistance strength of the exoskeleton (cuticle).

Regarding the relationship between muscle contraction, hypodermal tension, and collagen expression. The potential impact of hypodermal tension on collagen expression and its underlying mechanisms are indeed intriguing.

To address the first point raised by the reviewer, it is possible that there is a dose-dependent effect of hypodermal tension on collagen expression. While previous studies have suggested that loss of muscle contraction decreases tension in the hypodermis (Nature, 2011; Development, 2019), the precise relationship between tension and collagen expression requires further investigation. It would be valuable to explore whether different levels or durations of hypodermal tension influence collagen expression in a graded manner. This could be achieved through experimental manipulation of tension levels and subsequent analysis of collagen expression patterns. However, we are unclear how this would be experimentally possible without affecting ceHD.

Regarding the second point, the possibility that apical versus basal hypodermal tension changes have opposite effects on YAP activity and collagen expression is indeed an intriguing hypothesis. It is known that YAP (Yes-associated protein) is a mechanosensitive transcriptional co-activator that plays a crucial role in translating mechanical signals into gene expression changes. Apical and basal hypodermal tension changes may differentially modulate YAP activity, leading to distinct effects on collagen expression. This hypothesis warrants careful consideration and could be addressed through a combination of genetic and molecular approaches. Experimental manipulation of apical and basal tension levels, followed by

assessment of YAP activity and collagen expression, could shed light on the potential opposite effects of tension changes on collagen expression in different hypodermal regions. In mammalian fibroblasts and muscle cells, αSMA expression upregulates cell contractility and ultimately the expression of a wide range of ECM molecules, including collagens, suggesting a new research avenue for future investigations.

Taken together, as described in (Nature, 2011; Development, 2019) the ceHD acts as a mechanosensor that most likely will serve as the readout of the mechanical decrease of the hypodermal tension. Unclear, without further experiments, is whether a decrease in hypodermal tension can act independently of ceHD. This warrants further investigations. We mentioned the internal (hypodermal) pressure in the Supplementary discussion file as a mechanical input stimulus.

3. The previous publication by Ewald et al., Nature 2015 is closely related to this study. It should be cited and its content compared with these new findings. For example, some longevity intervention and collagen overexpression data included in this manuscript were already reported in the 2015 paper and therefore should be acknowledged. The authors also need to discuss how much does the SKN-1-mediated process contribute to longevity-regulated collagen remodeling, and whether there are any crosstalks/interactions between SKN-1 and this newly discovered feedback loop .

Response: As suggested, we incorporated our previous Ewald et al., Nature 2015 paper in the introduction to clarify that this observation was made; however, the underlying mechanism was not previously investigated (which we did in this study). We also specifically highlighted when the same collagens were used in the results section. Please, see below.

As indicated in our previous paper Ewald et al Nature 2015, the transcription factor SKN-1 does not directly bind these collagen gene promoters to induce them. SKN-1 and other transcription factors are required for longevity; thus, eliminating these transcription factors leads to a loss of the “longevity” signal and eliminates the prolonged collagen expression. It is possible that the ECM homeostasis is downstream of these longevity-signaling pathways, as we find in our screen.

Referral to Supplemental text for interested readers:

“We identified several required genes for longevity, such as daf-16/FOXO, pqm-1, and xbp-1/XBP1 (Supplementary Table 10). We also identified known longevity-promoting genes that, when knocked down, prolonged Pcol-144::GFP expression (e.g., let-363/mTOR, sams-1/MAT1, pat-4/ILK1, Supplementary Table 10). We included a full discussion of the screen, including the sentence above, in Supplementary Table 10. We also indicated this in the results text, referring the reader to it. “(Figure 4A, Supplementary Figure 7A, 7B, see Supplementary Table 10 for detailed screen description, choice of mutants, and validating results)”

Introduction:

*“Using *C. elegans*, we have previously shown that the effects of all so-far tested longevity interventions are abolished when non-essential collagens col-10, col-13, or col-120, are knocked down during adulthood (Ewald et al., Nature 2015). This suggests that these three collagens act downstream of diverse longevity interventions or function as a licensing signal for longevity. Conversely, overexpressing any of these three collagens is sufficient to increase *C. elegans*' lifespan (Ewald et al., Nature 2015). However, why these collagens are important, and the underlying mechanism linking collagen remodeling to longevity remained unknown. “*

Results:

*“As previously reported, overexpressing COL-120 extends *C. elegans*' lifespan (Ewald 2015), and we show that on dead bacteria COL-120 overexpression in the observer-unbiased lifespan machine still increases lifespan to the full amount (Figure 3G, Supplementary Table 7), excluding the idea that higher cuticular collagen levels would extend lifespan by improving barrier function. Furthermore, inducing endogenous col-120 expression post-development was sufficient to increase lifespan (Goyala 2023). As previously reported, not all collagens, when overexpressed, increase lifespan (Ewald 2015). To expand on this, we assessed several more collagen overexpression and their effects on lifespan. Overexpressing collagens that were not altered by longevity interventions were insufficient to increase lifespan (Supplementary Figure 6, Supplementary Table 7), suggesting the unique properties of longevity-promoting collagens.”*

4. The graphic abstract could benefit from a few modifications, as the current version does not have a good enough presentation of the involvement of the ageing process in this feedback loop.

Response: Good point. Although it is hard to depict aging or age-related changes in one picture, we included longevity interventions.

5. A few rounds of text editing is recommended to remove typing mistakes such as “LET-80S” (line 28, should be LET-805), and “disappeared faster from the ECM faster than...”(line127, one “faster” should be deleted).

Response: We fixed the LET-80S to LET-805 in the graphical abstract and deleted the “faster” in the text.

6. Panel labels should be added to supplementary Figure 8A.

Response: Thank you. We labeled panels in Supplementary Figure 8A with day 1 and day 8.

Reviewer #2 (Remarks to the Author):

Using proteomics and in-vivo reporter systems, Teuscher A.C. et al., identify at least three distinct dynamic ECM composition changes during aging. The authors go on to identify a regulatory feedback loop sensing ECM integrity via a mechano-sensitive mechanisms involving Yap-1, that responds to IIS longevity interventions. I appreciate how much work the authors have put into this manuscript. The work presented is extensive. However, the manuscript suffers from a major problem. The interchangeable use of “diverse” longevity interventions in ECM parameters measured isn't justified and obscures the interpretation of the data.

Response: Good point, and we apologize for forgetting to include the concept behind the interchangeability. In our previous 2015 paper (Ewald et al., Nature 2015), we had shown that no matter which longevity intervention was used, non-essential cuticular collagens, such as col-10, col-12, or col-120, were required for longevity. This suggested that collagen remodeling is either downstream of many longevity interventions or is a licensing signal. We included this concept in the introduction.

Introduction:

“Using C. elegans, we have previously shown that the effects of all so-far tested longevity interventions are abolished when non-essential collagens col-10, col-13, or col-120, are knocked down during adulthood (Ewald et al., Nature 2015). This suggests that these three collagens act downstream of diverse longevity interventions or function as a licensing signal for longevity. Conversely, overexpressing any of these three collagens is sufficient to increase C. elegans' lifespan (Ewald et al., Nature 2015). However, why these collagens are important, and the underlying mechanism linking collagen remodeling to longevity remained unknown. “

Likewise, the interchangeable use of distinct reporters when no justification seems plausible makes it hard to evaluate whether the interpretation of the data is correct.

Response: We agree. The underlying concept was missing in the manuscript to understand and follow the logic of the interchangeability of collagen reporters. We introduce now this concept in the results and also explain why we can use these collagen reporters interchangeably. Furthermore, we also observe these ECM clusters across six different longevity interventions.

“Collagens make up the majority of proteins in the ECM (Tarnutzer 2023). During C. elegans matrix synthesis, collagens forming functionally distinct ECM substructures are temporally co-expressed, suggesting that interacting collagens cluster together (McMahon 2003). Out of the 181 C. elegans collagens 25, we were able to assess the quantitative abundance data for 41 collagens proteins and mRNAs during aging (Supplementary Table 1).”

“To test whether this correlation is associated with prolonged production of the pattern I collagens, we used transgenic animals expressing GFP driven by the collagen col-144 promoter, whose expression gradually declines during aging 21,46. Since functionally interacting collagens temporally cluster together (McMahon 2003), suggesting that they can be

used interchangeably, we chose Pcol-144::GFP transgenic animals because from all pattern I transgenic lines, they had the brightest expression and thereby technically simplified the following assay.”

“Pattern I consists of 21/41 detected collagens for which the mRNA, protein levels, and abundance in the ECM steeply declined during aging (e.g., col-10, col-12, col-13, col-120, col-144; Figure 1J-K, Supplementary Figure 1I-J, Supplementary Table 1-3).”

“Consistent with our proteomics, across six different longevity interventions (dietary restriction, metformin, glp-1, daf-2, isp-1, eat-2) and data sets 34–38, the same signature of an increase of a subset of cuticular collagens (col-) protein levels, collagen-stabilizing and remodeling enzymes (dpy-18, phy-2, bli-, nas-, zmp-) and a decrease of cathepsin (cpl-, cpz-, cpr-) protease levels (Figure 3B, Supplementary Table 9).”

Major efforts need to be made to standardize the manuscript from an experimental point of view for a clearer and consistent message and to gain support for publication from this reviewer.

Response: We thank the reviewer for the helpful feedback.

Points to be addressed.

1. Why constantly refer to longevity interventions for generalization when only two interventions were tested? Please refrain from using generalizations.

Response: We answered this point above. By establishing this concept that many longevity interventions activate collagen remodeling. This phenomenon is observed by many groups in omics data ranging from RNA-seq to proteomics; however, not explained or discussed since the underlying biology was unknown. In our previous paper (Ewald et al., 2015 Nature), we showed that longevity interventions required certain collagens, and overexpression of these collagens was sufficient to increase lifespan. If something is required and sufficient, this indicates a potential mechanism of action. However, the underlying mechanism or biology remained unknown, which we addressed here in this manuscript.

We agree with the reviewer to refrain from using generalizations. Thus, we reworded any “longevity generalizations” where we show results using only one longevity intervention. For instance:

Page 8. “Longevity interventions prolong collagen pattern I expression during aging” to “Reduced Insulin/IGF-1 receptor signaling prolongs collagen pattern I expression during aging”

Page 9. “Age-dependent loss of mechanical tension of stably intercalated pattern II collagens is rescued by longevity interventions” to “Age-dependent loss of mechanical tension of stably intercalated pattern II collagens is rescued by glp-1-induced longevity”

Note: As we had stated in the paper, Coleen Murphy’s lab had shown that daf-2-longevity can counteract cuticle stiffness, and this improvement is lost when col-120 is knocked down (doi:10.1016/j.bpj.2022.01.013). Thus, we left the generalization for longevity interventions for interpreting the results in the bigger context.

Page 12. “Longevity interventions slow the age-dependent accumulation of basement membrane collagens” to “Reduced Insulin/IGF-1 receptor signaling does not counteract the age-dependent accumulation of basement membrane collagens”

Page 25 (Discussion): “We further show that longevity interventions not only require YAP-1 for lifespan extension but also enhance YAP-1 to localize to the hemidesmosomes, perhaps to sensitize mechanical readout from hemidesmosomes.” to We further show that daf-2-longevity not only requires YAP-1 for lifespan extension but also enhances YAP-1 to localize to the hemidesmosomes, perhaps to sensitize mechanical readout from hemidesmosomes.”

2. How is Figure 1K representative of data in Figure 1J? It looks that for Col-120::GFP there’s a clear higher signal at D8 than D1. Also, the expression seems to be mostly localized in the intestine rather than the extracellular matrix. Why is this the case and does this “mis-localization” have any role?

Response: This “higher signal” or “mis-localization” is autofluorescence from the gut granules.

In order to quantify collagen levels incorporated into ECMs, a major challenge to overcome was the continuous increase of autofluorescent age-pigments from the *C. elegans* intestine that masked the fluorescence from GFP signal. We first had to establish a method for separating this autofluorescence from the weak GFP signal coming from ECM proteins tagged with GFP. We were able to separate GFP from autofluorescence with a novel combination of a triple-band filter set (Teuscher and Ewald, 2018).

We clarified this in the figure legends and methods.

Fig. 1K “Note in K, the strong yellow-brownish fluorescence is autofluorescence from gut granules (see Methods for details). COL-120::GFP is in green localized in the cuticular ECM. Details in Supplementary Figure 1, Supplementary Table 1.”

Sup Fig 1 “(C-H) scale bar = 50 μ m, (C,D,G,H) * autofluorescent gut granules are in brown-yellowish. “

Sup Fig 3 (F, H, J): “The green fluorescence intensities (excluding autofluorescence from gut granules) of images...”

Methods:

“For the analysis of our collagen::GFP strains, we used a Python script written by Elisabeth Jongsma and Jeliasko Jeliaskov in ImageJ 76. The code is designed to measure the GFP intensity in C. elegans animals while ignoring the gut autofluorescence. The program takes the area of interest selected from the digital image and compares the intensities for the green and red channels within each pixel. C. elegans autofluorescence appears as yellow in the images, a blend of red and green 75. To remove the autofluorescence without affecting the GFP signal, the red channel intensities are subtracted from the green.”

For example, is this expression in the intestine responsible for its lifespan effects and other proteomic changes observed throughout the manuscript? Tissue specific expression of Col-120::GFP experiments should be performed (e.g lifespans, proteomics).

Response: As mentioned above, this “mis-localization” in the intestine is not COL-120 protein but autofluorescence from gut granules that are lysosomal-like structures that accumulate waste products of fats and proteins (i.e., lipofuscin). Therefore, they are also called age-pigments and were one of the biggest challenges that we had to overcome in order to detect the COL-120::GFP in the cuticular ECM. In our promoter-driven col-120 transgenic lines, col-120 is expressed from the hypodermis. And our translational reporters show it incorporates in the cuticular ECM, preferably in the furrow region (Supplementary Figure 1G and 1H). Furthermore, single-cell sequencing data confirms hypodermal expression of col-120 (seam cells are part of the hypodermis).

Figure from Roux et al., doi: <https://doi.org/10.1101/2022.06.15.496201> (<https://c.elegans.aging.atlas.research.calicolabs.com>).

Expression of cuticular collagens across tissues
 Scaled expression above 0.05 is displayed (full range 0 – 10).

Figure: Re-analyzed data from single-cell RNA-seq (Data from DOI: 10.1126/science.aam8940)

Tissue-specific proteomics: We looked through two published datasets of tissue-specific proteomics. COL-120 was not detected, but other cuticular collagens and mostly they were found expressed from the hypodermis (epidermis). Thus, we speculate, and given our reporter strains and the scRNAseq data, that *col-120* is mainly expressed from the hypodermis.

Figure. Matrisome proteins detected by in-vivo tissue-specific proteomics by Reinke et al., 2017 (DOI: 10.1126/sciadv.1602426)

Figure. Matrisome proteins detected by tissue-specific protein purification and proteomics by Waijers et al., 2016 (DOI 10.1186/s12915-016-0286-x).

Tissue-specific lifespan: Although we did not perform classical tissue-specific lifespans by generating novel transgenic lines using intestinal vs hypodermal promoters to drive the expression of COL-120::GFP, we confirmed higher levels of endogenous COL-120 is sufficient to increase lifespan from adulthood (doi: 10.17912/micropub.biology.000730). We used CRISPR-induced expression for inducing the endogenous expression of col-120 (and also col-10; doi: 10.17912/micropub.biology.000730), excluding the possibility that the GFP tag could interfere and might lead to mislocalization or drive the lifespan effects. We included a statement about endogenous col-120 expression being sufficient to increase lifespan.

“Furthermore, inducing endogenous col-120 expression post-development was sufficient to increase lifespan (doi: 10.17912/micropub.biology.000730).”

In summary, the mentioned intestinal fluorescence is autofluorescence from gut granules. COL-120::GFP or any other collagen-tagged GFP is not mislocalized in the intestine. We carefully dissected this by generating a new triple-set filter set to distinguish autofluorescence from GFP signal (Teuscher and Ewald, BioProtocol 2018) and shown this in Supplementary Figure 1 and Supplementary Table 1. Thus, proteomics and lifespan effects from COL-120::GFP can not stem from the mislocalization of COL-120 in the intestine. We apologize for not clearly labeling this yellow-brownish signal as autofluorescence and hope that now indicating it in the figure legends and more details in the Method section would not mislead any reader.

In Figure 1 J and K, and Supplementary Figure 1 J and K, the differences in col-120 mRNA and especially COL-120 protein levels across different timepoints do not correspond entirely (e.g., the levels of COL-120::GFP signal drop much more drastically in the Figure 1 J than in the corresponding table in Supplementary Figure 1 J) – how can this be explained? Were the same reporters used for both figures? Also, how did the authors assign the numerical values to expression levels in the Supplementary Figure 1?

Response: In Fig 1J, we show the overall model of col-120 mRNA levels and COL-120 levels incorporated into the ECM (cuticle) as a time course. Fig 1K shows representative images of the COL-120 levels incorporated into the ECM at day 1 and day 8 of adulthood, which are “semi-quantified” by visible scoring in numerical values from 0-3 in Supplement Figure 1J (BTW, there is no Supplementary Figure 1J and K; we assume Supplementary Figure 1I and 1J were meant). The observed fluorescence scoring is described in the figure legends (listed below). As stated in the Methods section: “The fluorescence of the animals was graded on a scale from 0 to 3 intensity. Intensity 3 indicates the highest fluorescence observed. Relative to the highest observed fluorescence of a given reporter line, a gradient scale in 0.5 intervals was categorized and scored, with 0 indicating no fluorescence above the background.”

The difference between Supplementary Figure 1I and 1J is that in 1I promoter drives the GFP (transcriptional reporters that is an approx. for mRNA level induction (not degradation since GFP levels are more stable for 48 hours than usually the endogenous mRNA)), whereas in 1J the GFP is fused with the collagen protein (i.e., translational fusion or Translation Reporter). Please, also see Supplementary Figure 1G for transcriptional and Supplementary Figure 1H for translational expression of col-120 side by side. As indicated in Supplementary Table 1, the same reporter strains were used (see strain names), and more images are included there. To come up with the time course model in Fig 1J, we incorporated all mRNA (RNAseq from Fig 1A, Supplement Figure 2, Supplementary Table 2), transcriptional reporter data, and translation reporters focusing on the ECM (Supplementary Figure 1 and Supplementary Table 1). Comparing Figure 1J with Supplementary Figure 1J (data for day 1 and 8), about the same drop is represented. Also, a higher resolution difference of COL-120 drop from the cuticle can be observed in Figure 2A.

Fig 1J and 1K

Supplementary Figure 1 partial

Figure 2A. COL-120::GFP day 1 vs day 8

Figure legends in the manuscript:

Supplementary Figure 1

“(I) Transcriptional reporters are driven by matrisome genes during development and aging. For details, see Supplementary Table 1. The fluorescent scale corresponds to the highest observed

fluorescence of a reporter line (intensity 3) graded to no observed fluorescence above the background (intensity 0).

(J) Translational reporters of matrisome and adhesome proteins are localized and incorporated into ECM structures during development and aging. For details, see Supplementary Table 1. * indicates CRISPR-Cas9 genome inserted tag in the endogenous gene locus. “

3. “While COL-120 protein tagged with GFP gradually disappeared from the cuticular ECM during ageing, slowing ageing by rIIS showed COL-120 in the ECM for a prolonged time (Figure 2A, 2B)”.

I agree with this but isn't it to be expected?

Response: It could be expected but has not been previously experimentally shown. For instance, for the collagen type IV, EMB-9, the mRNA slightly declines during aging (Fig 1A), but the EMB-9::cherry increases in the basement membrane during aging, which is not affected by rIIS (Supplementary Figure 3K, 3L). Furthermore, RNA levels rarely tightly correlate with protein levels (Liu et al., 2016; <https://doi.org/10.1016/j.cell.2016.03.014>), and the protein/mRNA abundance ratio is further complicated for proteins that form multicomplex structures, such as the ECM, where proteins are post-translationally processed, modified, secreted, incorporated, and crosslinked to form the matrix.

More important is whether this change is proportional to their lifespan extension. In other words, is there any gain in healthspan.

Response: Interesting point with gain in healthspan. We have not addressed that, but indirectly, we would speculate yes.

We show that worms that express more col-144 promoter-driven GFP live longer (Figure 3F). We had previously employed this observation to use col-144 promoter-driven GFP intensity levels at day 5 of adulthood as a surrogate marker for longevity induced by drugs (<https://doi.org/10.1111/accel.13441>). As col-144 and col-120 are in the same pattern I (as explained above), this is also transferable to col-120, suggesting that this col-120 change is proportional to their expected lifespan extension. The next step to nail this would be to use microfluidics or the Copas Worm sorter to individually quantify col-120 promoter-driven GFP of individual worms and lifespan them in single plates.

*How would this translate to healthspan? Nick Stroustrup had shown that the lifespan of *C. elegans* temporally scales (Stroustrup et al., 2016 Nature doi: 10.1038/nature16550). We have previously shown that healthspan temporally scales with longevity interventions (<https://doi.org/10.1016/j.isci.2022.103983>). Thus, we speculate it is likely. Furthermore, our proteomics data of processes downstream of COL-120 overexpression indicated that some of the downstream longevity-promoting processes are involved in stress responses. As we state in the text:*

“We found abundance changes in proteins governing the cytoskeleton dynamics (tbc1-1/Tubulin-specific chaperone B, pat-6/parvin, ifd-2/intermediate filament), as well as proteins involved in pathogen and oxidative stress response, and metabolism, respectively (Supplementary Table 9).”

This might suggest a gain in healthspan for stress and resilience-related phenotypes.

Likewise, similar measurements should be performed for the Dendra reporter.

Response: We had a collaboration with Prof. Zach Pincus on measuring the COL-120::Dendra reporter; however, the fluorescence of our reporter was too weak for his system to detect. We were building a transcriptional col-120 Dendra reporter; however, in the meanwhile, Prof. Pincus left academia. We are thus not able to perform these recommended experiments.

Why was daf-2 RNAi used? Are these effects similar in daf-2 worms of different mutant alleles (class I vs class II)?

Response: Reducing Insulin/IGF-1 receptor signaling of daf-2 can extend lifespan at least by two separate modes of action: a dauer-dependent and dauer-independent program. Using class II mutants, improper dauer phenotypes like lethargy mask the underlying health of the worms. Please, see Ewald et al., 2017 (Untangling Longevity, Dauer, and Healthspan in Caenorhabditis elegans Insulin/IGF-1-Signalling. <https://doi.org/10.1159/000480504>) for detailed explanations. Importantly, daf-2RNAi has the same propensity to increase lifespan without eliciting improper dauer-related phenotypes. In addition, we can post-developmentally reduce Insulin/IGF-1 receptor signaling with daf-2(RNAi).

More importantly, is COL-120 a good example to assess this question, considering that it disappears from cuticle quite rapidly, hence the Dendra conversion and follow-up measurements were done in worms that are only Day 2 and Day 4 adults, respectively?

Response: As shown in Figure 2A, on day 8, COL-120 levels in the cuticle are almost completely gone. And this is consistent with pattern I collagens that we wanted to investigate here. So from an experimental standpoint, Days 2 and 4 were chosen. BTW, this was a heroic experiment by the PhD students, first to convert 2 areas of worms on day 2, recover the worm, put it back into the incubator to culture for 2 more days, then harvest the same worm, find the proper spot (as you can imagine, when you anesthetize worms, you have little control on which side they lay on your slide, and your side of interest might be on the bottom where you can not image) and then image again for a whole day on the confocal.

The importance here is that we were able to consider the rate of disappearance of COL-120 in our calculations and show that although COL-120 disappears, new COL-120 is added on top but at a much slower rate on day 4. Interestingly, daf-2 RNAi prolonged the addition of new COL-120, similar to the prolonged mRNA expression. So, in this case here, surprisingly, prolonged mRNA expression correlates with prolonged new COL-120 added to the ECM. And this was the basis for our reporter screen with prolonged expression of col-144 to identify underlying molecular mechanisms driving this prolonged-expression or turnover.

4. “We found a stark increase in FRET transmission during old age (day 12) compared to young (day 2), and this age-dependent increase in FRET transmission was lower in long-lived glp-1 mutants at day 8 of adulthood compared to normal-lived wild-type controls (Figure 2H-J, Supplementary Table 4). This suggests that longevity interventions may counteract age dependent crosslinking of pattern II collagens.”

I don't understand why glp-1 mutants are being used here and not daf-2 RNAi for consistent reporting and mechanistic understanding. Why are distinct longevity interventions being

clustered as one? These interventions have specific mechanisms involved and should be treated independently. I do not agree with a wide generalization as the one being made by the authors. As before, for figure 2I and 2J, measurements should be done in proportion to their respective longevity since improvements in chronological age are expected.

Response: Please see our response above in your summary paragraph about the generalization of longevity interventions. Also to add is the concept from Stroustrup et al., Nature 2016 that longevity interventions temporally scale, suggesting that interventions are exchangeable in principle for C. elegans aging processes. Of course, as the reviewer pointed out, different longevity interventions use different signaling pathways, but they have several mechanisms of action in common that act systemically or downstream of their signaling pathway, such as improved stress resilience and better protein homeostasis. For some reason, ECM remodeling is a far downstream process, and we show it acts via mechanotransduction and probably as a licensing signal for longevity via the discovered feedback mechanism (See new graphical abstract and proposed model in Fig 7P. Also, for the explanation of this model see the answer to point one of reviewer 1 or Supplementary Text provided in the resubmission).

As an independent support, I wanted to point out (as mentioned above) that Coleen Murphy's lab had shown that daf-2-longevity can counteract cuticle stiffness, and this improvement is lost when col-120 is knocked down (doi:10.1016/j.bpj.2022.01.013). Thus, the interchangeability of longevity interventions.

Why glp-1? This was a cost-saving decision since FUDR is expensive and can interfere with testing other drugs. As shown in Supplementary Figure 4 I-N, we used chemicals that either accelerate or slow the accumulation of advanced glycation end products (AGEs) in the temperature sterile spe-9 wild-type background.

Fig 2I and 2J: We chose these time points based on the respective lifespan curves shown in Fig 2G. Day eight is right before death starts to occur, and beyond that time point, several stochastic deteriorative processes became visible (doi: 10.1038/nature01135).

As we wrote in the manuscript:

"We scored FRET ratios when animals were young (day 2 of adulthood), old but before death events occurred (day 8), and at very old age, when about 75% of the population had died (day 12; Figure 2G, Supplementary Figure 4C-F)."

The idea of the reviewer to adjust by "biological age" in principle is a great one, however, it in practice is unfortunately not feasible. At the timepoint where 75% are dead of glp-1 (i.e., day 16), all the normal-lived wild types had died (the last survivor was on day 12). Also, it is a different question. Given that these are AGEs and damages accumulate during aging and longevity interventions work through temporal scaling that correlates with temporal scaling of healthspan or damage repair, one would expect only a difference if glp-1 would directly act on AGE removal rather than indirectly via, for instance, improved proteohomeostasis.

5. When discussing the results depicted in Figure 3B, the authors mention that different longevity interventions counteract the age-related changes in protein abundances (lines 210-212). Nevertheless, the graph depicts only the changes for Day 1/Day 2, or even L4 worms,

except in the case of eat-2 mutants, where the data for Day 15 are shown. What is the reasoning behind showing all data in one graph given that the abundance of each protein is regulated in an age-dependent manner with distinct expression patterns with age? Ideally, one would show data for proteome changes in older worms across all these longevity interventions. As such, this panel does not sufficiently support the respective claims being made. Since I do not think this is fair to ask for such experiment to be done, I would ask the authors to consider focusing on the longevity interventions that this work primarily focuses on (daf-2 and glp-1) and perform proteomics experiments in late life and at mean/median lifespan for each intervention compared to control.

Response: Fig 3B is a comparison that supports our longitudinal data shown in Fig 3A using daf-2(RNAi). Fig 3B also supports the point that all these 6 different longevity interventions (dietary restriction, metformin, glp-1, daf-2, isp-1, eat-2), although each dataset is patchy and overlap is not perfect, the same main ECM proteins are changed by longevity interventions, broadening our observation from Fig.3A. As mentioned, these 6 longevity intervention data sets are from previously published data (as cited in manuscript and all detailed in Supplementary Table 9). Now most of these interventions are mutations, and the signaling is already different during development compared to wild type. One can already observe this in fig 3D showing that glp-1 mutants have higher total collagen levels than wild-type control on day 0 (=L4). In fact, for instance, the highest difference of wild type compared to daf-2 is seen at L4 for stress resilience and downstream signaling output. Furthermore, protein homeostasis collapses at 8 hours into the first day of adulthood in C. elegans (Please, see Figure 1 of <https://www.ncbi.nlm.nih.gov/pmc/articles/PMC3914504/> which illustrates this best). Thus, although in a perfect world, one would do all these longitudinal time courses of wild type vs longevity interventions, we already can extrapolate from current data that at least for the ECM changes relative to its time point, it is fairly consistently the same major ECM players.

In the manuscript describing these results, we make no statements that Fig 3B are age-related changes but extrapolate in the summary sentence taken together our proteomics time course in the context of previously published data.

6. “during ageing, and that longevity intervention started with more collagen mass which declined at a similar rate during ageing (Figure 3C, 3D).”

This goes with my previous comments that improvements seem to occur early in life but do not seem to improve aging rate and makes one wonder about the potential implications of such findings and their general interest to the aging field.

Response: Exactly. In the aging field, discoveries that are made with longevity mutants are confirmed by interventions that are elicited post-development. Such as using daf-2 RNAi starting from post-development. In my postdoctoral work, I already had shown that adult-specific daf-2 RNAi is sufficient to increase total collagen levels measured at day 8 of adulthood and also increase lifespan. See Fig. 4d Ewald et al., 2015.

Furthermore, we used CRISPR-induced expression for inducing the endogenous expression of col-120 (and also col-10; doi: 10.17912/micropub.biology.000730), excluding the possibility of spilling over from developmental effects on lifespan.

“Furthermore, inducing endogenous col-120 expression post-development was sufficient to increase lifespan (doi: 10.17912/micropub.biology.000730).”

We agree with the reviewer that we should clarify this point in the manuscript. Thus, we rewrote it to avoid generalization and incorporate the adult-specific point.

“We found that one-fifth of the total collagen mass normalized to total protein mass was lost during aging (Figure 3C). The longevity-promoting glp-1 mutants started with more collagen mass which declined at a similar rate during aging compared to wild type (Figure 3D). Adult-specific daf-2 RNAi is sufficient to promote higher collagen levels at day 8 of adulthood (Ewald 2015), suggesting that longevity interventions slow the loss of collagen mass during aging. “

Likewise, In line 188, the subtitle does not correspond to the text in lines 194-195 (i.e., “independent of daf-2 RNAi longevity interventions”) – the authors should clarify this discrepancy. Do longevity interventions slow or not the accumulation of basement membrane collagens?

Response: We apologize for the incorrect heading. Correct, it is independent. We changed it to. “Reduced Insulin/IGF-1 receptor signaling does not counteract the age-dependent accumulation of basement membrane collagens”

Also, most data seems to suggest a delay rather than a slowing down. In the same paragraph, the authors mention that the observation made is consistent with thickening of the human basement membranes with age. However, similarly as in the point 3, the photoconversion and quantification were done at Days 2 and 4, respectively, therefore, not in worms aged enough to describe this as an aging/anti-aging effect.

Response: As mentioned above, the aging of the protein homeostasis starts early in adulthood due to the protease collapse described by Morimoto’s lab (<https://www.ncbi.nlm.nih.gov/pmc/articles/PMC3914504/>). Also, as mentioned above, it is experimentally not feasible to go later with pattern 1 collagens (COL-120::Dendra), since it is excised out to the ECM and almost completely gone at day 8 of adult.

*The reviewer raises a good point that it seems to delay rather than slow down. This is in line with temporal scaling underlying all longevity interventions in *C. elegans* (Stroustrup et al., 2016 Nature). Since it seems these are more philosophical questions on what entails aging, I recommend seeing Prof. Uri Alon’s work explaining Gompertz law and mortality rate (<https://youtu.be/khgLk9IzFOM>). Also, at least from epigenetic clocks, we know that aging starts after embryogenesis (DOI: 10.1126/sciadv.abg6082). Thus, it is a time point of reference for what one defines as aging.*

Excitingly, we had previously shown that degrading the DAF-2 receptor at the end of life was sufficient to double the lifespan of these geriatric worms

(<https://doi.org/10.7554/eLife.71335>), suggesting either here all these damages had occurred, but it was still possible to live longer. So, in that case, it is unlikely that longevity is simply due to a delay in the onset of these pathologies.

Nevertheless, we tried in the manuscript to avoid the term aging itself in results gathered from early adulthood but extrapolated to age-related changes to build concepts incorporating these earlier data points.

7. Also In Figure 3 and the paragraph related to it, the authors argue for existence of a feedback loop that is based on the abundance of a certain collagen (e.g., COL-120, COL-144, COL-10). If the abundances of other matrisome proteins are changed in response to this, and can be modulated via longevity interventions, how do the authors explain that similar changes in levels of these “key collagens” are not present between the longevity models and the worms with COL-120 OE (see panels 3B, 3H, 3K and 3I)? Isn't this the central tenet of this manuscript? Yet, the data seems not to support this convincingly.

Response: We found that this feedback loop is mediated through mechanotransduction, adjusting collagen levels in the cuticle coupled to physical forces (See explanation by reviewer 1 point 1). In Fig. 3I, overexpression of col-120 leads to upregulation of col-10, col-13, and col-144. Why it is not the same level as col-120 is either because we used a multi-copy array or this is the right collagen level to sync the cuticle stiffness to muscular forces. All these collagens individually are detected in the proteomics data in panels 3B, 3H, 3K and 3I but not all at the same time. The reason is that proteomics of extracellular matrices is challenging. Since it is a matrix and insoluble, these collagens have to be solubilized, which we achieved with high Urea concentration (8M) and other methodological adjustments (see Methods for details). So proteomics of ECM will always lead to patchy and imperfect data, and thus, if collagen is not detected, one can not say it is absent.

Our data here is supported by 2 independent approaches. First, we used an RNAi screen to identify key regulators of prolonged collagen expression and found the hemidesmosome (also, you can see components of these hemidesmosomes in the proteomics data, such as unc-52). Second, to experimentally address the observation that some genes would induce collagen expression and longevity but others would block enhanced collagen expression and still are required for longevity, we have performed lifespan analyses of over 39 matrisome mutants (including hemidesmosome mutants) of more than 55'499 animals to compare the results to the Pcol-144::GFP expression screen.

8. “To test whether this correlation is associated with prolonged production of the pattern I collagens, we used transgenic animals expressing GFP driven by the collagen col-144 promoter, whose expression gradually declines during aging 5,31” .

The same experiments done here for col-120 should be done for col-19 and emb-9. Including splitting animals based on GFP levels of the reporters (col-19 and emb-9), and rnaï (col-19 and emb-9) and OE studies (except for emb-9 which has already been done), for lifespan. If there is any generalization possible in the mechanisms being proposed, and the rationale employed by the authors, why did the emb-9 oe did not lead to a shortened lifespan? This is a critical point, since the classification of the three patterns proposed in figure 1 may be mechanistically

meaningless without concrete proof that such classification is linked to a biological effect (e.g. lifespan), which is what the authors are trying to imply in this manuscript.

Response: In contrast to col-120 or pattern 1 collagen, both col-19 or emb-9 expressions are changed by longevity interventions. How to split P_{emb-9}::GFP animals when the fluorescence stays similar from day 1 to day 5 or 8 (Supplementary Figure 1I). More importantly, is the collagen that is incorporated in the ECM. Thus, in Supplementary Figure 3, we demonstrate that daf-2(RNAi) does not alter COL-19::GFP nor EMB-9::mCherry levels in the cuticle during aging.

To gain mechanistically meaningful proof, we showed that overexpression of COL-19 and EMB-9 is not sufficient for longevity. As mentioned, we performed lifespans of more than 55 thousand animals to dissect which ECM proteins are required for longevity. We summarized this in Figure 6F. Please, see all the lifespan data listed in Supplementary Table 7.

9. “Furthermore, overexpressing collagens that were not altered by longevity interventions was insufficient to increase lifespan (Supplementary Figure 6, Supplementary Table 7), suggesting the unique properties of longevity-promoting collagens. To identify downstream mechanisms mobilized by collagen overexpression”.

How are these two statements compatible? On one hand the authors state that no generalization can be made and then use one specific intervention to find “generalized mechanisms” of collagen overexpression! This is unfortunately, very confusing and unconvincing.

Response: We apologize for the confusion and rephrased this paragraph for clarity.

“As previously reported, overexpressing COL-120 extends C. elegans' lifespan (ref 24), and we show that on dead bacteria COL-120 overexpression in the observer-unbiased lifespan machine still increases lifespan to the full amount (Figure 3G, Supplementary Table 7), excluding the idea that higher cuticular collagen levels would extend lifespan by improving barrier function. Furthermore, inducing endogenous col-120 expression post-development was sufficient to increase lifespan (ref 47). As previously reported, not all collagens, when overexpressed, increase lifespan (ref 24). To expand on this, we overexpressed several more collagen and assessed their effects on lifespan. Overexpressing collagens that were not altered by longevity interventions were insufficient to increase lifespan (Supplementary Figure 6, Supplementary Table 7), suggesting the unique properties of longevity-promoting collagens.”

10. Why were daf-2 mutants used for J and K and RNAi for LM. Given the different longevity curves from these two interventions, how much does this affect the interpretation of the results? Do the authors observe the same results if a daf-2 mutant is used instead of the mutant for 3L and 3M?

Response: These daf-2(e1368) are class I alleles, not improbable inducing dauer-related phenotypes (Untangling Longevity, Dauer, and Healthspan in Caenorhabditis elegans Insulin/IGF-1-Signalling. <https://doi.org/10.1159/000480504>). The relative propensity for these daf-2(e1368) mutants or daf-2(RNAi) to increase lifespan is the same. As found by many other labs (<https://doi.org/10.1159/000480504>). The choice here is to show that irrespective of RNAi (that can have off-target effects) or of mutants (that affect animals from birth and sometimes can

have background mutations partly affecting lifespan 10.1038/nature10296) the requirements of a proper collagen feedback loop are important. Using both approaches establishes confidence in the underlying results.

11. “Since the decline in the pattern I collagen transcription preceded its decline or remodelling out from the cuticle, we scored col-144 promoter driven GFP levels (Pcol-144::GFP) that progressively declined from day 1 to day 8 of adulthood (Figure 4A, Supplementary Figure 7A, 7B)”

Same issue as before, why wasn't the col-120::GFP promoter used when all the work has been done so far with this gene? When the authors themselves state that each collagen has a unique properties, without additional data showing that col-144 and col-120 have exactly the same biological properties, I am not convinced about the value of this screen in the context of the data up until this point.

Response: We have performed a small selected assessment with col-120 for hemidesmosome and other screen hits and scored the intensity decline. These experiments confirmed the results of the col-144 expression and are shown in Supplementary Table 10 (tab: ScreenValidation col-120).

10X

REVIEWER COMMENTS

Reviewer #1 (Remarks to the Author):

The authors have addressed all my concerns and presented a much improved study. I recommend acceptance.

Reviewer #2 (Remarks to the Author):

I find the justifications provided by the authors to my queries mostly satisfactory. However, some key points remain unsatisfactory.

1. The constant referring to the Soustrup et al. manuscript about temporal scaling as a justification to support the authors results leaves a lot to desire given the many experimental flaws and inconsistencies this work has on its subsequent message. In particular, I find the lack of engagement by the authors to provide additional experimental evidence in regards to healthspan improvements through the mechanisms described here detracts from the potential interest of the manuscript to the general audience. Even more so disappointing, when this is an active area of investigation conducted by these authors. It is therefore within their ability to provide such data and not an unreasonable request.

2. Likewise, the justifications for the lack of additional proteomics data are not sufficient. In particular for points 5 and 7. If the authors admit themselves that the proteomics approach isn't "reliable" then how can we convince others that the mechanism proposed is correct when so many over-reaching generalisations have been made throughout the study?! Further experimental work is necessary to convince this reviewer of this key aspect of the work.

Reviewer #3 (Remarks to the Author):

This manuscript from Teuscher et al. describes the finding that how distinct ECM composition altered with age in *C. elegans* by using different approaches including omics and in-vivo reporter systems. The authors identified a regulatory feedback loop modulating ECM remodeling via mechano-sensitive hemidesmosomes to signal into the cells via YAP-1 to regulate ECM components to enhance the mechanical properties of the surrounding matrices.

The findings are very interesting and provide a valuable contribution to the field of aging. Furthermore, the experiments are well performed overall to make a solid conclusion and the manuscript is also well

written. Some important missing points have already been pointed out by other reviewers. In addition to them, I have some minor recommendations to strengthen the manuscript even further before publication.

1) The authors have shown that there is a decline, unchanged or increase in the protein levels of the majority of cuticular collagen with age in Figures 1B, E and H respectively. Although the comparison between early adulthood and day 8 leads to this conclusion, it seems that some of the cuticular collagen show different trends in their protein level during worms' reproductive stage with age (d0 to d4). The authors should discuss the possible explanation of this phenomenon in discussion. (They provide some information about the extension of the cuticular exoskeleton during early adulthood in line 291-292 but more detailed explanations are required for Figure 1 B,E and H).

2) The presentation of transcriptional changes in supplementary figures 2 c and f is not easy to read. Although I appreciated that the authors are trying to give an overview of all data, it can be ideal to add additional heat map analysis at least for important candidates which were highlighted.

3) In Figure 3 B, the authors suggested that longevity interventions increased the normally age-related decrease of collagen levels and dampened the normally age-related elevation of extracellular proteases. The conclusion is not very solid due to early adulthood conditions being analyzed in different data sets (only eat-2 (ad116) day 15 is really supporting this conclusion). Of course, it is not feasible to do new proteomics experiments with all different longevity paradigm models. Therefore, optionally, the authors can also use different proteomics data sets for some of the longevity interventions during aging in the literature. If it is not possible to determine with other proteomics data sets, it would be more accurate to state that longevity interventions may affect collagen protein levels in early adulthood, but it would not be appropriate to attribute this to age-related regulation. Moreover, it is not ideal to show the protein abundance (fold change) from different proteomics datasets in the same graph which can lead to wrong conclusions. I recommend the authors show the data separately for each proteomics data set. They can use different heat maps or volcano plots for each proteomics data set.

4) In Supplementary Table 9, the authors provided a data set that shows the changes enhancing ECM composition that are specific to COL-120 overexpression by comparing different conditions. It is important for the authors to include a comparison between Col-120oe EV and WT EV in the data set, as this would indicate the overexpression effects of COL-120 when compared to the endogenous level of COL-120 expression. This comparison in data set is currently missing and should be provided.

5) The authors performed a quantitative proteomics analysis for comparisons wild type with a P4H dpy-18 (ok162) loss of function mutant treated either with control or daf-2 (RNAi) to analyze the effect of daf-2 depletion on changes in protein abundance ratios. They called this proteomics analysis 'ECM-enriched proteomics'. However, the term "ECM-enriched proteomic" is not accurate and should be corrected in the manuscript. This is because the study was actually a quantitative whole proteome analysis, not an enrichment or pull-down experiment followed by quantitative proteomics.

6) If I don't miss, the authors didn't provide raw data for their proteomics analysis. They just provide output data in a supplementary table. Proteomics data should be deposited in the ProteomeXchange Consortium via the PRIDE partner repository.

7) Like Figure 3B, in supplementary Figure 8b, the data from two different proteomics data sets was shown in the same graph. It is crucial to change the figures that show a combination of different proteomics data sets in the same graph. This is because the protein abundance and fold change from different data sets cannot be compared, and it could be very misleading to the audience.

REVIEWER COMMENTS

Reviewer #1 (Remarks to the Author):

The authors have addressed all my concerns and presented a much improved study. I recommend acceptance.

Response: Thank you.

Reviewer #2 (Remarks to the Author):

I find the justifications provided by the authors to my queries mostly satisfactory. However, some key points remain unsatisfactory.

Response: We thank the reviewer and are happy that most of our previous responses were satisfactory. We addressed below the two remaining points.

1. The constant referring to the Soustrup et al. manuscript about temporal scaling as a justification to support the authors results leaves a lot to desire given the many experimental flaws and inconsistencies this work has on its subsequent message. In particular, I find the lack of engagement by the authors to provide additional experimental evidence in regards to healthspan improvements through the mechanisms described here detracts from the potential interest of the manuscript to the general audience. Even more so disappointing, when this is an active area of investigation conducted by these authors. It is therefore within their ability to provide such data and not an unreasonable request.

Response: We are sorry to hear that the reviewer was not satisfied with our previous response regarding the relationship between life- and health-span. We agree with the importance of healthspan. Part of the reviewer's question has been previously addressed. Furthermore, in the meantime, we have addressed the reviewer's question about the gain in healthspan in another publication (doi: 10.17912/micropub.biology.000730). The reason is that for the induction of collagens, we had to employ a novel CRISPR-dCas system, of which explaining and validating this method was beyond this manuscript. However, we conducted two sets of healthspan assays as requested by the reviewer.

To restate the previous reviewer's question: " 3. "While COL-120 protein tagged with GFP gradually disappeared from the cuticular ECM during ageing, slowing ageing by rIIS showed COL-120 in the ECM for a prolonged time (Figure 2A, 2B)". I agree with this but isn't it to be expected? More important is whether this change is proportional to their lifespan extension. In other words, is there any gain in healthspan."

Here, we are not going to repeat our previous answers but specifically address the "experimental evidence in regards to healthspan improvements through the mechanisms described here".

1) “Slowing ageing by rIIS” increases several aspects of healthspan that were completely abolished by adult-specific col-120 knockdown.

- Pharyngeal pumping rates decline during aging. The higher pharyngeal pumping rates of *daf-2(e1370)* at day 10 of adulthood were reduced by *col-120RNAi* (Ewald et al., Nature 2015). This was also observed by adulthood-specific rapamycin treatment showing higher pharyngeal pumping rates at day 8 of adulthood, whereby adulthood-specific *col-120(RNAi)* abolished this (Ewald et al., Nature 2015), suggesting that for reducing mTOR-mediated longevity, our proposed mechanism is required for healthspan benefits.
- Lipofuscin levels increase during aging. The lower lipofuscin levels of *daf-2(e1370)* at day 10 of adulthood were increased by *col-120RNAi* (Ewald et al., Nature 2015).
- Oxidative stress resilience is increased by rIIS. The increased oxidative stress resistance of *daf-2(e1370)* at day 4 of adulthood was reduced by adulthood-specific *col-120RNAi*, *col-10RNAi*, *col-13RNAi*, *col-167RNAi* but not by *col-97RNAi*, *col-133RNAi*, *col-180RNAi*, *col-127RNAi*, *col-176RNAi* (Ewald et al., Nature 2015). Indicating the specificity of our proposed mechanism and not the general loss of any collagen.
- As control experiments, body length, cuticle leakage, and vulva integrity (*P-vul*) were not affected by the same experimental procedure of *col-120(RNAi)* (including also *col-10RNAi* and *col-13RNAi*) on *daf-2(e1370)* mutants (Ewald et al., Nature 2015).
- Cuticle stiffness increases with age shown by Coleen Murphy lab (Rahimi et al., 2022 Biophys J. DOI: 10.1016/j.bpj.2022.01.013). This stiffness increase is slowed by *daf-2(e1370)* mutants but completely abolished by *col-120RNAi* (Rahimi et al., 2022 Biophys J. DOI: 10.1016/j.bpj.2022.01.013).
- Here, to assess thermotolerance post-reproductive ages, we placed wild type (*N2*), collagen *col-120* deletion mutant (*PHX2339*), and as control mechano-sensitive collagen *mec-5(CB1340)*; which does not block rIIS longevity) on the empty vector (*L4440*) or *daf-2RNAi* and measured heat survival at 32°C at day 8 of adulthood. We found that loss of *col-120* blocked the improved heat resistance of *daf-2(RNAi)*.

Strain	n assayed / n total	Temp.	median	25th per.	75th per.	mean \pm se	mean lifespan change [%]	p-value (logrank)
N2 (wild type), L4440	322 / 322	32°C	0.9	0.7	1.1	0.9 \pm 0	NA	NA
N2 (wild type), daf-2RNAi	363 / 363	32°C	1.1	0.9	1.5	1.2 \pm 0	22	4e-12
PHX2339 (col-120(syb2339)), L4440	120 / 120	32°C	0.9	0.7	1.0	0.9 \pm 0	-4	5e-01
PHX2339 (col-120(syb2339)), daf-2RNAi	99 / 99	32°C	0.8	0.6	0.9	0.8 \pm 0	-14	6e-04
CB1340 (mec-5(e1503)), L4440	43 / 43	32°C	0.9	0.7	1.0	0.9 \pm 0	-1	8e-01
CB1340 (mec-5(e1503)), daf-2(RNAi)	32 / 32	32°C	1.2	1.0	1.3	1.2 \pm 0	28	4e-03

Thus taken together, this suggests that the improvements in healthspan under rIS are not licensed when our proposed mechanism is blocked.

2) Overexpression of collagen increasing lifespan with regards to healthspan

- *COL-43 or COL-80 overexpression extends lifespan and resistance to paraquat (oxidative stress) without changing the permeability of the cuticle. Whereas col-43(RNAi) or col-80(RNAi) makes worms more susceptible to paraquat (Deng et al., 2021 Aging doi.org/10.18632/aging.202406).*
- *Transgenic overexpression of collagen starts when the promoter becomes active. During larval development, both collagens col-10 and col-120 are expressed, with their highest expression occurring at L4, followed by a rapid decrease in expression in young adults. "However, whether post-developmental enhancement of collagen expression could also increase the lifespan and healthspan was unknown. Recently, we described a method to induce the expression of a target gene using catalytically dead Cas9 (dCas9)-engineered C. elegans via ingestion of bacteria expressing a pair of promoter-specific single guide RNAs (sgRNA). Here, we cloned col-120 promoter-specific sgRNA oligo pair into L4440-BioBrick-sgRNA and fed these bacteria to dCas9::VP64 transgenic C. elegans. We observed a similar percentage of lifespan extension by post-developmentally dCas9-induced expression of col-120, as previously reported through transgenic overexpression of col-120. Consistent with this result is that induction of another previously shown longevity-promoting collagen, col-10, also increased lifespan. Furthermore, we found an enhanced resilience to heat stress and increased expression of hsp-16.2 upon dCas9-activated col-120 expression." (adapted from publication Goyala and Ewald 2023 MicroPubl.Biol. doi: 10.17912/micropub.biology.000730). Hence, these results validate that the increased longevity by enhancing col-120 expression is also improving resilience and healthspan marker of thermotolerance."*
- *Although dCas9-activated col-10 and col-120 expression increased lifespan and heat resistance, 2 out of 5 trials col-10 or col-120 induction mildly increased oxidative stress resistance. We conclude that under these conditions, dCas9-activated col-10 and col-120 expression was not sufficient to increase oxidative stress resistance to arsenite. Furthermore, while tracking the thrashing/swimming of worms in a physiological buffer (M9; faded lines), no decline under any conditions in swimming was observed.*

Figure: Survival in 14 mM Arsenite (solid lines) and in physiological M9 buffer (faded lines) of dCAS9 gene inductions.

Thus taken together, this suggests that inducing collagens involved in our feedback mechanisms sufficiently increases lifespan and improvements in healthspan parameters.

3) Lastly, our manuscript already sparked huge interest from the general audience that had read our bioRxiv version. Readers were curious about the conservation of our mechanisms proposed here, since in skin fragility diseases, hemidesmosomes are central for skin integrity, and a recent finding found that YAP1 becomes dysregulated in these diseases. Thus, we were invited to write a general review of our mechanism in human disease settings.

Collin Y. Ewald, Alexander Nyström; *Mechanotransduction through hemidesmosomes during aging and longevity*. *J Cell Sci* 1 August 2023; 136 (15): jcs260987. doi: <https://doi.org/10.1242/jcs.260987>

Therefore, we think that the current manuscript and the discovery of our mechanism are of the highest interest to a general audience.

In summary, we provide the experimental evidence for the proposed mechanism here, not only increasing lifespan but also healthspan during aging and is important for licensing healthspan under longevity conditions.

As said, we agree with the reviewer that healthspan is important. To accommodate the reviewer's request about the relationship between lifespan and healthspan, we summarized the above in the Supplementary discussion for more interested readers and experts.

“Experimental evidence supporting improvements in healthspan with lifespan induced by our proposed collagen homeostasis mechanism

Previous studies, including from our lab, explored two distinct aspects of healthspan regulation: one involving the attenuation of age-related deterioration through the slowing aging by reduced IIS (rIIS) pathway and the other involving the overexpression of certain collagens.

*Firstly, with respect to the rIIS pathway, previous findings provide evidence that our proposed mechanism plays a crucial licensing or assurance role in enhancing healthspan. The age-associated decline in pharyngeal pumping rates, an indication marker of healthspan, is significantly mitigated in *daf-2(e1370)* mutants, which exhibit reduced IIS and longevity²³. However, at older ages (e.g., on day 10 of adulthood), this pharyngeal pumping improvement was entirely abolished upon adult-specific knockdown of *col-120*²⁴, underscoring the indispensability of our proposed mechanism for realizing healthspan benefits. Additionally, higher lipofuscin levels, another aging-associated marker, were effectively curbed in *daf-2(e1370)* mutants but were elevated by *col-120RNAi* at day 10 of adulthood²⁴. Moreover, oxidative stress resilience, a critical determinant of healthspan, was notably increased in *daf-2(e1370)* mutants, yet this benefit was nullified by adult-specific *col-120RNAi*, highlighting the specificity of our proposed mechanism²⁴. Importantly, control experiments involving body length, cuticle leakage, and vulva integrity confirmed that the observed effects were specific to *col-120RNAi*, *col-10RNAi*, and *col-13RNAi*²⁴. Lastly, cuticle stiffness increases with age shown by Coleen Murphy's lab²⁵. This stiffness increase is slowed by *daf-2(e1370)* mutants but completely abolished by *col-120RNAi*²⁵, further substantiating the role of our proposed mechanism in rIIS-mediated healthspan improvements.*

*Conversely, the impact of collagens on healthspan by examining the role of collagen overexpression. Previous results from the Ouyang lab demonstrated that the overexpression of collagens, specifically COL-43 and COL-80, extended lifespan and enhanced resistance to oxidative stress induced by paraquat without affecting cuticle permeability²⁶. This implicates collagen enhancement as a critical determinant of healthspan. Notably, we explored the post-developmental enhancement of collagen expression using dCas9-engineered *C. elegans*, revealing that inducing the expression of collagen genes, such as *col-120* and *col-10*, post-developmentally significantly increased lifespan and improved resilience to heat stress²¹. Upon dCas9-activated *col-120* expression, the small heat shock protein *hsp-16.2* expression was increased²¹. This newfound approach validates that the augmentation of *col-120* expression enhances longevity and healthspan parameters, including thermotolerance. These findings provide valuable insights into the experimental evidence supporting healthspan improvements and emphasize the importance of these mechanisms in modulating healthspan and longevity.*

2. Likewise, the justifications for the lack of additional proteomics data are not sufficient. In particular for points 5 and 7. If the authors admit themselves that the proteomics approach isn't "reliable" then how can we convince others that the mechanism proposed is correct when so

many over-reaching generalisations have been made throughout the study?! Further experimental work is necessary to convince this reviewer of this key aspect of the work.

Response: We listed the previous points 5 and 7 below, and importantly our conclusions do not solely rely on proteomics data but are confirmed by transgenic in vivo data, biomechanical and genetic approaches. Furthermore, in Supplementary Table 9, we list 5 proteomics studies using 5 different longevity interventions validating our findings. Please, see our detailed analysis, including graphs of the additional proteomics data sets in Supplementary Table 9 (see Tabs Longevity_glp1_Pu2017, Longevity_daf2_Depuydt2013, Longevity_dietRestr_Depuydt2013, Longevity_young_metformin_24h, Longevity_young_metformin_48h, LongevityJung21_e1370vsWT_L4, LongevityJung21_isp-1vsWT_L4, Longevity_e1370vsN2_Koyuncu_d1, Longevity_e1370vsN2_Koyuncu_d5, Longevity_e1370vsN2_Koyuncu_d10, Longevity_e1370vsN2_Koyuncu_d15, Longevity_eat2vsN2_Koyuncu_d1, Longevity_eat2vsN2_Koyuncu_d5, Longevity_eat2vsN2_Koyuncu_d10, Longevity_eat2vsN2_Koyuncu_d15). Please, see the response to reviewer 3 for more explanations on the proteomics data. The fact that the proteomics data is not as “reliable” is a technical challenge to the best of our abilities and the current technology existing in the field we had addressed. Therefore, as mentioned before, our conclusions do not solely rely on proteomics but also on transcriptomics, qRT-PCR, genetics, protein-fusion fluorescent reporters, and mechanical modeling.

Here we listed points 5 and 7 with our previous response.

Previous point 5

“5. When discussing the results depicted in Figure 3B, the authors mention that different longevity interventions counteract the age-related changes in protein abundances (lines 210-212). Nevertheless, the graph depicts only the changes for Day 1/Day 2, or even L4 worms, except in the case of eat-2 mutants, where the data for Day 15 are shown. What is the reasoning behind showing all data in one graph given that the abundance of each protein is regulated in an age-dependent manner with distinct expression patterns with age? Ideally, one would show data for proteome changes in older worms across all these longevity interventions. As such, this panel does not sufficiently support the respective claims being made. Since I do not think this is fair to ask for such experiment to be done, I would ask the authors to consider focusing on the longevity interventions that this work primarily focuses on (daf-2 and glp-1) and perform proteomics experiments in late life and at mean/median lifespan for each intervention compared to control.

Response: Fig 3B is a comparison that supports our longitudinal data shown in Fig 3A using daf-2(RNAi). Fig 3B also supports the point that all these 6 different longevity interventions (dietary restriction, metformin, glp-1, daf-2, isp-1, eat-2), although each dataset is patchy and overlap is not perfect, the same main ECM proteins are changed by longevity interventions, broadening our observation from Fig.3A. As mentioned, these 6 longevity intervention data sets are from previously published data (as cited in manuscript and all detailed in Supplementary Table 9). Now most of these interventions are mutations, and the signaling is already different during development compared to wild type. One can already observe this in fig 3D showing that glp-1 mutants have higher total collagen levels than wild-type control on day 0 (=L4). In fact, for instance, the highest difference of wild type compared to daf-2 is seen at L4 for stress resilience

and downstream signaling output. Furthermore, protein homeostasis collapses at 8 hours into the first day of adulthood in *C. elegans* (Please, see Figure 1 of <https://www.ncbi.nlm.nih.gov/pmc/articles/PMC3914504/> which illustrates this best). Thus, although in a perfect world, one would do all these longitudinal time courses of wild type vs longevity interventions, we already can extrapolate from current data that at least for the ECM changes relative to its time point, it is fairly consistently the same major ECM players.

In the manuscript describing these results, we make no statements that Fig 3B are age-related changes but extrapolate in the summary sentence taken together our proteomics time course in the context of previously published data.”

Previous point 7

“7. Also In Figure 3 and the paragraph related to it, the authors argue for existence of a feedback loop that is based on the abundance of a certain collagen (e.g., COL-120, COL-144, COL-10). If the abundances of other matrisome proteins are changed in response to this, and can be modulated via longevity interventions, how do the authors explain that similar changes in levels of these “key collagens” are not present between the longevity models and the worms with COL-120 OE (see panels 3B, 3H, 3K and 3I)? Isn’t this the central tenet of this manuscript? Yet, the data seems not to support this convincingly.

Response: We found that this feedback loop is mediated through mechanotransduction, adjusting collagen levels in the cuticle coupled to physical forces (See explanation by reviewer 1 point 1). In Fig. 3I, overexpression of col-120 leads to upregulation of col-10, col-13, and col-144. Why it is not the same level as col-120 is either because we used a multi-copy array or this is the right collagen level to sync the cuticle stiffness to muscular forces. All these collagens individually are detected in the proteomics data in panels 3B, 3H, 3K, and 3I but not all at the same time. The reason is that proteomics of extracellular matrices is challenging. Since it is a matrix and insoluble, these collagens have to be solubilized, which we achieved with high Urea concentration (8M) and other methodological adjustments (see Methods for details). So proteomics of ECM will always lead to patchy and imperfect data, and thus, if collagen is not detected, one can not say it is absent.

Our data here is supported by 2 independent approaches. First, we used an RNAi screen to identify key regulators of prolonged collagen expression and found the hemidesmosome (also, you can see components of these hemidesmosomes in the proteomics data, such as unc-52). Second, to experimentally address the observation that some genes would induce collagen expression and longevity but others would block enhanced collagen expression and still are required for longevity, we have performed lifespan analyses of over 39 matrisome mutants (including hemidesmosome mutants) of more than 55’499 animals to compare the results to the Pcol-144::GFP expression screen.”

Reviewer #3 (Remarks to the Author):

This manuscript from Teuscher et al. describes the finding that how distinct ECM composition altered with age in *C. elegans* by using different approaches including omics and in-vivo reporter systems. The authors identified a regulatory feedback loop modulating ECM remodeling via mechano-sensitive hemidesmosomes to signal into the cells via YAP-1 to regulate ECM components to enhance the mechanical properties of the surrounding matrices.

The findings are very interesting and provide a valuable contribution to the field of aging. Furthermore, the experiments are well performed overall to make a solid conclusion and the manuscript is also well written. Some important missing points have already been pointed out by other reviewers. In addition to them, I have some minor recommendations to strengthen the manuscript even further before publication.

Response: We thank the reviewer for reading our manuscript and suggesting these minor recommendations. We happily integrated them into our revised manuscript.

1) The authors have shown that there is a decline, unchanged or increase in the protein levels of the majority of cuticular collagen with age in Figures 1B, E and H respectively. Although the comparison between early adulthood and day 8 leads to this conclusion, it seems that some of the cuticular collagen show different trends in their protein level during worms' reproductive stage with age (d0 to d4). The authors should discuss the possible explanation of this phenomenon in discussion. (They provide some information about the extension of the cuticular exoskeleton during early adulthood in line 291-292 but more detailed explanations are required for Figure 1 B,E and H).

Response: We thank the reviewer for pointing this out. We updated the results section to clarify that these changes occur in early adulthood. Furthermore, we updated the previous line 291-2 statement to point out the coincidence with the end of reproduction. However, since these are observations that could posit a hypothesis that we did not test in this manuscript, we refrained from speculating underlying mechanisms that might explain this, such as discussing "collapse of proteostasis that might halt the turnover of collagens, or trade-off scenarios like the disposable soma theory, where resources are allocated to reproduction and taken away from the soma (i.e., the costly collagen production).

"Pattern I consists of 21/41 detected collagens for which the mRNA, protein levels, and abundance in the ECM steeply declined during early adulthood (e.g., col-10, col-12, col-13, col-120, col-144; Figure 1J-K, Supplementary Figure 1I-J, Supplementary Table 1-3)."

"Our observed time course of the collagen mass changes coincides with the decline of pattern I collagens at the final days of reproduction (days 4-8 of adulthood) and with the growth rates in body size during adulthood, whereby after the final molt from L4 to adult, C. elegans continuously grows until day 6-8 of adulthood and then starts to shrink 43-45."

2) The presentation of transcriptional changes in supplementary figures 2 c and f is not easy to read. Although I appreciated that the authors are trying to give an overview of all data, it can be ideal to add additional heat map analysis at least for important candidates which were highlighted.

Response: We agree with the reviewer, and for that reason, a more understandable graph of this data with all the candidates in this manuscript labeled (and for instance, col-120 highlighted in bold) are shown in Figure 1A, 1D, 1G. Furthermore, for curious readers, the processed data values are reported in Supplementary Table 2, easy for anyone to plot their favorite ECM gene. Also, in Supplementary Table 2, we plotted all matrixome genes and labeled them.

3) In Figure 3 B, the authors suggested that longevity interventions increased the normally age-related decrease of collagen levels and dampened the normally age-related elevation of extracellular proteases. The conclusion is not very solid due to early adulthood conditions being analyzed in different data sets (only eat-2 (ad116) day 15 is really supporting this conclusion). Of course, it is not feasible to do new proteomics experiments with all different longevity paradigm models. Therefore, optionally, the authors can also use different proteomics data sets for some of the longevity interventions during aging in the literature. If it is not possible to determine with other proteomics data sets, it would be more accurate to state that longevity interventions may affect collagen protein levels in early adulthood, but it would not be appropriate to attribute this to age-related regulation. Moreover, it is not ideal to show the protein abundance (fold change) from different proteomics datasets in the same graph which can lead to wrong conclusions. I recommend the authors show the data separately for each proteomics data set. They can use different heat maps or volcano plots for each proteomics data set.

Response: Fair point. We corrected our conclusion to early adulthood. We also show all proteomics datasets and graphs of Fig 3B (as a summary: see tab "Fig3B_summary_see right tabs") and, as requested, additional proteomics data sets in Supplementary Table 9 (see Tabs Longevity_glp1_Pu2017, Longevity_daf2_Depuydt2013, Longevity_dietRestr_Depuydt2013, Longevity_young_metformin 24h, Longevity_young_metformin 48h, LongevityJung21_e1370vsWT_L4, LongevityJung21_isp-1vsWT_L4, Longevity_e1370vsN2_Koyuncu_d1, Longevity_e1370vsN2_Koyuncu_d5, Longevity_e1370vsN2_Koyuncu_d10, Longevity_e1370vsN2_Koyuncu_d15, Longevity_eat2vsN2_Koyuncu_d1, Longevity_eat2vsN2_Koyuncu_d5, Longevity_eat2vsN2_Koyuncu_d10, Longevity_eat2vsN2_Koyuncu_d15). Please, see the requested individual graphs that are shown in Fig 3B in Supplement Table 9 with all the other additional proteomics graphs. To avoid any confusion, we state in the Figure legend that it is a composite of different proteomics data and individual data and graphs are found in Supplementary Table 9.

Results:

"Consistent with our proteomics, across six different longevity interventions (dietary restriction, metformin, glp-1, daf-2, isp-1, eat-2) and datasets 34–38, the same signature of an increase of a subset of cuticular collagens (col-) protein levels, collagen-stabilizing and remodeling enzymes (dpy-18, phy-2, bli-, nas-, zmp-) and a decrease of cathepsin (cpl-, cpz-, cpr-) protease levels

(Figure 3B, Supplementary Table 9). We thus propose that longevity interventions mobilize compensatory adjustments in early adulthood to counteract age-related ECM changes, presumably to maintain homeostasis of the ECM proteins. “

Figure Legend 3B:

“(B) Composite of proteomics data showing that distinct longevity interventions increased the normally age-related decrease of collagen levels and dampened the normally age-related elevation of extracellular proteases. Note that individual values of log fold changes (FC) from different proteomics datasets shown here as a composite are not comparable but should indicate the directionality of protein abundance change. For individual volcano plots, data, and details, see Supplementary Table 9.”

4) In Supplementary Table 9, the authors provided a data set that shows the changes enhancing ECM composition that are specific to COL-120 overexpression by comparing different conditions. It is important for the authors to include a comparison between Col-120oe EV and WT EV in the data set, as this would indicate the overexpression effects of COL-120 when compared to the endogenous level of COL-120 expression. This comparison in data set is currently missing and should be provided.

Response: This data is now added to Supplementary Table 9 (ThisStudy_COL120oeEVvsN2EV).

5) The authors performed a quantitative proteomics analysis for comparisons wild type with a P4H dpy-18 (ok162) loss of function mutant treated either with control or daf-2 (RNAi) to analyze the effect of daf-2 depletion on changes in protein abundance ratios. They called this proteomics analysis ‘ECM- enriched proteomics’. However, the term "ECM-enriched proteomic" is not accurate and should be corrected in the manuscript. This is because the study was actually a quantitative whole proteome analysis, not an enrichment or pull-down experiment followed by quantitative proteomics.

Response: To clarify this point, we revised “ECM-enriched proteomics” with “ECM-enriched sample preparation proteomics” throughout the manuscript, as described in DOI: 10.3791/53057.

6) If I don't miss, the authors didn't provide raw data for their proteomics analysis. They just provide output data in a supplementary table. Proteomics data should be deposited in the ProteomeXchange Consortium via the PRIDE partner repository.

Response: Data are available via ProteomeXchange with identifier PXD046470.

Project Name: Mechanotransduction coordinates extracellular matrix protein homeostasis promoting longevity in C. elegans

Project accession: PXD046470

Project DOI: Not applicable

Reviewer account details:

Username: reviewer_pxd046470@ebi.ac.uk
Password: Wz7Mt33r

7) Like Figure 3B, in supplementary Figure 8b, the data from two different proteomics data sets was shown in the same graph. It is crucial to change the figures that show a combination of different proteomics data sets in the same graph. This is because the protein abundance and fold change from different data sets cannot be compared, and it could be very misleading to the audience.

Response: We have split Supplementary Figure 8b into two graphs as suggested by the reviewer. Furthermore, we updated the figure legend.

Figure Legend Supplementary Figure 8B

“(B) Relative increase of UNC-52 perlecan levels during aging quantified from two different proteomics studies (See Supplementary Table 4 for details).”

REVIEWERS' COMMENTS

Reviewer #2 (Remarks to the Author):

Despite the request for additional experimental work, the authors have thoroughly and somewhat satisfactorily justified their answers. I have no further requests.

Reviewer #3 (Remarks to the Author):

I am grateful for the author's effort, and I found that they have mainly addressed my concerns. However, I have a minor suggestion that needs to be addressed. Although the authors have revised the expression "ECM-enriched proteomics" to "ECM-enriched sample preparation proteomics," it is still not entirely accurate. It would be best to point out that this is a quantitative whole proteome analysis and not an enrichment experiment. Unfortunately, current terminology is still misleading for readers. With this minor correction, I highly recommend accepting the paper for publication.

REVIEWER RESPONSE LETTER 3

REVIEWERS' COMMENTS

Reviewer #2 (Remarks to the Author):

Despite the request for additional experimental work, the authors have thoroughly and somewhat satisfactorily justified their answers. I have no further requests.

Response: Thank you.

Reviewer #3 (Remarks to the Author):

I am grateful for the author's effort, and I found that they have mainly addressed my concerns. However, I have a minor suggestion that needs to be addressed. Although the authors have revised the expression "ECM-enriched proteomics" to "ECM-enriched sample preparation proteomics," it is still not entirely accurate. It would be best to point out that this is a quantitative whole proteome analysis and not an enrichment experiment. Unfortunately, current terminology is still misleading for readers. With this minor correction, I highly recommend accepting the paper for publication.

Response: We thank the reviewer and removed "ECM-enriched sample preparation proteomics" from the manuscript.